Manuscript prepared for Atmos. Chem. Phys.
with version 2015/04/24 7.83 Copernicus papers of the LaTeX class copernicus.cls.
Date: 26 April 2021

# The Middle Atmospheric Meridional Circulation for 2002-2012 derived from MIPAS observations

Thomas von Clarmann[1], Udo Grabowski[1], Gabriele P. Stiller[1],
Beatriz M. Monge-Sanz[2,3], Norbert Glatthor[1], and Sylvia Kellmann[1]

[1]Karlsruhe Institute of Technology, Institute of Meteorology and Climate Research,
Karlsruhe, Germany
[2]University of Reading, Department of Meteorology, Reading, UK
[3]Now at Physics Department, University of Oxford

*Correspondence to:* T. von Clarmann (thomas.clarmann@kit.edu)

**Abstract.** Measurements of long-lived trace gases ($SF_6$, CFC-11, CFC-12, HCFC-22, $CCl_4$, $N_2O$, $CH_4$, $H_2O$, and CO) performed with the Michelson Interferometer for Passive Atmospheric Sounding (MIPAS) have been used to infer the stratospheric and mesospheric meridional circulation. The MIPAS data set covers the time period from July 2002 to April 2012. The method used for this pur-
pose was the direct inversion of the two-dimensional continuity equation for the concentrations of trace gases and air density. This inversion predicts an 'effective velocity' that gives the best fit for the evolution of the concentrations on the assumption that an explicit treatment of Fickian diffusion can be neglected. These 'effective velocity' fields are used to characterise the mean meridional circulation. Multiannual monthly mean effective velocity fields are presented along with their variabilities.
According to this measure the stratospheric circulation is found to be highly variable over the year, with a quite robust annual cycle. The new method allows to track the evolution of various circulation patterns over the year in more detail than before. According to the effective velocity characterisation of the circulation, the deep branch of the Brewer-Dobson circulation and the mesospheric overturning pole-to-pole circulation are not separate but intertwined phenomena. The latitude of stratospheric
uplift in the middle and upper stratosphere is found to be quite variable and is not always found at equatorial latitudes. The usual schematic of stratospheric circulation with the deep and the shallow branch of the Brewer-Dobson circulation and the mesospheric overturning circulation is an idealization which best describes the observed atmosphere around Equinox. Sudden stratospheric warmings and the quasi-biennial oscillation cause a pronounced year-to-year variability of the meridional cir-
culation.

## 1  Introduction

The meridional circulation of the stratosphere was discovered by Brewer (1949) and Dobson (1956) and is called the 'Brewer-Dobson circulation' (BDC) today. With its lower and upper branch as well

as the mesospheric overturning circulation, it is the major transport pattern in the middle atmosphere
(Butchart, 2014). As such it governs the distribution of atmospheric constituents in the stratosphere.
The BDC therefore controls the distribution of radiative gases, such as ozone and water vapour, as
well as aerosols, all of which affect major chemistry-climate processes (Dunkerton, 1978; Plumb,
2002).

Possible changes in the intensity of the BDC as a consequence of climate change have been
proposed by, e.g., Butchart et al. (2006). This triggered observation-based studies by Engel et al.
(2009), using balloon-borne observations, and Stiller et al. (2012b), as well as Haenel et al. (2015),
using satellite data. These studies based on satellite data suggest that the true picture of the middle
atmospheric circulation is more detailed and too complicated to be fully characterized by a scalar
intensity of the circulation. Also, offline model simulations driven by ERA-Interim analysis data
confirmed this heterogeneous picture for the stratosphere (Diallo et al., 2012; Monge-Sanz et al.,
2012; Ploeger et al., 2015b). A shift of the entire circulation pattern by $5°$ to the South below 800 K
and a widening of the tropical pipe above that altitude have been detected by Stiller et al. (2017).

The relative importance of transport versus mixing has been investigated by, e.g., Garny et al.
(2014) or Ploeger et al. (2015a). Structural changes of the BDC and evidence of a transition branch
situated below the shallow branch have been reported by Lin and Fu (2013) and Diallo et al. (2019).
(Oberländer-Hayn et al., 2016)derived an upward shift of the BDC from model simulations and
concluded that it could explain the apparent acceleration of the BDC in models.

The polar winter downward branch of the mesospheric overturning circulation, which brings large
amounts of mesospheric $NO_x$-rich air down into the stratosphere where it participates in ozone
chemistry, has been investigated in depth by, e.g., Funke et al. (2005). It has been shown that the
$NO_x$ flux into the stratosphere depends both on the thermospheric source strength which depends
largely on solar particle precipitation and the strength of subsidence of air into the stratosphere in the
polar winter vortex (Funke et al., 2008, 2011, 2014b, a, 2017). In this context, major stratospheric
warmings play an essential role and are coupled with the lower atmosphere (Funke et al., 2010).
Perturbations of stratospheric composition by downward transport of air into the middle atmosphere
have also been investigated by Smith et al. (2011).

The direct comparison of modelled trace gas fields with measured ones is very unspecific with
respect to causes of discrepancies, because it reveals only the consequences of any deficiency in the
model but provides no direct clue how the discrepancies came about. The drawbacks of age-of-air
based methods are the dependence on assumed age of air spectra (see, e.g., Fritsch et al., 2020) and
artificial overaging to unaccounted mesospheric sinks of tracers (see, e.g., Hall and Waugh, 1998;
Reddmann et al., 2001; Ray et al., 2017). Beyond this, age-of-air based methods integrate over the
time an air parcel spent in the stratosphere and thus provide information on the middle atmospheric
circulation only at quite limited temporal and spatial resolution.

In this study we aim to provide a picture of the meridional middle atmospheric circulation better
resolved in space and time than that provided by age-of-air based methods. For this purpose, we
infer effective circulation vectors from measurements of long-lived trace gases from July 2002 to
April 2012 obtained with the Michelson Interferometer for Passive Atmospheric Sounding (MIPAS,
Fischer et al. 2008). From these we calculate multiannual monthly mean circulations along with their
interannual variabilities. Our results contain considerably more specific information on the circula-
tion than the trace gas fields and their variation with time alone. They also provide an understanding
of the circulation better resolved in space and time than the age-of-air method (which integrates over
the time an air parcel spent in the stratosphere).

First we present the methods and data sets used for our analysis (Section 2). This includes a de-
scription of the method of the direct inversion of the continuity equation (Section 2.1), of the trace
gas data sets used (Section 2.2) and our scheme to calculate multiannual monthly mean circulation
fields from the individual monthly circulation fields (Section 2.3). Our derived multiannual monthly
mean circulation fields are discussed along with related variabilities in Section 3. Finally, in Sec-
tion 4, we summarize our results, draw conclusions on their plausibility, and identify possible future
work.

## 2    Method and Data

Stratospheric circulation is inferred in this work from monthly zonal mean mixing ratio distributions
of long-lived tracers by the direct inversion of the continuity equation, using the method by von Clar-
mann and Grabowski (2016). Zonal mean volume mixing ratio fields are calculated from global trace
gas distributions retrieved from limb emission spectra measured with MIPAS. The resulting multi-
annual monthly mean circulation fields are analyzed in terms of first and second moment statistics.

### 2.1    The Direct Inversion of the Continuity Equation

The direct inversion of the continuity equation uses the scheme developed by von Clarmann and
Grabowski (2016), which is called 'Analysis of Stratospheric Circulation Using Spectroscopic Mea-
surements, ANCISTRUS'. This approach avoids certain limitations associated with the traditional
observation-based characterization of the circulation via the mean age of stratospheric air. These are:
(a) no age spectra (Andrews et al., 1999; Waugh and Hall, 2002) are required; (b) the so-called 'over-
aging' due to subsidence of mesospheric air depleted in tracer concentrations (Stiller et al., 2012b;
Reddmann et al., 2001; Ray et al., 2017) does not bias the analysis because observation-based upper
boundary mixing ratios of these gases are used. Any calculation of mixing ratio gradients relies on
measurements of air already depleted in $SF_6$ and no direct reference is made to undepleted tropo-
spheric air. And (c) we provide circulation fields resolved in space and time. By doing so we can
trace back the causes of possible discrepancies between model data and observation-based data better

than with the age-based method. The observational information provided by the age-of-air method is only available as the integrated travel time of the air parcel since it entered the stratosphere.

In the next subsection we shortly summarize the basic rationale behind this approach. Thereafter, updates of the ANCISTRUS inversion method are described.

### 2.1.1 The General Approach

The prognostic formulation of the continuity equation allows to predict subsequent trace gas and air density distributions when their initial values as well as the velocities, mixing coefficients, and source/sink terms are known. We invert this equation to obtain velocities and from given air density and trace gas distributions at different times (In principle it is possible also to determine mixing coefficients from this inversion, but that is not done in the calculations used for this paper). For this purpose, first the prediction step is formalized, using a matrix which contains the partial derivatives of the later atmospheric state with respect to the initial atmospheric state. From this we calculate the Jacobian containing the partial derivatives of the final atmospheric state with respect to the velocities and mixing coefficients. A constrained inversion of the prognostic equation involving the latter Jacobian finally gives the field of velocities and mixing coefficients.

For example, we use monthly mean trace gas mixing ratio fields of March in a certain year, solve the prognostic form of the continuity equation for an initially guessed velocity field to calculate the expected mixing ratio fields for April. The residual between the measured mixing ratio fields for April and the predicted ones contains the information needed to adjust the velocity field. This process is started with all-zero velocity fields and iterated until convergence. The final velocity field is then labelled 'March-April'.

Since inferred velocities, due to correlation of velocities and atmospheric composition, are not the zonally averaged velocities but include eddy transport effects, we call the inferred velocities 'effective velocities'. For further details, see von Clarmann and Grabowski (2016), and Appendix A.

### 2.1.2 Recent Updates and Current Setup

The major amendment to the code since von Clarmann and Grabowski (2016) has been the inclusion of sinks of $CCl_4$, CFC-11, CFC-12, $CH_4$, CO, HCFC-22, $H_2O$, and $N_2O$. For each month, latitude band, and altitude a chemical box model has been run to calculate which fraction of the initial concentration was still present after one month. The following sink reactions were considered: photolysis with TUV-based photolysis rates (Madronich and Flocke, 1998), and reactions with OH, O($^1$D), and atomic chlorine (Sander et al., 2010). The OH concentrations were estimated using the parametrization scheme suggested by Minschwaner et al. (2011). O$^1$(D) mixing ratios were estimated using the steady-state assumption (Equation 5.38 in Brasseur and Solomon 2005), applied to MIPAS ozone. Atomic chlorine was estimated by application of a diurnal cycle to the noon profile shown in Figure 5.50 in Brasseur and Solomon (2005). Inaccuracies in the latter estimates

are deemed tolerable since the related loss reaction is only one of three relevant stratospheric loss reactions (Brasseur and Solomon, 2005).

For $H_2O$ and CO, source reactions were also considered, namely methane oxidation and photolysis of $CO_2$, respectively. In cases where these source reactions outweigh the sinks, the monthly survival rate can be larger than unity. These box model calculations were performed offline and results were tabulated, allowing the ANCISTRUS code to operate with reasonably large time steps. For $SF_6$, no sinks were considered. Since for all species under consideration values at the upper boundary are prescribed using MIPAS measurements, the neglect of sinks above that altitude will not cause artificial 'over-aging' as described by Stiller et al. (2012b). The relevance of the inclusion of sinks and sources is demonstrated in von Clarmann and Grabowski (2021).

While in principle ANCISTRUS is designed to infer both effective 2D velocities and mixing coefficients, in the current version a regularization has been chosen to impose the mixing coefficients to be zero. This choice stabilizes the inversion although it does not provide full information on how mixing propagates onto the velocities. Thus the derived velocities have to be understood as the effective 2D circulation velocities which best describe the redistribution of trace species, under the constraint that Fickian mixing[1] makes no contribution.

The ANCISTRUS method has been validated by von Clarmann and Grabowski (2021), and the validation data have been made available to the public (Grabowski et al., 2020b). The validation study has shown that below 30 km altitude, results are robust even in a quantitative sense. Above, where less reliable measurement information is available, all structures and patterns are still reliably reproduced. Peak velocities, however, are not always reproduced accurately. They are more frequently underestimated than overestimated. The latter effect is attributed to the regularization of the inversion. In no case, the inversion procedure generated artificial circulation patterns which were not present in the reference data. The method proved to be sufficiently robust with respect to missing input data. That is to say, effective velocity differences between a full ANCISTRUS run and a run with information on one particular species missing were considerably smaller than the effective velocities retrieved with a full ANCISTRUS run. For the purpose of this paper, ANCISTRUS proved to be an adequate tool.

## 2.2 MIPAS

The Michelson Interferometer for Passive Atmospheric Sounding (MIPAS; Fischer et al. 2008) is a Fourier transform infrared limb emission spectrometer on Envisat. The sun-synchronous polar orbit of the satellite, with an inclination of about $98.5°$, allowed global measurements of trace gases with dense coverage. MIPAS was operational from July 2002 to April 2012, with some sizeable data

---

[1]We use the term 'Fickian mixing' for any mixing which abides to Fick's law of diffusion. While 'diffusion' (without further qualification) is often understood as a physical process on a molecular scale, Fick's law is also applicable to some macroscopic processes. For this reason we think that the term 'Fickian mixing' is more adequate in this context than 'diffusion'.

gaps in 2004 to 2006. Due to a failure of the interferometer mirror slide in 2004, operation at high spectral resolution was stopped in March 2004. In January 2005 operation was resumed, however at degraded spectral resolution. This went along with an improvement of spatial sampling. The altitude

coverage of useable tangent altitudes in the nominal measurement mode of MIPAS ranges from cloud top altitude to the middle mesosphere. Data products relevant to this study are temperature and $H_2O$ (von Clarmann et al., 2003, 2009), $CH_4$ and $N_2O$ (Plieninger et al., 2015), CFC-11, CFC-12 (Kellmann et al., 2012), HCFC-22 (Chirkov et al., 2016), $CCl_4$ (Eckert et al., 2017), $SF_6$ (Stiller et al., 2012b; Haenel et al., 2015), and CO Funke et al. (2009). The products have been widely

validated, e.g., by Stiller et al. (2012a); Plieninger et al. (2016); Eckert et al. (2016), just to name a few.

### 2.3 The Multiannual monthly mean middle atmospheric meridional circulation

From MIPAS measurements of the considered trace species, monthly zonal mean distributions were calculated for latitude/altitude bins of 6°/3 km. Monthly distributions were available from July 2002

to April 2012 with data gaps as reported above. Therefore, the velocity field of each month is constructed from its own set of years of available data. For each pair of subsequent months, two-dimensional circulation fields were calculated, using the ANCISTRUS-tool described above. This resulted in 89 circulation fields, the first representing August to September 2002, the latest March to April 2012. Then, all January to February fields were averaged, all February to March fields, etc., to

form the 12-monthly data set.

The use of language is not uniform in the community. For the description of the figures, we use the following terminology: The 'overturning circulation' we understand is the mesospheric pole to pole circulation, consisting of one single rotation cell and mainly driven by gravity wave breaking (Plumb, 2002; Dunkerton, 1978).

The 'deep branch of the BDC' is the circulation from the equator to the poles in the middle/upper stratosphere with uplift in the tropics and subsidence over the poles, (Plumb, 2002; Birner and Bönisch, 2011; Bönisch et al., 2011). For the transport pattern from the tropics to midlatitudes in the lower stratosphere we use the term 'shallow branch of the BDC'.

### 3 Results

Figures 1–2 show the resulting circulation fields in the full altitude range from 6 to 68 km. In the supplement, all figures of this paper are reproduced with a colour scale that is better legible for readers with a non-standard colour perception. We also show the circulation patterns for altitudes up to 30 km only (Figs. 3–4), where the reduced altitude range along with the reduced maximum velocities allows to better discern the smaller effective velocities found in the UTLS and troposphere.

The years which went into the mean fields are indicated. Missing years are chiefly attributed to

MIPAS data gaps. Only in a few cases (November-December 2003, March-April 2007, December-January 2009, March-April 2009, and June-July 2011 the inversion did not converge or another technical problem was encountered.

Since these monthly circulation patterns are built from multiannual averages covering the period August 2002 to April 2012, the following characteristic has to be kept in mind when interpreting these results. A strong circulation feature which appears in every year but not always exactly at the same latitude, altitude, or time appears weaker in these multiannual monthly means. Conversely, a weaker pattern, which appears every year at the same latitude, altitude, or time will appear stronger.

To diagnose this effect, the standard deviations of the components of the circulation vectors, which are a measure of their variability, are also shown for meridional and vertical effective velocities, respectively, in Figs. 5–6 for January to June, and in Figs. 7–8 July to December. These variabilities are caused both by the natural variability of the circulation over the years and its random uncertainty. The latter is the random uncertainty of the MIPAS measurements propagated onto the circulation vectors.

## 3.1 An average year of middle atmospheric circulation

### 3.1.1 January-February

The circulation pattern inferred from the change of trace gas distributions from January to February shows two major circulation cells with opposite rotation (Fig. 1, upper left panel). In the Northern hemisphere (NH) there is strong transport from the Southern tropics (up to 30°S) to the Northern subarctic latitudes, with maximum effective velocities in the upper stratosphere, between 40 and 45 km altitude. This can be associated with the upper branch of the BDC. Separated by a local minimum of effective velocities at 30 km altitude, there is a further branch of the Brewer-Dobson circulation in the lower to middle stratosphere. While its effective velocities and vertical extension are smaller, due to the larger air density at these lower altitudes smaller velocities still can transport considerable airmass to higher latitudes. The direct vertical motion over 30°S suggests the existence of a region where horizontal transport is minimal compared to vertical transport; the location of this region is in good agreement with the location of the subtropical transport barrier (e.g., Stiller et al., 2017).

Above 50 km at Northern polar latitudes there is some subsidence. Associated year-to-year variability in vertical effective velocities is large, reflecting the irregular appearance of sudden stratospheric warmings (Fig. 6, upper left panel). Their irregular occurance and the related impact on subsidence is discussed, e.g., in Funke et al. (2014a). Large variability over the North pole at stratospheric altitudes does not come unexpected, since Haenel et al. (2015, see their Fig. 9) found in their age-of-air time series analysis largest amplitudes of the signal of the quasi-biennial oscillation (QBO) at polar latitudes. Strahan et al. (2020, and references therein) highlight the importance of the QBO for stratospheric circulation. Baldwin et al. (2001) and Baldwin et al. (2021) also discuss

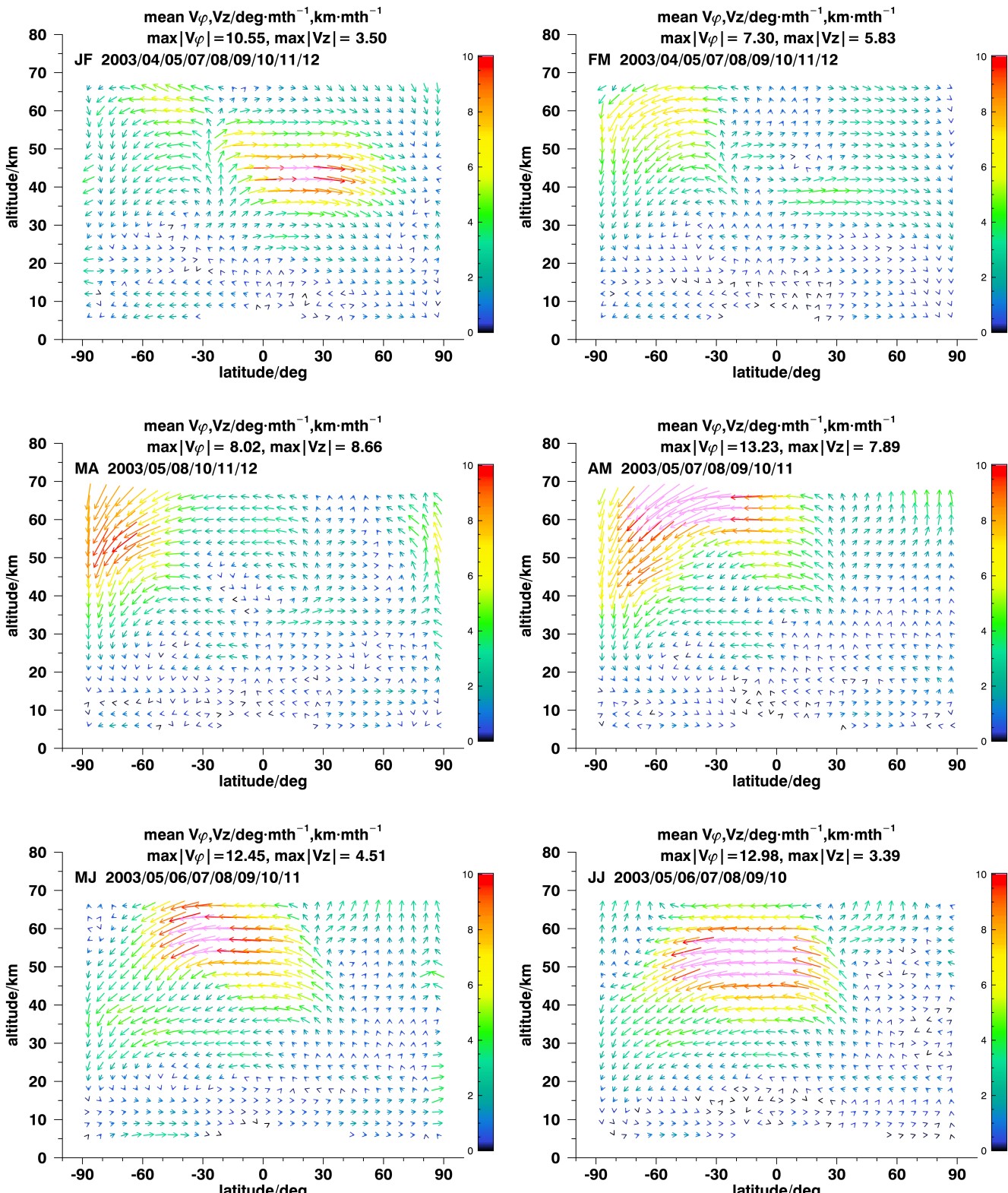

**Figure 1.** : Mean monthly circulation patterns from January–February (top left, JF) to June–July (bottom right, JJ). The headers give quantitative information about maximal effective velocities and the months and years considered. Missing years are due to MIPAS data gaps and non-converged inversions. The colour scales refer to $\sqrt{(v_\phi \mathrm{deg}^{-1}\mathrm{mth})^2 + (v_z \mathrm{km}^{-1}\mathrm{mth})^2}$ for $v_\phi$ and $v_z$ in units of $\mathrm{deg}\,\mathrm{mth}^{-1}$ and $\mathrm{km}\,\mathrm{mth}^{-1}$. Pink arrows refer to velocities higher than representable by the colour scale chosen.

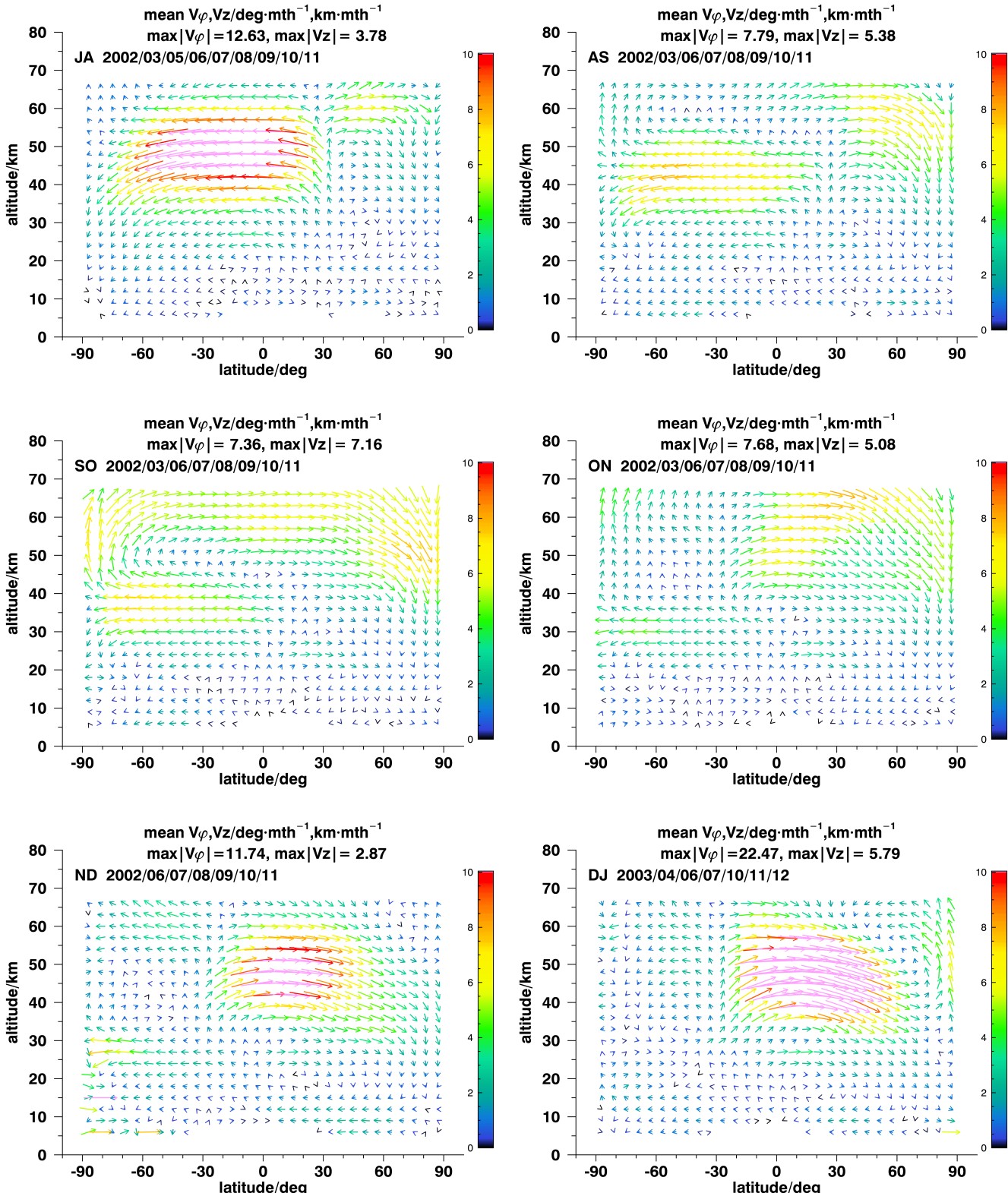

**Figure 2.** : Mean monthly circulation patterns from July–August (top left, JA) to December-January (bottom right, DJ). For details, see Fig. 1.

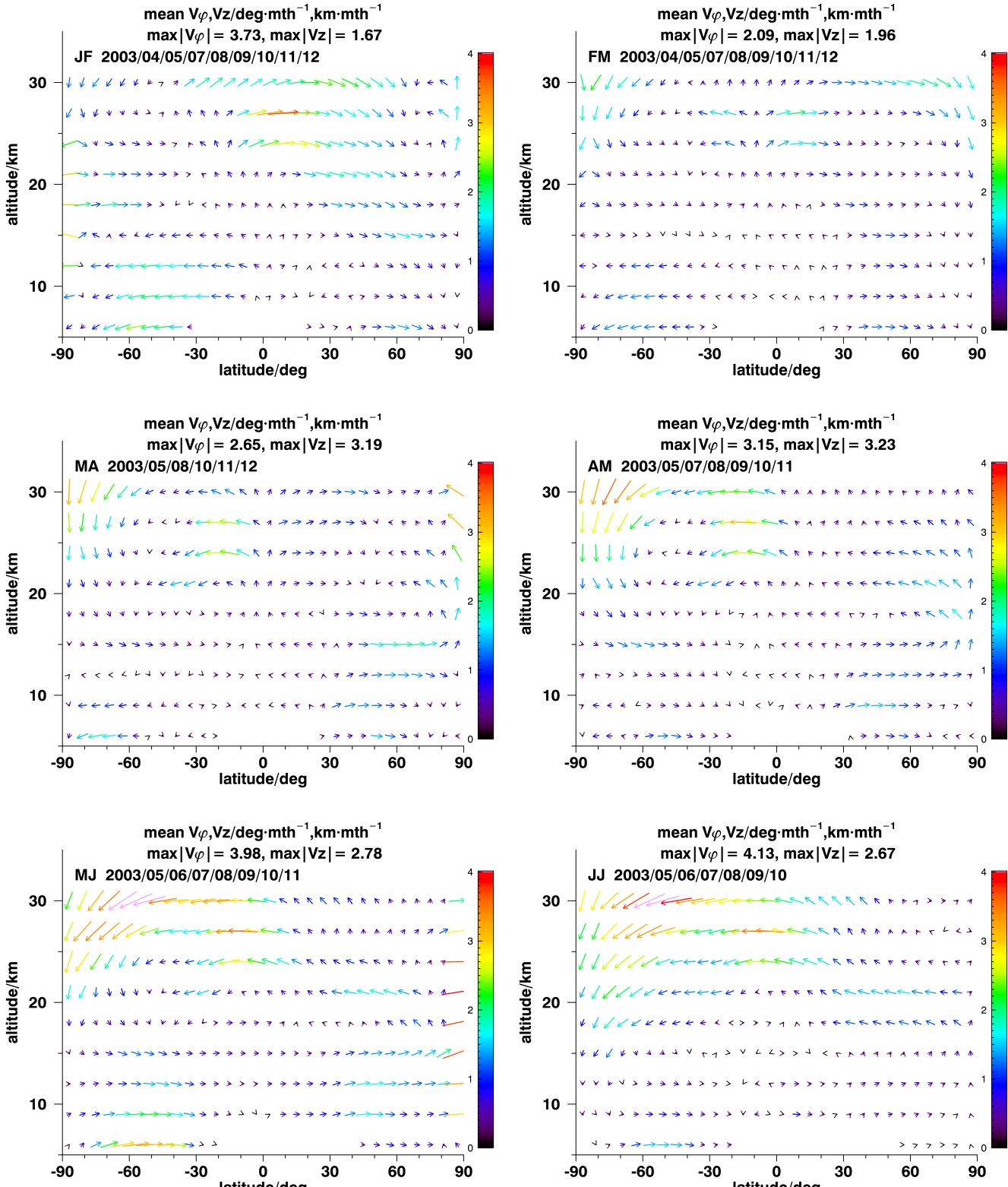

**Figure 3.** : Same as Fig. 1 but for altitudes up to 30 km only.

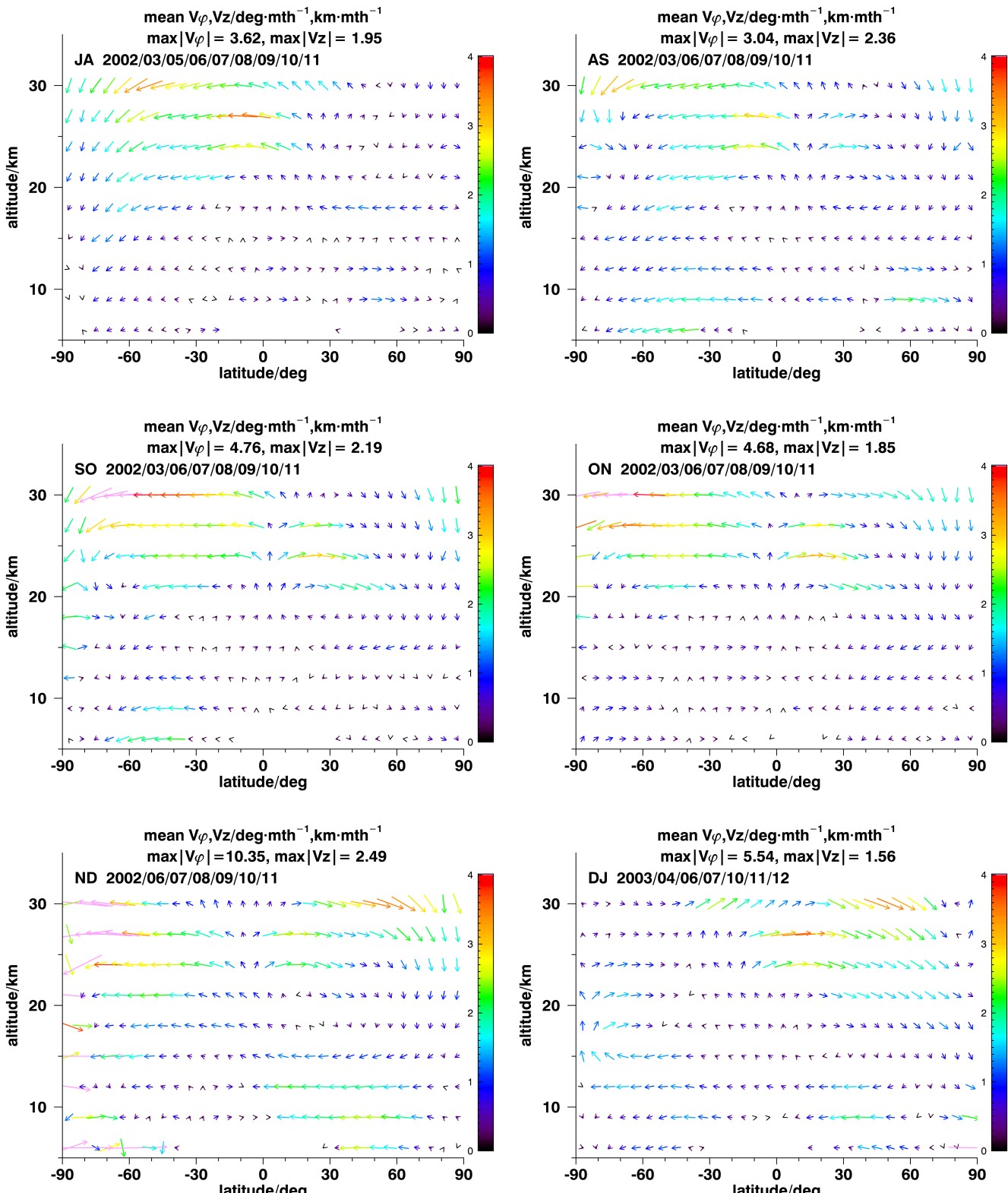

**Figure 4.** : Same as Fig. 2 but for altitudes up to 30 km only.

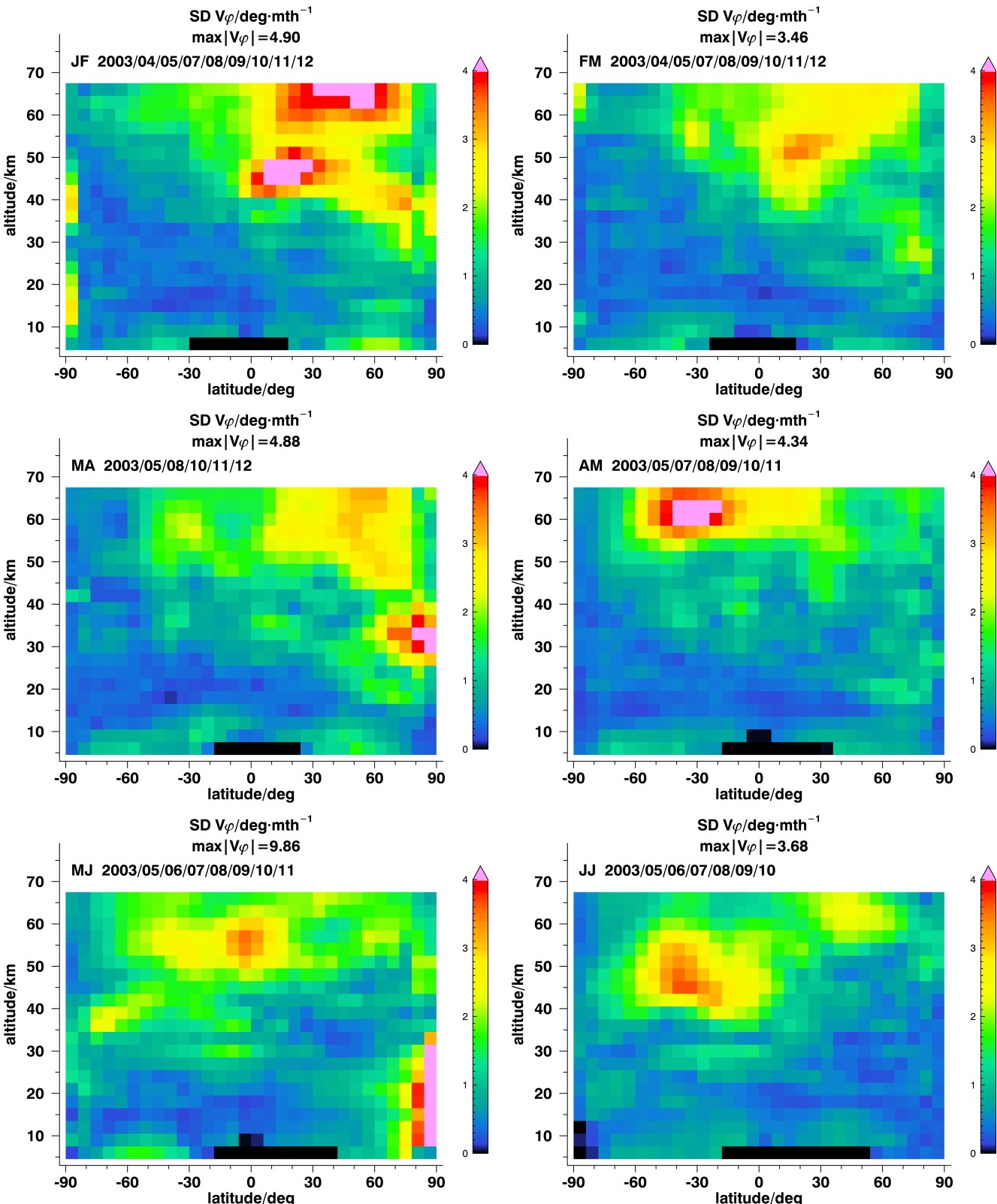

**Figure 5.** : Interannual variability of the middle atmospheric meridional effective velocities in terms of sample standard deviations from January–February (top left, JF) to June–July (bottom right, JJ).

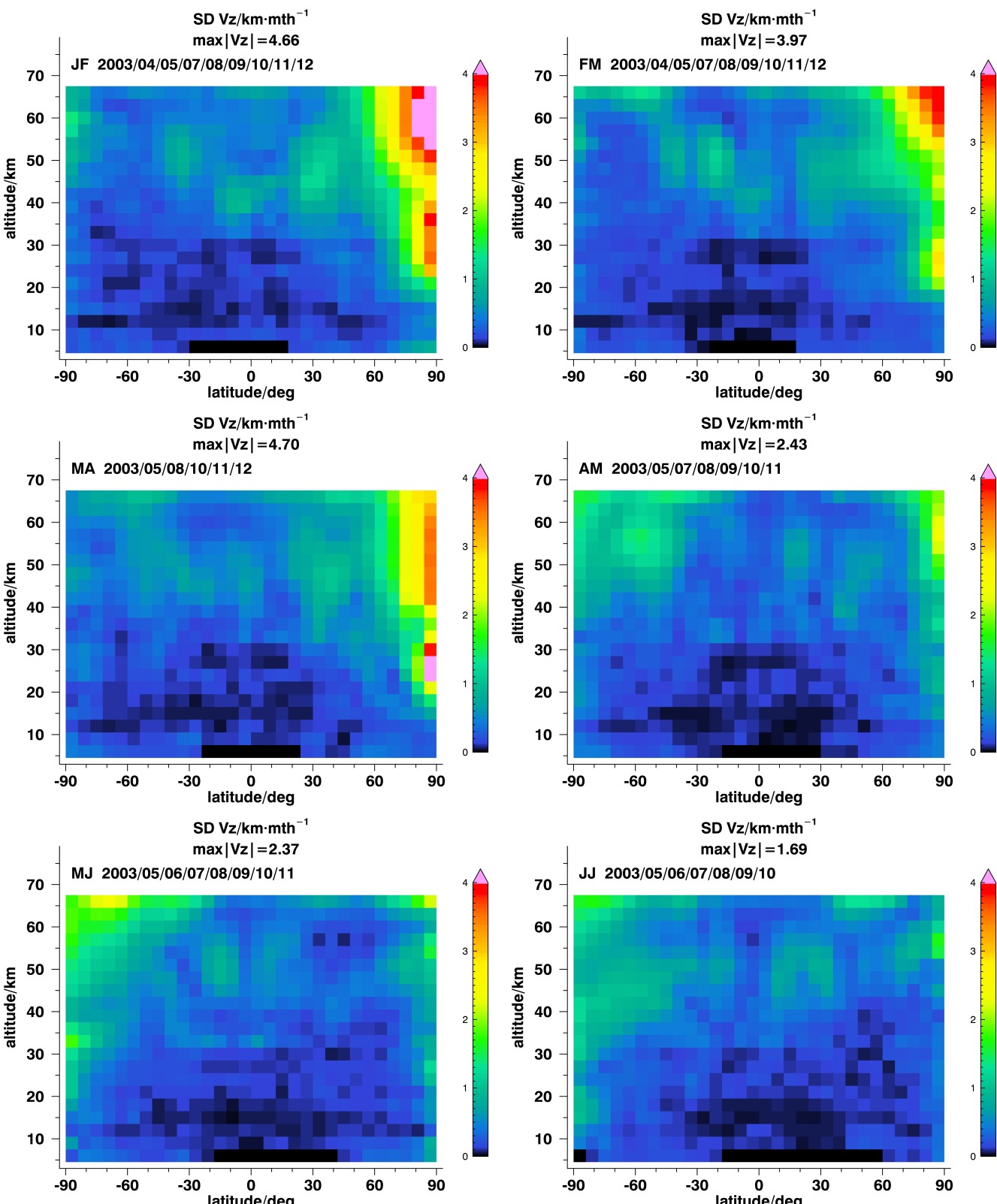

**Figure 6.** : Interannual variability of the middle atmospheric vertical effective velocities in terms of sample standard deviations from January–February (top left, JF) to June–July (bottom right, JJ).

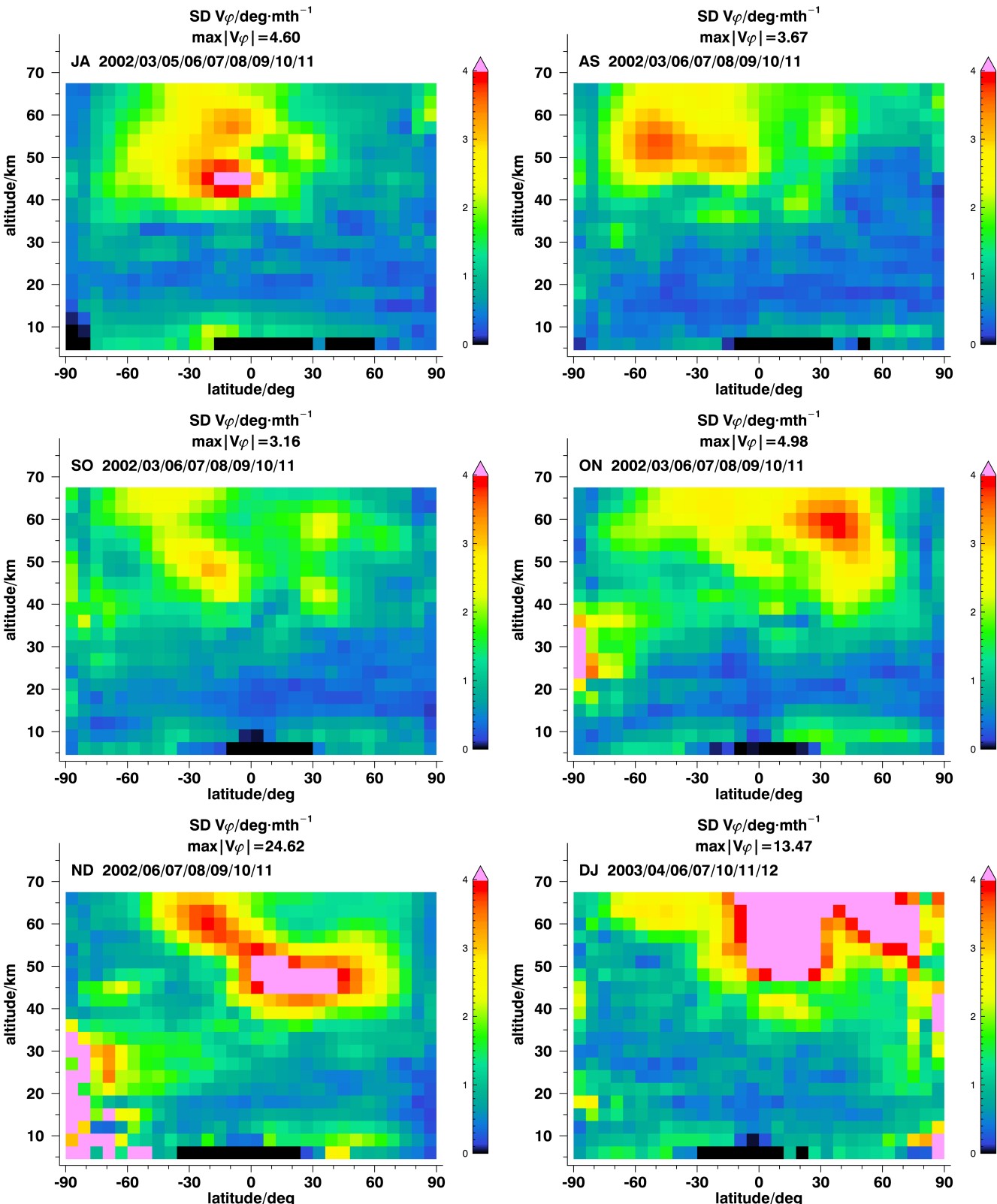

**Figure 7.** : Interannual variability of the middle atmospheric meridional effective velocities in terms of sample standard variations from July–August (top left, JA) to December–January (bottom right, DJ).

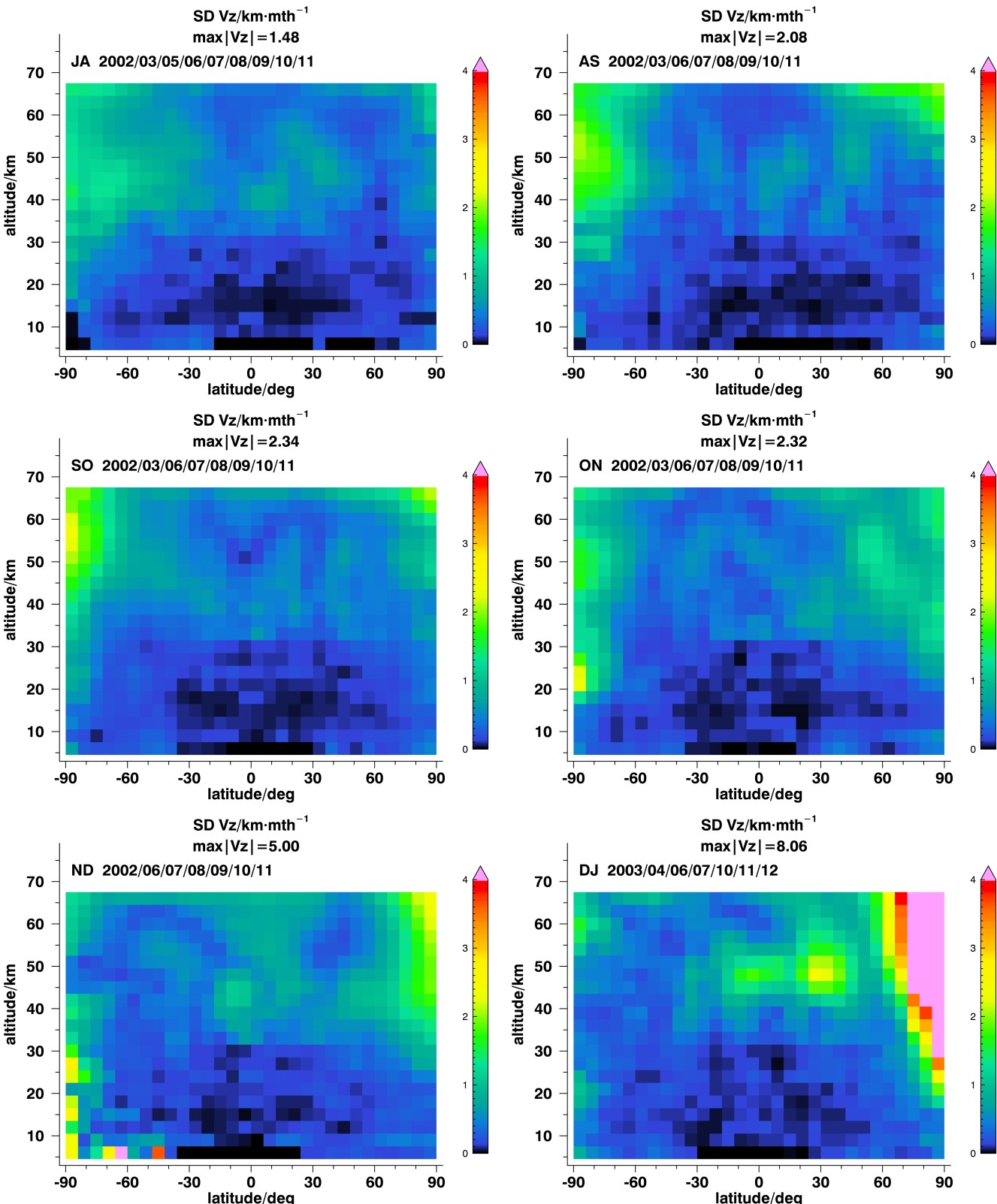

**Figure 8.** : Interannual variability of the middle atmospheric vertical effective velocities in terms of sample standard variations from July–August (top left, JA) to December-January (bottom right, DJ).

the possible interaction between the QBO and sudden stratospheric warmings and mention mesospheric QBO effects.

In the middle and lower Northern polar stratosphere there is no clear signal of subsidence (Fig. 1, upper left panel). This can be explained by the fact that the northern polar vortex is known to be regularly displaced from the pole, which often causes subsidence effects to be averaged out when latitudinal averages are considered. Representation in equivalent latitudes would be more adequate to analyze this phenomenon but since that representation would not be optimal for global analyses, it is deferred to a future study.

At the same time, we find in the Southern hemisphere (SH) a region of southward/poleward transport around 60 km altitude starting around 30°S, and subsidence over high Southern latitudes in the mesosphere and upper stratosphere. It would be bold to associate it with the Brewer-Dobson circulation, because measured temperature profiles suggest that the stratopause during this time is at about 50 km. Instead, this is a mesospheric branch of an Equator-pole circulation. Most parts of the SH stratosphere are quite calm. Low variabilities indicate that this is a very typical condition. At Southern midlatitudes there is an isolated cell of pronounced poleward transport at tropospheric altitudes between 8 and 12 km (Fig. 3, top left panel). We associate this with the transition branch of the BDC reported by Lin and Fu (2013) and Diallo et al. (2019).

Tropical uplift, which separates the anti-parallel circulation patterns described above, has its maximum at 50 km altitude and at 30°S. At lower altitudes effective uplift velocities are small and maxima seem to be found further equatorward compared to higher altitudes. This leads to a warped tropical pipe.

In the altitude range covered by our data no indication of a pole to pole overturning circulation is seen.

Except for the highly variable subsidence at Northern polar regions and its feeding in the Northern midlatitudinal mesosphere, inferred velocities largely exceed their variabilities by a factor of 3, indicating that these results are robust and, due to only moderate year-to-year variability, are fairly representative for the atmospheric state at this time of the year.

### 3.1.2 February-March

The upper stratospheric transport pattern from the tropics to the North pole seen in January to February around 40 km has considerably weakened, and at least in the middle stratosphere (around 30 km) there is now a stronger signal of Northern polar subsidence (Fig. 1, right upper panel). The SH circulation pattern has become considerably stronger. Southern polar subsidence is more clearly pronounced than the month before and is seen to reach lower altitudes. Effective velocities decrease towards lower altitudes which is an immediate and expected effect of increasing air density during subsidence. The NH middle stratospheric poleward circulation pattern below 30 km remains intact,

and a SH counterpart has emerged. The deep branch of the BDC starts to form from the tropical region above 20 km.

Roughly following the seasonal movement of the solar zenith, the tropical pipe has moved slightly towards the equator. Between 40 and 45 km there is still a poleward offset of the latitudinal region of uplift which separates the northern and southern circulation patterns at 18°S. Between 45 and 55 km the upward movement bifurcates into a Northern and Southern transport branch. The Northern one feeds into a mesospheric circulation pattern that exists above the deep branch of the BDC (at 55 km and above), while the Southern branch feeds into the dominating circulation pattern of the SH upper stratosphere and mesosphere. At 55 km a reverse offset of the latitude of strongest uplift is observed towards northern latitudes. This movement will proceed further in the following months and will give rise to a mesospheric overturning pole-to-pole circulation later in the year.

Shallow branches of the BDC are seen in both hemispheres. In northern midlatitudes this pattern extends higher up into the stratosphere while in the SH it is confined to the lowermost stratosphere. Arguably, the classification of this feature as the lower branch of the BDC is not straightforward because it is not clear enough whether this feature is fed by the tropical uplift. In the lower southern midlatitude stratosphere there is even an altitude region ($\sim$ 15 km) where equatorward backflow is observed which possibly closes the loop of the circulation above.

Again the regions of large variability are confined to high Northern polar latitudes and a small region at 53 km altitude, 26° latitude. This again shows moderate year-to-year variability and representative results elsewhere.

### 3.1.3 March-April

An overturning circulation feature forms which is also fed by tropical air above about 30 km (Fig. 1, left middle panel). The uplift barrier between 40 and 60 km has moved to 30°N and makes this overturning circulation feature bifurcate into two branches, one developing into the deep pole-to-pole circulation above 65 km, and another one from the tropics to the southern high latitudes below 65 km. In the SH, a deep BDC branch has clearly developed from the subtropical pipe cell above 20 km (Fig. 3), and adds poleward and subsidence motion to the overturning circulation branch over the Southern Pole. This subsidence reaches further down than in the preceding month. At 15 km altitude there is some back-flow of southern polar air towards lower latitudes.

In the NH, the overturning circulation seems to be fed from a second altitude region. Besides the deep branch of the BDC around 35 km, poleward transport in the lower stratosphere (10 - 15 km) from the subtropics (30°N) to the high latitudes is feeding into the uplift over the North Pole as well. Above 15 km the arriving air is uplifted, while air arriving at lower altitudes subsides (Fig. 3,middle left panel).

The structure of circulation over the NH midlatitudes leaves an isolated region between 35°N-65°N in the altitude range of 30-65 km without a sizeable vertical velocity component (Fig. 1).

This feature will evolve in the following months as a region where uplift motion clearly overtakes horizontal transport around 60°N.

Large variability in the lower stratosphere in high northern latitudes (Figs. 5–6, middle left panels) is attributed to late winter vortex dynamics.

### 3.1.4 April-May

In the southern polar stratosphere there is still strong subsidence (Fig. 1, right middle panel). Maximum velocities have shifted higher up (around 70 km) and towards lower latitudes. The northern polar uplift branch of the mesospheric overturning circulation has also moved upwards and broadened so that it covers NH middle and high latitudes now. It does not show the tropical origin anymore, i.e., this part of the pole-to-pole circulation branch is now disconnected from the tropical pipe. Instead, a strong transport branch from the Northern subtropics (30°N) to the South Pole has developed for altitudes above 40 km. This transport path avoids the detour over the Northern polar latitudes but feeds into the mesospheric circulation at about 30°N.

In the Northern lower stratosphere between 10 and 25 km a circulation has established with poleward transport below 15 km and upward and southward transport between 15 and 25 km (see Fig. 3, middle right panel). In the SH there exists a second transport branch from the equator to subarctic latitudes between 20 and 30 km, the deep branch of the BDC, that has strengthened compared to the previous month (see Fig. 3, middle right panel). As in the preceding month, some back-flow of polar air towards low latitudes is detected in the SH at 15 km, most pronounced in sub-polar regions. Large interannual variability is observed above 55 km at SH mid and lower latitudes (Fig. 5, middle right panel), related to the varying vertical extent of the pole-to-pole overturning circulation.

### 3.1.5 May-June

The mesospheric branch of the BDC in the SH is still present, now with maximum velocity shifted towards subtropical latitudes and lower altitudes (Fig. 1, lower left panel); this circulation branch now causes subsidence over SH midlatitudes at higher levels. At about 30–40 km descending air encounters the lower deep branch of the BDC. Air is further transported towards the polar region where these merged branches of the BDC result into subsidence. While in the preceding month subsidence over the Antarctic was observed up to the highest altitudes, now the circulation above 60 km starts to reverse, which will lead to a south-north pole-to-pole circulation in the next six months. There is indication of a reversal of the shallow branch in the SH with strongly increased equatorwards transport velocities (compared to the previous month) in the range below 15 km.

Again, the NH is quite calm in terms of absolute velocities but the circulation of air with transport towards the pole below 15 km and back to the mid-latitudes and subtropics around 20 km has strengthened (Fig. 3, lower left panel). The region of vertical transport around 30°N has moved down

to altitudes between 30 and 50 km. Vertical uplift in between 40°N - 60°N in the altitude range of 40-60 km feeds the SH mesospheric circulation branch.

### 3.1.6 June-July

A clear pattern of the deep branch of the BDC has established in the SH, with transport from the Northern subtropics (35°N) to the South Pole and subsidence there below 40 km (Fig. 1, lower right panel). Relatively large variability in the range of $30° - 60°$ S and 45 to 55 km altitude indicates that the extent of high transport velocities towards the South Pole varies from year to year (Fig. 5, lower right panel). A very weak backward flow to the equator is present below 13 km (Fig. 3, lower right panel). The southern polar uplift now starts already at 50 km altitude. It will intensify during the following months. Eventually, three months later, it will form the upwelling branch of the overturning pole-to-pole circulation.

The middle to upper stratosphere in the NH is rather quiet. Some uplift is still present in mid-latitudes above 55 km that is fed from the subtropical region (30°N). An area of uplift from 25-50 km above 30°N forms a horizontal transport barrier in the sense that air southward of this barrier is transported in direction to the South Pole; this feature persists until August-September.

The circulation pattern in the lower Northern stratosphere is still present. The poleward transport in the lower branch as observed the month before has weakened. Now we observe an upwelling pattern from 30–40°N that transports air upwards and polewards up to around 75°N before turning towards the tropics at about 20 km altitude (Fig. 3, lower right panel). This feature is also observed in the following month, but loses strength in August-September, when this pathway is overpowered by the descent of the already forming overturning circulation in the NH high latitudes. The low standard deviations of meridional effective velocities, which are about a factor of three smaller than the effective velocities in this region and time of the year, indicate that this is the usual pathway tracers follow (Fig. 5, lower right panel.). This, together with the lack of a SH counterpart, allows us to link this pattern to the occurrence of the Asian Monsoon, which has been recently shown to be an effective pattern for transport of tropospheric tracers into the stratosphere (e.g. Randel et al., 2010; Ploeger et al., 2017; Yu et al., 2017). Our results are in agreement with Fig. 5 shown by Ploeger et al. (2017) with respect to the circulation structures. Differences related to the transport of tracers during the monsoon period can be attributed to the different time resolution in our study and theirs. A clear monsoon signal is visible in MIPAS data resolved in time and longitude (See, e.g., Vogel et al., 2019), and is obviously strong enough to survive zonal averaging.

### 3.1.7 July-August

In general, the circulation patterns resemble those of the previous month, although shifted towards lower altitudes (Fig. 2, upper left panel). The only exception is the uplift barrier at 30°N which is displaced to higher levels. The circulation branch above 50 km from 30°N to the poles is strength-

ening. While in the previous month this branch seemed to feed chiefly the overturning pole-to-pole circulation, it now establishes the onset of northern polar subsidence, which will continue to gain importance during the subsequent months. That is to say, what looked like overturning circulation the month before becomes more and more a BDC-like symmetric feature. The maximum updraft is located at 30°N. In the second half of the year, the patterns observed during the first six months are largely mirrored to the other hemisphere, however shifted in altitude, and the transport velocities are somewhat larger. The July-August patterns resemble widely the mirrored patterns seen in JF (Fig. 2, upper left panel). The pronounced deep branch of the BDC resides in the middle to upper stratosphere of the SH (35 to 60 km). A zone of vertical transport around 30°N and between 30 and 55 km forms a horizontal transport barrier. Above this barrier, upward and poleward transport with a source region in the Northern subtropics has established. Indication of uplift over the South Pole is seen above 60 km.

The circulation pattern in the Northern lower stratosphere with poleward transport below 15 km and equatorward transport around 20km is still existent. As in the previous months we assign this circulation cell to the NH monsoon systems (Fig. 4, upper left panel).

The variability of meridional effective velocities in the upper stratosphere over tropics and southern midlatitudes is similar to the NH counterpart in January-February (Fig. 7, upper left panel). Although small, there is a sizeable variability in vertical velocity over the South pole related to year-to-year variability of the descent inside the polar vortex (Fif. 8, upper left panel).

### 3.1.8 August-September

The general circulation pattern resembles that of the preceding month (Fig. 2, upper right panel). However, the maximum of peak velocities of the SH circulation branch are now found at lower altitudes. A bifurcation of the circulation is found near the southern polar stratopause. This poleward circulation branch around 45 km altitude feeds both the mesospheric overturning circulation which now has established in south north direction, and the southern polar subsidence. In the northern polar upper stratosphere and mesosphere strong poleward transport and subsidence starts to establish, which is fed by the mesospheric overturning circulation, by tropical uplift south of 30°N from above 55 km, and by Northern mid-latitude air from 35 to 55 km. Meridional velocities in the winter hemisphere are stronger in the 0-70°S, 40 km altitude range in August-September compared to the NH counterpart in February-March.

In the lowermost stratosphere, we see isolated shallow branches of the BDC below 15 km transporting air from the subtropics to the poles in both hemispheres (Fig. 4, upper right panel). Similar to February-March and March-April, vertical transport is present over the equator up to about 27 km; this vertical transport feeds into two rather weak circulation cells in midlatitudes below 25 km. In the SH this circulation merges with the poleward transport further down, while in the NH, the

circulation merges with the weakened equatorward transport around 20 km that was observed in the previous months.

We can see a reversal in the horizontal velocities in the NH lower stratosphere above about 10 km from around 40°N to the equator. We associate this with Asian-monsoon-related transport.

### 3.1.9 September-October

The circulation in September-October is dominated by a pronounced overturning pole-to-pole circulation feature, being lower in altitude than its boreal spring counterpart (Fig. 2, middle left panel). All transport above 55 km is directed upwards and towards the North Pole. The deep branch cell of the BDC in the SH observed in the previous months has moved further down (25 to 40 km) and further polewards and feeds, by bifurcation, into the overturning circulation in its upper part. The lower part still feeds into the subsidence area in the middle to lower South polar atmosphere (below 30 km). The uplift region that forms a horizontal transport barrier is now located between 25 and 40 km and closer to the equator, at about 20°N. Above this barrier, all transport in the NH is directed towards the North Pole. A second region of purely vertical transport is found directly above the equator between 20 and 30 km. This region of vertical transport separates two weak circulation cells in both hemispheres. The SH one merges with the deep branch of the BDC, while the NH counterpart provides poleward and downward transport towards 60°N and then turns into equatorward transport around 15 to 20 km. In the lowermost stratosphere and UTLS (below 13 km, Fig. 4, middle left panel) we observe some poleward transport in the SH while the NH atmosphere in this altitude range is rather quiet.

### 3.1.10 October-November

The transport cell related to the deep branch of the BDC in the SH has weakened considerably, and has shrunk and moved further down to 25 to 35 km (Fig. 2, middle right panel). Although its upper part still feeds into the uplift over the South Pole, it is now more separated from the above regions and more clearly causing also air descent over the South Pole.

The horizontal transport barrier formed by vertical transport has jumped back to the SH and is located around 30°S from this month on (until March-April). Its altitude range is again 45 to 65 km. It creates an isolated region at 40° − 60°S and 40-55 km, similar to the one found in the NH during March-April, although smaller now. All transport above 40 km and northward of 30°S is directed towards the North Pole and leads to subsidence there. Increased upward velocities over the South Pole above 55 km indicate the existence of an overturning mesospheric pole-to-pole circulation, but this seems to take place at altitudes mainly above 68 km and can thus not be diagnosed here. In contrast, a tropical feeding of the former overturning circulation takes over and will develop in the subsequent month into the deep branch of BDC in the NH. A second region of purely vertical transport is present over the equator between 20 and 30 km and acts as horizontal transport barrier

there. It is flanked by a weak NH circulation cell transporting air into midlatitudes above 20 km and equatorwards below 15 km. (Fig. 4, middle right panel).

Largest variability at 20 to 35 km over the South pole is related to interannual variations in the timing of the SH polar vortex break-up, associated with the high values seen in Figures 7–8 (middle right panels). This had already started in the previous month in the upper stratosphere.

### 3.1.11 November-December

The most pronounced feature in this month is the deep branch cell of the BDC in the NH at very high altitudes (35 to 60 km), transporting air from the Southern subtropics ($30°$S) to the North Pole (Fig. 2, lower left panel). Subsidence over the winter pole is present below 50 km. Some upward and Southern poleward transport is present over the SH midlatitudes above 55 km, similar to the May-June and June-July situation in the NH. A very weak remnant of the deep branch of the BDC is present in the SH between 20 and 30 km. In the lowermost stratosphere, a reversal of the shallow branch of the BDC with transport towards the equator can be seen (Fig. 4, lower left panel). This is especially clear for the NH. This equatorward transport seems to be fed by the subsidence over the North pole. A similar but weaker feature was observed in the SH in March-April.

### 3.1.12 December-January

The deep branch of the BDC is still seen as the main circulation feature near the Northern tropical and midlatitudinal stratopause. However, it has shrunk in latitudinal extension and feeds the subsidence over the North Pole only below 25 km (Fig. 2, lower right panel). In the northern polar upper stratosphere the vertical velocity has reversed, showing upward velocities. Interannual variability, however, is high here, probably caused by frequent sudden stratospheric warmings appearing at different times of the NH winter (Figure 7, lower right panel). As discussed in Section 3.1.1, the QBO is another important driver of the interannual variability of circulation.

In the shallow branch of the BDC, equatorward transport in the NH midlatitudes below 15 km and poleward transport in the SH at the same altitude range is present. In the SH, the circulation pattern is closed by upward and equatorward transport in the 15 to 25 km region (Fig. 4, lower right panel).

### 3.2 Summary

We have presented here a new data-set of multiannual monthly mean middle atmospheric circulation fields derived from long-lived tracer measurements from MIPAS. These circulation fields have been constructed from the results of independent ANCISTRUS runs (von Clarmann and Grabowski, 2016) with a latitudinal/vertical resolution of $6°$/3km; resulting fields are stable over the years of the MIPAS mission (2002–2012) in the sense that the annual variation of the resulting circulation patterns is large only in regions where large interannual variability is expected based on current theory. The stability of results from independent ANCISTRUS runs increases confidence in the robustness

of the analysis method in the sense that it produces similar results for similar input fields. This fur-
nishes evidence that the results do not depend in any sizeable way on the MIPAS sampling in a
particular year. Other phenomena, which also appear on a regular basis but vary in the exact latitude
or time where/when they appear, average out. This issue is further discussed in Section 3.2.3. In the
following, we present a synopsis of the main phenomena found.

### 3.2.1   The BDC and the Overturning Circulation

The upper branch of the BDC and the overturning pole-to-pole circulation are not, according to our
data, two independent phenomena as suggested by the schematic shown in Fig. 1 Bönisch et al.
(2011), but we observe quite smooth transitions between both. While from July to September there
are still two major, roughly antiparallel circulation cells below 68 km, in September-October we
have one single northward circulation pattern above 50 km. Our data are consistent with - but do not
directly support - a southward pole-to-pole circulation from March to May at altitudes above those
covered by MIPAS data.

From our results, the direct uplift of air from the tropopause above the equator seems not to be
the preferred tracers' path in shorter timescales. Such tropical uplift is clearly seen only in January-
February and October-November. On the other hand, this uplift is known to be slow and robust in a
statistical sense. This seems to suggest that the tropical pipe may not be an actual transport path but
instead a residual which emerges when fluctuations at shorter timescales cancel out.

### 3.2.2   Inter-hemispheric Differences

While we see corresponding features in the SH and NH, the NH atmospheric circulation is not
merely a mirrored SH circulation phase-shifted by six months. The main differences are:

1. The deep branch of the BDC in the local winter stratosphere is stronger in the SH than in the
   NH. This is consistent with stronger southern than northern polar winter subsidence which is
   associated with less perturbed polar vortices there (Butchart, 2014, Section 5.1).

2. While the major patterns can be found in both hemispheres, their typical altitude is different
in the NH compared to the SH. Overall, they appear at higher levels in the SH. This applies
   for instance to the low-midlatitudes feature in May-June and June-July in the SH, and to
   the November-December and December-January feature in the NH. Another example is the
   overturning circulation feature in March-April in the SH and September-October in the NH.
   To the best of our knowledge these altitude differences have not been reported before either.

3. The location of the regions with near-zero vertical effective velocities is different in both
   hemispheres.

4. There are differences in the structure of the overturning circulation: only one pathway is ob-
   served in the SH towards NH branch in September-October while two pathways are seen in

the NH towards SH branch in March-April. This creates a large region of isolated air in the

NH for these months, without a SH counterpart. To the best of our knowledge this also has not been reported before.

5. In the NH we detect a summer signal (June to September), that has no SH counterpart, which we attribute to the Asian monsoon.

### 3.2.3 Variable Phenomena

The variability of atmospheric transport is particularly large at winter polar latitudes. This applies both to the lower mesosphere/upper stratosphere region and the middle stratosphere and the connected midlatitudinal mesospheric transport pattern (Figs. 5–8). In the NH this variability is related to sudden stratospheric warmings. This variability causes an underrepresentation of the related transport pattern in the multiannual monthly mean. Conversely, in the SH, the interannual variability in

the polar vortex break-up is associated with very high variability in transport in October–November in the SH polar region between 20-35 km, which is further enhanced by an interaction between the QBO and vortex dynamics (e.g., Strahan et al., 2015).

Another region of large variability in meridional effective velocities is found over mid-low latitudes in the winter hemisphere (Figs. 5–8, January-February in the NH, June-July and July-August in

the SH), reaching maximum standard deviation values between 40-50 km of altitude. In the SH this high variability pattern persists also in August-September and September-October. This region coincides with that where planetary-wave breaking is observed, which was first discovered by McIntyre and Palmer (1984) and to which they named the stratospheric 'surf zone'.

### 4 Discussion and Conclusion

The ANCISTRUS method applied to MIPAS data broadly reproduces the known atmospheric meridional circulation patterns, but with some unexpected features. Additional information has been obtained from this new data set regarding how some of these patterns evolve over the year. Compared to established methods, it provides circulation fields at largely improved temporal and spatial resolution and at altitudes not accessible by the classical methods such as age-of-air analysis on the basis

of air sampling instruments or satellite based $SF_6$ measurements. The results are stable in the sense that the interannual variability of a pattern seen at a certain time of the year is small, that is to say, the patterns do not average out when the mean circulation is calculated although the input circulation patterns for the averaging process have been generated independently. This behaviour cannot be attributed to the use of any a priori velocities that would push the results towards the expected

circulation patterns. On the contrary, our a priori effective velocities are zero throughout, which guarantees that all structures seen in the results are produced by the measured trace gas contributions. The zero a priori field is also used as initial guess of the iterative inversion, but its only effect on the

results is a certain smoothing of the retrieved structures (see, von Clarmann and Grabowski, 2021). Another sign of the stability of our method is that the transitions between the circulation patterns of subsequent months are reasonably smooth, which is another indicator of the robustness of the results. Large interannual variability is mainly limited to situations where it can be explained by processes known to have large interannual variability in themselves, e.g. sudden stratospheric warmings. The QBO is another driver of stratospheric variability (see, e.g., Strahan et al., 2015), and Haenel et al. (2015) found that the contribution of the QBO to the explanation of age-of-air time series is largest in the polar stratosphere.

The main novel results seen in these multiannual monthly averages are that the upper branch of the BDC and the overturning pole-to-pole circulation are heavily intertwined phenomena; the latitude of stratospheric uplift in the middle and upper stratosphere is more variable than previously established; and the schematics of the BDC usually shown seem to be representative for certain months only and do not capture enough detail and interactions between the various circulation branches. For example, the schematic of Bönisch et al. (2011, their Figure 1) seems to represent rather spring/autumn months instead summer/winter months as indicated.

Obvious future steps for this work are the analysis of interannual variability of transport patterns. Further planned activities consist in the application of this method to data from other space missions, such as the Microwave Limb Sounder (MLS) on the Aura satellite (Waters et al., 2006) and the distinction of transport versus mixing. More ambituous researchers may even plan an ANCISTRUS model in other than geometric altitude coordinates or even a three-dimensional version of ANCISTRUS, which would be a very versatile tool to infer velocities from concentration distributions for various applications.

*Code and data availability:* The data used in this paper and the ANCISTRUS software package are available via the KITopen repository under doi:10.5445/IR/1000127803 (Grabowski et al., 2020a) and doi:10.5445/IR/1000127728 (Grabowski and von Clarmann, 2020), respectively.

*Author contributions.* TvC has written major parts of the paper and has provided the code to consider sink reactions of trace gases. UG has maintained the ANCISTRUS code and developed it further according to the actual needs. NG and SK have preprocessed the MIPAS mixing ratios. GPS and BMS have contributed to the interpretation of the results. All authors have discussed the results and contributed to their final presentation.

## Appendix A:  The interpretation of 'effective velocities'

The effective velocities presented in this study cannot be interpreted as zonal mean velocities. The reason is twofold.

First, due to possible correlations between velocities and mixing ratios, products of prime terms in the zonal mean of the Reynolds decomposition of the tendency formulation of the continuity equation do not cancel out.

Following Tung (1982)[2] and applying approximations suggested therein, von Clarmann and Grabowski (2016, Appendix A) rewrite the continuity equation as


$$\frac{\partial}{\partial t}\overline{\text{vmr}_g} = -\frac{v^*}{r}\frac{\partial}{\partial \phi}\overline{\text{vmr}_g} - w^*\frac{\partial}{\partial z}\overline{\text{vmr}_g} + \frac{1}{r^2}\frac{\partial}{\partial \phi}\left[K_\phi^*\frac{\partial}{\partial \phi}\overline{\text{vmr}_g}\right] + \frac{\partial}{\partial z}\left[K_z^*\frac{\partial}{\partial z}\overline{\text{vmr}_g}\right],$$ (A1)

where, contrary to the notation of the main text, velocities, mixing coefficients and state variables with a bar indicate zonal averages while quantities without a bar indicate longitudinally resolved quantities, and where

$$v^* = \overline{v} - \frac{1}{\overline{\rho}}\frac{\partial}{\partial t}\overline{\rho'\eta'} - \frac{1}{\overline{\rho}r}\frac{\partial\overline{\rho}}{\partial \phi}K_{\phi\phi} - \frac{\partial}{\partial z}K_{z\phi} - \frac{1}{\overline{\rho}}\frac{\partial\overline{\rho}}{\partial z}K_{z\phi},$$ (A2)


$$w^* = \overline{w} - \frac{1}{\overline{\rho}}\frac{\partial}{\partial t}\overline{\rho'\Phi'} - \frac{1}{\overline{\rho}}\frac{\partial\overline{\rho}}{\partial z}K_{zz} - \frac{1}{r}\frac{\partial}{\partial \phi}K_{\phi z} - \frac{1}{\overline{\rho}r}\frac{\partial\overline{\rho}}{\partial \phi}K_{\phi z},$$ (A3)

$$K_\phi^* = K_{\phi\phi},$$ (A4)

and

$$K_z^* = K_{zz},$$ (A5)


$$K_{\phi\phi} = \frac{1}{\overline{\rho}}\overline{(\rho v)'\eta'},$$ (A6)

$$K_{zz} = \frac{1}{\overline{\rho}}\overline{(\rho w)'\Phi'},$$ (A7)

$$K_{\phi z} = \frac{1}{\overline{\rho}}\overline{(\rho v)'\Phi'},$$ (A8)

and where $\eta'$, $\Phi'$ and $\sigma'$ are defined by


$$\left(\frac{\partial}{\partial t} + \frac{\overline{u}}{r\cos\phi}\frac{\partial}{\partial \lambda}\right)\eta' = v',$$ (A9)

$$\left(\frac{\partial}{\partial t} + \frac{\overline{u}}{r\cos\phi}\frac{\partial}{\partial \lambda}\right)\Phi' = w',$$ (A10)

and

$$\left(\frac{\partial}{\partial t} + \frac{\overline{u}}{r\cos\phi}\frac{\partial}{\partial \lambda}\right)\sigma' = S'.$$ (A11)

To generate effective quantities comparable to our results from 3D fields requires not only the calculation of the

zonal mean velocities but also the evaluation of the second to fifth terms of Equations (A2) and (A3).

The second reason why our results cannot be understood as zonal mean velocities is that in our inversion we constrain to zero the effective mixing terms $\frac{1}{r^2}\frac{\partial}{\partial \phi}\left[K_\phi^*\frac{\partial}{\partial \phi}\overline{vmr_g}\right]$ and $\frac{\partial}{\partial z}\left[K_z^*\frac{\partial}{\partial z}\overline{vmr_g}\right]$, i.e., those terms in the continuity equation (Eq. A10 in von Clarmann and Grabowski, 2016) which act upon second derivatives of state variables. Thus, their effect is aliased onto our effective velocities. From comparison of our effective

velocities with zonal mean velocities information on the relevance of eddy transport and eddy mixing can be gained.

*Acknowledgements.* We thank the reviewers for their thorough, detailed and insightful reviews.

---

[2]There exist other approaches than that of Tung (1982) to interpret 2D circulation, using different approximations. Depending on the approach chosen, the calculation of effective 2D velocities from 3D fields involves different terms.

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
