# Peer review of "The Middle Atmospheric Meridional Circulation for 2002-2012 derived from MIPAS observations"

_Atmospheric Chemistry and Physics, 2019_

## Referee Comment (RC1) · Anonymous Referee #1 · 23 Oct 2019

- - - - - - - - General Comments - - - - - - - -

von Clarmann et al. 2019 present results from an inverse method which uses observed (MIPAS) zonal-mean tracer fields to calculate residual circulation fields which are resolved in altitude, latitude, and time. This work expands on the work of von Clarmann and Grabowski 2016 (hereafter CG16) by providing time-resolved circulation fields, and continues the line of investigation of a number of other studies which have sought to constrain the strength of the residual circulation. However, the present work seeks to provide a substantial expansion in this direction by quantifying the circulation strength in terms of two-dimensional, time-resolved velocities. Only one previous work, to my knowledge, has quantified velocities at all - that being Fu, Hu, and Yang 2007 GRL - but this was only for a single profile of upwelling, while other work has provided some

sense of two-dimensional motion but without velocities, such as the work of Stiller et al. 2012.

The results show several inconsistencies with current theory. For example: The mesospheric overturning circulation is considerably higher (at least 10 km, which seems very unexpected) when southward-bound as opposed to northward-bound; the tropical pipe shows quite a bit of meridional movement rather than isolated upwelling. If reliable, such results would be of substantial and immediate interest for a large section of the middle atmospheric research community.

However, there are considerable issues with the validity of the results, and I do not think the work should be published until they are addressed. I outline them in the following three points:

1. The inverse model robustness (specifically in terms of sensitivity to input fields) has not been explored.

My impression from reading CG16 is that the inverse method requires multiple tracers but that the limit on the minimum number of tracers needed is rather soft (i.e. it is not strictly necessary to have X or more tracers). I suppose that it is possible to use a subset of the nine tracers applied, and thereby estimate the robustness of the method with respect to input data. In my opinion, having even a simple estimate is necessary. Having read this paper, I am left without an idea of how the results depend on the input tracer fields. Even if the method is mathematically sound, there may be biases in the results which extend from errors in tracer measurement or the calculation of chemical sources and sinks, and until this possibility is addressed the results remain in a somewhat skeptical position.

In particular, I would propose something like a jackknife test, calculating the fields with only eight tracers, excluding one tracer for each calculation, each tracer in turn, and seeing how strongly the velocity fields vary. It may not need to be a recalculation of the entire approx. decade-long climatology, but I think a test like this (or something more

advanced) in multiple seasons is necessary to establish the validity of the method.

2. The inverse model accuracy has not been quantified well.

In CG16, an accuracy test was performed where dynamical quantities were prescribed in simple distributions, which showed that the method had some inaccuracy in the center of the domain and much more inaccuracy on the borders. I think a test with dynamical and chemical fields that are more similar to observations (i.e. less homogenous and, simply-said, messy) is necessary to ascertain the uncertainty of the results. I would suggest using CCM model data to produced inversted circulation fields and compare them with the actual, directly-calculated model fields. To be clear, the results of such a test could not be used to pin a quantity directly onto the method's results (since, for example, the inverse method includes some effects of mixing, although knowing how the "effective" velocities could differ from the actual velocities would be valuable). However, this would provide some level of confidence in the results where, at the moment, the only indication extends from an example which simply does not resemble the observed atmosphere. As a final note on this, it would be best if the test could also examine the difference between CCM and inverted velocities, and not the residual between tracer distributions (as was done in CG16). The velocities are what matter, after all.

As an example of possible data, the ESCIMO (Joeckel et al. 2016 GMD) simulations have a variety of chemical species included (all the chemicals used in this study are in the model output) and calculations of the residual circulation strength have also been performed. I am not involved in that work, but I would guess that the members of that project would be interested in sharing the data.

3. The inverse model result uncertainty has not been quantified.

This is a rather small point, and not as important as the previous two. In CG16, quantifications of uncertainties of wind and diffusion coefficient fields were shown (in Figure 5) for fields computed using MIPAS data. Those uncertainties seem rather small,

Interactive
comment

but it seems imprudent to exclude an estimation of uncertainties when a method for estimating them is possible. I suggest including a description or depiction of those uncertainties, if the method still applies for the present work.

In my view, the first two points are necessary before these results should be published. Without a clear indication of the model validity, the novel results shown in this work seem as likely to be artifacts of the inverse method (or the calculation of chemical balances) as they are to correspond to reality.

Furthermore, I find that the figures in the manuscript are difficult to interpret, and should be changed before publication. I have made more specific comments on this topic below. In general, the figures provide a qualitative idea of the circulation, in that I can examine the figures and know where the circulation is the strongest and which way it is headed at any one time and know how that strongest location changes with time. That does provide a general characterization of the circulation, but knowing how the circulation strength changes with time in each location is of equal value to knowing where the criculation is strongest. The changing color scales in the first 12 panels makes that assessment difficult. The figures showing standard devations need to be explained clearly, because they are so unusual, but do not seem to be properly explained anywhere. I think the figures should be remade showing contours or heatmaps, or some combination of both methods, so that readers can assess the strength of the vertical and meridional velocities and the variability of them. Maybe including stream lines would maintain the ease in understanding the flow direction. I understand that this likely means a doubling of the number of figures (assuming there is not a clever way to combine the meridional and vertical information), but I think that it is warranted for the clarity of interpretation.

Furthermore, I have the following two points which I think warrant explanation, but are not addressed in the text:

Why do boundaries seem to be included in results, when they were problematic following CG16? Are they somehow excluded?

How were the sources and sinks of each species calculated?

To conclude, I would like to stress that results from this method should be published in the future if the reliability of the method can be addressed. They would be of very substantial interest, and therefore I wholeheartedly recommend resubmission.

- - - - - - - - - Specific Comments - - - - - - - - -

Line 14: Neither Brewer nor Dobson suggest upper and lower branches to the BDC, nor the mesospheric overtunring circulation. I think it would be best to cite somebody here who talks about that. Maybe Butchart's 2014 review paper would be good, to provide a general reference.

Line 20: The new sentence in this line seems to dismiss the value of having a single estimate of the upwelling mass flux / intensity. Perhaps it's just the use of the words "merely" and "far" (more/too). I'm not sure if anyone has ever suggested that a single quantity could sufficiently characterize the circulation, but certainly the upwelling mass flux has provided a lot of value as a broad estimate of the circulation. I suggest simplifying the sentence - "These studies suggest that the ... is too complicated and detailed to be fully characterized by a scalar intensity." - or something similar.

Line 56: I don't understand what is meant by point b. Stiller et al. also uses MIPAS data. What does the present method do differently? Since the chemistry of SF6 is not considered, any chemistry that is actually happening would create biases in those regions where it occurs (but I am no expert on this, and cannot say if those regions are included here). As a secondary point, there is no over-aging involved in this method because ages are not calculated. What would you expect in the case of your inversion method, if you had this bias? I would assume the result would be a faster lower-strat to upper-middle atmosphere circulation.

Line 78: Where did the data for OH, O1(D), and Cl come from? As I understand it,

those JPL publications only contain reaction rate and cross-section information. Did you obtain that data from somewhere else (does MIPAS have those species?) or did you model those in some other way?

Line 86: How necessary is the stabilization of the inversion, that mixing coefficients were assumed to be zero? Can you compute mixing coefficients for even a single pair of months, or perhaps for a pair of boreal summer and a pair of boreal winter months? Otherwise, it's difficult to say how the effective velocities compare to advective velocities.

Line 88: What does the word "efficient" in "efficient 2D circulation" mean? Do you mean effective? If not, the meaning should be clarified.

Line 89: I have never seen the term "Fickian mixing" before. Having done some searching, I think what you are referring to is more commonly called diffusion. If that's the case, I would use that term. Otherwise, the meaning should be explained.

Section 2.3: Why do you average every pair of months? I would guess that is due to interannual variability of the phase (and I use that word very loosely here, perhaps it would be better to say timing) of the circulation. Whatever the case is, it should be stated.

Figures 1,2,3, and 4: I find these figures very difficult to interpret. First, the changing color scale does not seem necessary, as the maximum values do not vary strongly between the figures - most of them are around 11/12 - although it would create an issue for the December-January figure. At the moment, however, it is difficult to assess how the magnitude of the circulation changes at each point, month-to-month, except in the most starkly contrasting cases (. Second, it is difficult (if not for most cases pratically impossible) to obtain even a rough magnitude of the vector components because the color scale shows a norm of the vectors. I suggest using contours or heatmaps of the separate vector components instead. Third - and this has nothing to do with interpretation - you show the boundary velocities of these results although it is clear from CG16

that the vectors at the boundaries are difficult for the inverse model.

Figures 5 and 6: These figures are unusual and require some considerable effort to comprehend. I assume the width versus height of the bubbles shows the stardard deviation in the meridional and vertical directions, and that the colors show the standard devation of the norm, but no information is given about that. I would suggest simply relacing these with contours and/or heat maps.

All figures: I suggest using a perceptually uniform colorscale, which makes viewing much easier for those who do not have a standard perception of color.

Line 133: I am not sure that I can agree that the vertical motion over this range creates a transport barrier. I would rather say it suggests one, at the most. But it could also be interpreted as a latitude (or a section of a latitude) where the circulation splits. In that case, there's not really a barrier to horizontal transport, but only relatively little forcing towards horizontal transport / a stronger forcing towards vertical transport. I'm not sure what the case truly is, but I don't think it's certain that this represents a barrier.

Line 138: "would likely look" - I think it's highly likely, even, but no definitive statement can be made until the analysis is done. On that topic, I think that analysis would be very interesting for future work.

Line 142: You might consider showing the tropopause and stratopause level sin your figures. I think that would be very helpful, and could alleviate a lot of confusion.

Line 150: It would be more useful to replace the values shown in Figures 5 and 6 with the values discussed in this sentence. That would be a more direct indication for the reader of where the circulation is consistent. Otherwise, they need to compare these values themselves, which is rather tedious if a thorough comparison is desired.

Line 189: Leaving aside the term "latitudinal barrier" (again, I am not sure how to distinguish between a barrier to latitudinal transport or a region of weak latitudinal transport), I do not see that agree that with the term "contribute to the formation".

[Figure]

Line 234: I've mentioned this already, but I think this case shows clearly why the variability should be depicted in another way. It's too difficult to compare the stanard deviations here to the circulation strength, for the most part. But your argument does seem plausible.

Line 237: It seems like you wanted to specify a figure in Ploeger et al. 2017. I suppose you mean Figure 5? You should specify the figure.

Line 323: What is independent about the ANCISTRUS results?

Line 323: "resulting fields are stable" – The statement regarding field stability should be more nuanced. Some parts of the fields do seem to be stable, sure, but this statement suggests that the fields are generally/always consistent.

Line 324: "increases confidence in the robustness of the analysis method" - I do not agree. If the method is robust, then a rather stable circulation field over the approx. decade of measurements in one region would suggest that the circulation field is a typical phenomenon for that region, at least for that decade. However, the robustness of the inverese method cannot be assessed by seeing consistency in its results without a second point of reference (preferably, other observation-based circulation estimates, which - to my knowledge - do not exist).

Line 330: It's not clear to me if a clear separation between these two pathways would be expected or not. Could you provide any context on that, in terms of earlier literature?

Line 333: "consistent with the assumption" - Has anyone previously suggested this idea, or are you saying that your results would only be consistent with a northward pole-to-pole circulation if it was above the domain of MIPAS? If nobody has suggested this, then this statement should be written differently to clarify the novelty of the result.

Line 337: To my understanding, this intrepretation of the tropical pipe would be novel. I only wish to note that here.

Line 341: This could be consistent with some earlier results. See Butchart 2014 (The

[Figure]

Brewer-Dobson Circulation, Rev. Geophys.) Figure 6 and discussion of that figure. If mean downwelling during winters where the polar vortex is not disturbed is the same between both hemispheres, then one would expect stronger climatological descent in the southern hemisphere because the vortex is disturbed less often there.

Line 343: To my knowledge, this result is not expected.

Line 347: I've mentioned it already, but I do not think the term "barriers" is justified in this context.

Line 348: Same to point 2.

Line 364: In the broad stroke, I agree with this statement. However, the absence of a southward mesospheric overturning circulation means that this statement cannot be written in the absolute. Furthermore, the results do not characterize these patterns in an expected fashion (tropical pipe, for example). This statement should be rewritten to reflect those points.

- - - - - - - - Technical Comments - - - - - - - -

Abstract, line 1: HCFC-22.

Line 14: The citation of Brewer and Dobson 1949 is incorrect. That's a single-author publication, just from Brewer.

Line 14: I think it's better to write abbreviations in a separate set of parentheses, just so it's clear that the abbreviation isn't some part of a citation.

Line 16: The last part of this sentence seems to suggest that only aerosols affect major chemistry-climate processes. It would be more clear with "as aerosols, all of which affect major chemistry".

Line 19: This sentence is a bit of a run-on. It would be better to write one sentence for Engel's balloon studies and another for the satellite studies.

[Figure]

Line 22: "Offline model simulations ... analysis data have also confirmed ...", or add a comma after "Also".

Line 28: It looks like you meant to write "has been investigated" or something similar. At the moment, the sentence doesn't make sense.

Line 30: If Funke et al. (from all those years) showed this, then I would write "has been" not "could be".

Line 35: It would be more precise to simply say that the picture of the middle atmospheric circulation is better resolved in space and time, cutting out the "more detailed" part.

Line 38: The ending, "over the years", isn't necessary here.

Line 46: I would say not just monthly but "monthly-mean". Furthermore, the sentence suggests that this is the only way to infer the circulation, so you should specify "is inferred in this work".

Line 49: "The resulting circulation fields..."

Line 62: I don't understand what "related software" refers to. That's the inversion method, right?

Line 79: "source reactions were also considered"

Line 83: "the neglect of sinks above that altitude"

Line 91: See comment on line 14 about abbrevations.

Line 103: "From MIPAS, measurements"

Line 104: You explained the data gaps in the last section.

Line 116: "up to 30 km"

Line 123: move "also" form "also the standard..." to "are also shown". Furthermore, it

should be clear to all readers that standard deviations are a measure of variability, so the "which are a measure of their variability" is not necessary.

Line 127: You can just say northern hemisphere or winter hemisphere.

Line 146: "has its maximum ... and at 30*S"

Line 155: You are clearly comparing this month-pair with the previous, but this sentence should make that explicit.

Line 167: "will give rise"

Line 170: The abbreviation SH was already used before this point. The notification of the abbrevation should be shifted to the first usage of "southern hemisphere". Ditto for NH.

Line 379: "their figure" - This part of the sentence addresses a particular figure, but the earlier part speaks generally of schematics. I suggest sticking to one approach or the other.

<hr />

---

## Referee Comment (RC2) · Anonymous Referee #2 · 5 Nov 2019

An observation-based climatology of middle atmospheric meridional circulation T. von Clarmann et al.

The authors present an estimate of meridional circulation patterns in the middle atmosphere based on measurements from 2002 though 2012 by the MIPAS instrument of a range of trace gas species. The estimate is based on an inverse method that infers an effective flow field in the meridional plane from the continuity equation along with an estimate of chemical sources and sinks. The methodology is updated from previous work by the first two authors through inclusion of further chemical sources and sinks, and by inferring only an 'effective' meridional flow that includes the effects of mixing/eddy transport.

The main results shown are the month by month estimates of the decadal-averaged

meridional flow, as well as an estimate of the interannual variability of the flow. In as much as this estimate is a relatively direct observational estimate of a difficult to measure quantity, this result is of potential value to the broader community. My main concern is that if this is to be the case, enough quantitative details should be given in order to facilitate comparisons with these results; this is largely the case but there are a few ambiguities that should be addressed (see below). Beyond this I have a few questions and comments about the presentation of these results (in particular I find the presentation of the interannual variability difficult to understand), but otherwise feel this is appropriate for publication with some minor revisions.

Specific comments

1) As mentioned above there are a few points that would be helpful for making quantitative comparisons with these results. Firstly, does the inferred circulation conserve mass? If so the authors may want to consider showing a mass stream function instead of the vector plots. If not, the choice of units for Figs. 1 through 4 are a bit confusing; surely the velocities should be homogeneous in units (e.g. m/s)? The note regarding the colour scales in the

2) Figures 5 and 6 show standard deviations of the inferred effective velocities, but the visualization is not explained. One assumes that the axes of the ellipses are scaled relative to the variance of the y and z components of the velocities but it seems no account is being taken of their covariance if that is the case. How are the colors chosen? More importantly, are these estimates of the standard deviation of the mean (implied by the figure caption) or sample standard deviations? In sum the interannual variability is difficult to assess in comparison to the mean circulation from these figures and is not very satisfyingly discussed in the text.

3) In all figures, different years are included in each panel with no discussion; why?

4) The methodology used in the present work includes the role of chemical sources and sinks; this has been updated from von Clarmann and Grabowski (2016). The

value of these updates should be demonstrated. It would also be useful to make some assessment of the role of mixing, again in order to facilitate quantitative comparisons with the mean meridional flow from models, for instance.
* * *

---

## Short Comment (SC1) · 1 Dec 2019

In this paper, the authors use measurements of a variety of trace gases from MIPAS to infer the stratospheric and mesospheric circulation. They calculate a climatology and determine that the deep branch of the Brewer Dobson circulation is connected to the mesospheric pole-to-pole circulation. They verify a number of known characteristics of the circulation, such as sudden stratospheric warmings increasing variability.

Using chemical tracers to infer the circulation is an excellent idea. Tracers are what we can measure from space, so to validate any model, we need to quantitatively relate the tracers to the dynamical output from climate models. The inverse methods used here are promising. Unfortunately, the approach from the authors is lacking in a number of

ways. 1) The validity of the method has not been established. 2) Uncertainties are not calculated and, perhaps most importantly, 3) The utility of the resulting product from the method is unclear.

The authors have not demonstrated that this inversion recovers velocity fields in a model. The closest is an idealized case in their 2016 paper and even in that case, idealized tracers are used instead of real tracers. In order to use the method on data, essentially an entire separate modeling study needs to be performed: a) using a CCM, with full knowledge of the tracer fields used here, the inversion needs to be performed and compared to the model velocity and stream function. If this is successful, then the next step is: b) with the same CCM, the sampling characteristics of MIPAS (coverage and averaging kernels) need to be applied to the tracer fields so that now the limitations due to sampling and retrieval characteristics are applied. This seems especially important for vertical and horizontal resolution. Then the inversion needs to be done and compared once again to the model velocity and stream function output. This test will illuminate what the method actually means.

The errors caused by the method with full tracer fields and then with the limited sampling can then be characterized for the model as well, hopefully beginning to address point 2) above. This would also work towards addressing point 3) above. This "transport circulation" that the authors obtain is not obviously relevant. Without being able to meaningfully compare values to model output or reanalysis products, this quantity does not seem to be of interest, and so the authors claims of being able to assess quantitatively the variability of the circulation fall flat. In fact, this is the reason that age of air is such a useful tracer—it has been quantitatively related to the circulation of the stratosphere in a way that allows direct comparison of data (including the MIPAS data) to models (Linz et al. 2017).

I would strongly encourage the authors to perform such a study and then to rethink their results for this work in the context of the information provided by their validation study. That would be an excellent paper that I would be truly excited to see.

Beyond this overall assessment, I have included a few additional comments below: 24: What about the lifting of the circulation? (e.g. Oberlander-Hayn 2016)

Introduction: What is the gap that this research is filling? The introduction reviews the literature but does not identify any motivation for the present study.

2.1 This discussion of age of air is surprising. What is the so-called "traditional observation-based characterization of the circulation"? The authors do not provide a citation. Some recent work that uses age of air observations to characterize the circulation is Linz et al. 2017. Recent work by Ray et al 2016 combines the TLP with chemistry to examine transport, and the improvement this offers over that method should be addressed. Ray et al. 2010 is also a relevant comparison here. Furthermore, it is not clear how the method can reveal causes of "discrepancies" between these age and chemical tracer based methods since there are no error estimates.

2.1.2 There is no discussion of degrees of freedom. How much independent information is gained by including additional tracers? Specifically, how are sinks included?

3.1: Plots are very hard to understand. Streamfunctions would be much better.

3.1.1 How are horizontal transport barriers identified? Why, physically, are they associated with this vertical motion? Is this purely a result of continuity and the fact that this is a 2-D calculation? If so, this should not be referred to as a barrier.

3.1.6 How precisely do you identify that this circulation is associated with the monsoon? Are there particular tracers (e.g. water vapor) that mark this as a monsoon signal? Or is it just about the timing and the fact that it's in the Northern Hemisphere? "Our results show overall agreement with the one shown by Ploeger et al. (2017)." What "one"? What is meant by "overall agreement"?

3.2.1 337: What is meant by this? How does this reconcile with the well-established water vapor tape recorder results?

4 368-374: This seems to be saying that this study is a good validation of the method.

That may be, but it's not the stated goal, and more stringent validation is needed especially so as to be able to actually interpret the resulting "effective velocities". 384: What applications would use these effective velocities?

Citations: Linz et al. 2017 10.1038/NGEO3013 Oberlander-Hayn et al. 2016 10.1002/2015GL067545 Linz et al. 2016 10.1175/JAS-D-16-0125.1 Ray et al. 2016 10.1002/2015JD024447 Ray et al. 2010 10.1029/2010JD014206.

Final note: I apologize to the authors for the tardiness of this review. I really wanted to like this paper, and I hope that they follow up with the modeling study to put these results in context.

---

## Author Comment (AC1) · 13 Jan 2020

The authors thank the reviewers for their thorough, detailed, and insightful reviews. We will use all these recommendations to improve the clarity of this paper. In the following we insert our replies directly into the review.

**Review #1:**

**Review: General Comments**
von Clarmann et al. 2019 present results from an inverse method which uses observed (MIPAS) zonal-mean tracer fields to calculate residual circulation fields which are resolved in altitude, latitude, and time. This work expands on the work of von Clarmann and Grabowski 2016 (hereafter CG16) by providing time-resolved circulation fields, and continues the line of investigation of a number of other studies which have sought to constrain the strength of the residual circulation. However, the present work seeks to provide a substantial expansion in this direction by quantifying the circulation strength in terms of two-dimensional, time-resolved velocities. Only one previous work, to my knowledge, has quantified velocities at all - that being Fu, Hu, and Yang 2007 GRL - but this was only for a single profile of upwelling, while other work has provided some sense of two-dimensional motion but without velocities, such as the work of Stiller et al. 2012.
The results show several inconsistencies with current theory. For example: The mesopheric overturning circulation is considerably higher (at least 10 km, which seems very unexpected) when southward-bound as opposed to northward-bound; the tropical pipe shows quite a bit of meridional movement rather than isolated upwelling. If reliable, such results would be of substantial and immediate interest for a large section of the middle atmospheric research community.

**Reply:** The authors thank the reviewer for the appreciation of our results.

**Planned Action:** None to this point.

**Review: However, there are considerable issues with the**

validity of the results, and I do not think the work should be published until they are addressed. I outline them in the following three points:

1. The inverse model robustness (specifically in terms of sensitivity to input fields) has not been explored. My impression from reading CG16 is that the inverse method requires multiple tracers but that the limit on the minimum number of tracers needed is rather soft (i.e. it is not strictly necessary to have X or more tracers). I suppose that it is possible to use a subset of the nine tracers applied, and thereby estimate the robustness of the method with respect to input data. In my opinion, having even a simple estimate is necessary. Having read this paper, I am left without an idea of how the results depend on the input tracer fields. Even if the method is mathematically sound, there may be biases in the results which extend from errors in tracer measurement or the calculation of chemical sources and sinks, and until this possibility is addressed the results remain in a somewhat skeptical position. In particular, I would propose something like a jackknife test, calculating the fields with only eight tracers, excluding one tracer for each calculation, each tracer in turn, and seeing how strongly the velocity fields vary. It may not need to be a recalculation of the entire approx. decade-long climatology, but I think a test like this (or something more advanced) in multiple seasons is necessary to establish the validity of the method.

**Reply:** We have meanwhile a validation paper of the ANCISTRUS (Analysis of the circulation of the stratosphere using spectroscopic measurements) method available, which is ready for submission. It includes an assessment of the relevance of sinks versus transport of patterns, the jackknife tests, model recovery tests, and an assessment of the adequacy of the regularization strength chosen. In a nutshell, the results are: below 30 km ANCISTRUS is quite accurate in a quantitative sense. Above, due to the regularization of the inversion along with less measurement information, peak velocities are underestimated. All structures and patterns, however, are nicely reproduced. Since none of the conclusions of the paper under review are fully quantitative but are related to structures and patterns, we are confident that the validity of our method for the purpose of our paper has now been sufficiently established, and it can safely be excluded that the patterns detected are mere artefacts.

**Planned Action:** The draft of the validation paper will be made available to the reviewers. In the climatology paper we will make reference to the results of the validation paper.

**Review: 2. The inverse model accuracy has not been quantified well. In CG16, an accuracy test was performed where dynamical quantities were prescribed in simple distributions, which showed that the method had some inaccuracy in the center of the domain and much more inaccuracy on the borders.**

**Reply:** The inaccuracies at the borders were caused by the fact that the test fields, which were chosen in an ad hoc manner, did not satisfy the continuity equation. A method that allows only solutions which do satisfy the continuity equation will never be able to reproduce such fields. That is to say, the problem was the test fields, not the method. More adequate tests are included in the validation paper mentioned above.

**Planned Action:** See above.

**Review: I think a test with dynamical and chemical fields that are more similar to observations (i.e. less homogenous and, simply-said, messy) is necessary to ascertain the uncertainty of the results. I would suggest using CCM model data to produced inversted circulation fields and compare them with the actual, directly-calculated model fields. To be clear, the results of such a test could not be used to pin a quantity directly onto the method's results (since, for example, the inverse method includes some effects of mixing, although knowing how the "effective" velocities could differ from the actual velocities would be valuable). However, this would provide some level of confidence in the results where, at the moment, the only indication extends**

from an example which simply does not resemble the observed atmosphere. As a final note on this, it would be best if the test could also examine the difference between CCM and inverted velocities, and not the residual between tracer distributions (as was done in CG16). The velocities are what matter, after all. As an example of possible data, the ESCIMO (Joeckel et al. 2016 GMD) simulations have a variety of chemical species included (all the chemicals used in this study are in the model output) and calculations of the residual circulation strength have also been performed. I am not involved in that work, but I would guess that the members of that project would be interested in sharing the data.

**Reply:** A validation with realistic test fields has been performed and included in the validation paper mentioned above. Comparisons to models are interesting in their own right, and this is on our agenda for the future. We think, however, that validation does not necessarily rely on model data.

**Planned Action:** See above.

**Review: 3.** The inverse model result uncertainty has not been quantified. This is a rather small point, and not as important as the previous two. In CG16, quantifications of uncertainties of wind and diffusion coefficient fields were shown (in Figure 5) for fields computed using MIPAS data. Those uncertainties seem rather small, but it seems imprudent to exclude an estimation of uncertainties when a method for estimating them is possible. I suggest including a description or depiction of those uncertainties, if the method still applies for the present work.

**Reply:** The method is still applicable and we have the data available, but we consider this as largely redundant in this context. We present the standard deviations of the monthly averages, and this quantity includes both the uncertainty of the inversion and the year-to-year atmospheric variability. Thus, the standard deviations characterize how well the climatology of a month can represent any particular month of the sample. The CG16 estimates of the random uncertainty represent the mapping of the measurement errors on the inferred velocity fields only, but not the representativeness problems due to natural variability. But since both types of uncertainties are independent from each other and add up quadratically, small standard deviations indicate that also the measurement-error-based uncertainties must be small. The standard deviations are even a more reliable estimate of the upper bound of the precision because they do not depend on any assumption on the measurement uncertainty.

**Planned Action:** We add to the manuscript: "(To diagnose this effect, also the standard deviations of the circulation vectors, which are a measure of their variability, are shown in Figs xy–xy.) This variability is caused by the natural variability of the circulation over the years and its random uncertainty. The latter is the random uncertainty of the MIPAS measurements propagated onto the circulation vectors.

**Review: In my view, the first two points are necessary before these results should be published. Without a clear indication of the model validity, the novel results shown in this work seem as likely to be artifacts of the inverse method (or the calculation of chemical balances) as they are to correspond to reality.**

**Reply:** The validation is reported in a separate paper, see above.

**Planned Action:** As stated above, the draft of the validation paper will be made available to the reviewers.

**Review: Furthermore, I find that the figures in the manuscript are difficult to interpret, and should be changed before publication. I have made more specific comments on this topic below. In general, the figures provide a qualitative idea of the circulation, in that I can examine the figures and know where the circulation is the strongest and which way it is headed at any one time and know how that strongest location changes with time. That does provide a general characterization of the circulation, but knowing how the**

**circulation strength changes with time in each location is of equal value to knowing where the circulation is strongest. The changing color scales in the first 12 panels makes that assessment difficult.**

**Reply:** While we think that a qualitative empirical representation at this detail level is still unprecedented, we do agree that the figures should be improved.

**Planned Action:** Circulation patterns will be represented on a common color scale. The original data will be made publically available, allowing each user to plot them in their preferred representation.

**Review: The figures showing standard devations need to be explained clearly, because they are so unusual, but do not seem to be properly explained anywhere. I think the figures should be remade showing contours or heatmaps, or some combination of both methods, so that readers can assess the strength of the vertical and meridional velocities and the variability of them. Maybe including stream lines would maintain the ease in understanding the flow direction. I understand that this likely means a doubling of the number of figures (assuming there is not a clever way to combine the meridional and vertical information), but I think that it is warranted for the clarity of interpretation.**

**Reply:** We have decided to follow the suggestion to represent the variabilities in a different way.

**Planned Action:** Separate plots for variabilities in $v$ and $w$ will be provided.

**Review: Furthermore, I have the following two points which I think warrant explanation, but are not addressed in the text: Why do boundaries seem to be included in results, when they were problematic following CG16? Are they somehow excluded?**

**Reply:** No, boundaries are not excluded. These problems in CG16 were not a problem of our transport and inversion schemes but they were due to the simplified velocity fields used there which were not consistent with the continuity equation. E.g., if I have a poleward flow but no backward flow it does not come as a surprise that funny things happen near the pole. In CG16 such simple velocity fields were used because they allowed much simpler tests whether the transport scheme does what it is supposed to do, e.g. if a feature is transported with the right velocity etc. Applied to real data this type of problem does not exist because the true concentration fields are well described by the continuity equation, and the Earth's air is not forced to accumulate at the pole because of unrealistic velocity fields, or to do other funny things.

**Planned Action:** None specific to this point. We will, however, summarize the results of the validation paper and discuss which kind of conclusions are supported by the data.

**Review: How were the sources and sinks of each species calculated?**

**Reply:** The general description of sources and sinks was already described in Section 2.1.2 of the original paper. Some missing information will be included, particularly the assumptions on the abundancies of involved species.

**Planned Action:** See comment on line 78.

**Review: To conclude, I would like to stress that results from this method should be published in the future if the reliability of the method can be addressed. They would be of very substantial interest, and therefore I wholeheartedly recommend resubmission.**

**Reply:** A validation paper is ready for submission. It confirms that the conclusions in the paper under review are robust.

**Planned Action:** A draft validation paper is now available, see above.

**Review: Specific Comments**
Line 14: Neither Brewer nor Dobson suggest upper and lower branches to the BDC, nor the mesospheric overturning circulation. I think it would be best to cite somebody here who talks about that. Maybe Butchart's 2014 review paper would be good, to provide a general reference.

**Reply:** Agreed.

**Planned Action:** The related paragraph will be rewritten and the Butchart (Rev. Geophys. 52(2), doi10.1002/2013RG000448, 157-184, 2014) reference will be included.

**Review: Line 20:** The new sentence in this line seems to dismiss the value of having a single estimate of the upwelling mass flux / intensity. Perhaps it's just the use of the words "merely" and "far" (more/too). Im not sure if anyone has ever suggested that a single quantity could sufficiently characterize the circulation, but certainly the upwelling mass flux has provided a lot of value as a broad estimate of the circulation. I suggest simplifying the sentence - "These studies suggest that the ... is too complicated and detailed to be fully characterized by a scalar intensity." - or something similar.

**Reply:** agreed to remove the words which make the sentence sound dismissive.

**Planned Action:** These words will be deleted.

**Review: Line 56:** I don't understand what is meant by point b. Stiller et al. also uses MIPAS data. What does the present method do differently? Since the chemistry of $SF_6$ is not considered, any chemistry that is actually happening would create biases in those regions where it occurs (but I am no expert on this, and cannot say if those regions are included here). As a secondary point, there is no over-aging involved in this method because ages are not

**calculated. What would you expect in the case of your inversion method, if you had this bias? I would assume the result would be a faster lower-strat to upper-middle atmosphere circulation.**

**Reply:** The main $SF_6$ depletion happens in the mesosphere. Thus in the following we refer only to air parcels travelling through the mesosphere and back to the stratosphere. The method by Stiller et al., as well as earlier studies using this approach, are sensitive to the destruction of $SF_6$ along the entire trajectory of an air parcel from the stratospheric entry point, through the stratosphere and mesosphere and back into the stratosphere, because the age is calculated by comparison of actual $SF_6$ stratospheric mixing ratios and past $SF_6$ mixing ratios at the entry-point. Our method is different in that we use $SF_6$ mixing ratios measured at the upper boundary as a reference for gradient calculations in the uppermost model layer. Thus, any calculation of differences used for the calculation of the gradients (needed to solve the transport equation) is based on $SF_6$ mixing ratios which are, if coming from the mesosphere, already depleted in $SF_6$. Thus, mesospheric $SF_6$ destruction cannot lead to artefacts in the gradients. The only $SF_6$ loss that we possibly miss is $SF_6$ destruction within the **stratosphere** from one month to the other, which is a minor inaccuracy compared to the problem of mesospheric $SF_6$ depletion in age-of-air applications. In short: The reference $SF_6$ used by Stiller is the (past) tropospheric $SF_6$ while our reference $SF_6$ is the depleted $SF_6$ in airmasses intruding from the mesosphere.

**Planned Action:** A clarifying sentence will be added.

**Review: Line 78: Where did the data for OH, $O^1(D)$, and Cl come from? As I understand it, those JPL publications only contain reaction rate and cross-section information. Did you obtain that data from somewhere else (does MIPAS have those species?) or did you model those in some other way?**

**Reply:** We estimate OH using the parametrization by Minschwaner et al. (Atmos. Chem. Phys. 11(3), 955-962, doi10.5194/acp-11-

955-2011, 2011). $O^1(D)$ is estimated using the equilibrium equation 5.38 in Brasseur & Solomon (2005, Springer), applied to MIPAS ozone; Cl is estimated by interpolating a climatological noon profile (Brasseur & Solomon, 2005, Fig 5.50) to the actual atmospheric state. Inaccuracies in the latter are considered to be less important because the atomic Cl sink is of much less relevance than the other sinks.

**Planned Action:** This missing information will be provided in (old) Section 2.1.2.

**Review: Line 86: How necessary is the stabilization of the inversion, that mixing coefficients were assumed to be zero? Can you compute mixing coefficients for even a single pair of months, or perhaps for a pair of boreal summer and a pair of boreal winter months? Otherwise, it's difficult to say how the effective velocities compare to advective velocities.**

**Reply:** We think that the effective velocities represent the essence of the Brewer-Dobson circulation in the sense that they conflate all the effects (advection, correlation effects, mixing) that bring, in a 2D world, a trace gas from here to there. From comparison with zonal mean advective velocities we can learn about the relative contribution of the non-advective terms.
Technically speaking, all monthly results are inferred independently. That is to say, the instability does not come from accumulation of errors over the months but is inherent in the analysis of each single month. The cause of the instability is the following: The system of equations solved tends towards linear dependencies as soon as velocities and mixing coefficients are to be retrieved simultaneously. The matrix to be inverted has an extremely high condition number. This can, in principle, be remedied by regularization. We found out, however, that in this case (contrary to the velocity-only analysis) the result depends strongly on the chosen regularization and is, thus, not robust. As a consequence, we have decided to constrain the mixing coefficients to zero and to re-interpret the resulting velocities as those 2D-velocities which best describe the combined effect of advection, eddy transport and eddy mixing. It cannot be expected that these effective velocities equal the zonal mean advective

velocities.

**Planned Action:** A clarifying sentence will be added.

**Review: Line 88: What does the word "efficient" in "efficient 2D circulation" mean? Do you mean effective? If not, the meaning should be clarified.**

**Reply:** This is a wording error and should read 'effective' instead. Thanks for spotting!

**Planned Action:** This will be corrected.

**Review: Line 89: I have never seen the term "Fickian mixing" before. Having done some searching, I think what you are referring to is more commonly called diffusion. If that's the case, I would use that term. Otherwise, the meaning should be explained.**

**Reply:** We intentionally avoid the term 'diffusion' in this context for the following reason: 'Diffusion' we understand is a physical process happening on a molecular scale. The processes we describe still abide to Fick's law of diffusion but are macroscopic processes. Thus we consider the term 'diffusion' in this context as misleading.

**Planned Action:** We will add a footnote explaining the meaning of 'Fickian mixing'.

**Review: Section 2.3: Why do you average every pair of months? I would guess that is due to interannual variability of the phase (and I use that word very loosely here, perhaps it would be better to say timing) of the circulation. Whatever the case is, it should be stated.**

**Reply:** There seems to be a fundamental misunderstanding. We do not average over two months. The velocity field labelled, say, March-April, is the velocity field that best reproduces the change of the monthly mean March mixing ratio field to the monthly mean April mixing ratio field.

**Planned Action:** We will add some clarification to avoid this misunderstanding. We think, however, that this clarification fits much better in the Section 'The General Approach' than here.

**Review: Figures 1,2,3, and 4: I find these figures very difficult to interpret. First, the changing color scale does not seem necessary, as the maximum values do not vary strongly between the figures - most of them are around 11/12 - although it would create an issue for the December-January figure. At the moment, however, it is difficult to assess how the magnitude of the circulation changes at each point, month-to-month, except in the most starkly contrasting cases.**

**Reply:** We do agree that the changing colour scales are disadvantageous.

**Planned Action:** New plots with fixed colour scales will be provided.

**Review: Second, it is difficult (if not for most cases pratically impossible) to obtain even a rough magnitude of the vector components because the color scale shows a norm of the vectors. I suggest using contours or heatmaps of the separate vector components instead.**

**Reply:** The length and direction of the arrows represent the velocity components. The colour scale was meant just as an additional guide of the eye. The length units of the arrows are ad hoc, that is to say, they are not consistent with the $\phi$ and $z$ axes intervals of the plots.

**Planned Action:** The original data will be made available.

**Review: Third - and this has nothing to do with interpretation - you show the boundary velocities of these results although it is clear from CG16 that the vectors at the boundaries are difficult for the inverse model.**

**Reply:** As already stated above, within the reply to the general comments, the problem at the boundaries is not a problem of the inversion scheme but a problem of inconsistent test data which represent non-realistic circulations where air accumulates at the boundary of the domain. By the way, the arrows in the boundary tiles refer to transport in the 80-90 degrees latitude band (and not to the 90 deg. latitude; i,e., we have no northward transport at the North pole.)

**Planned Action:** None.

**Review: Figures 5 and 6: These figures are unusual and require some considerable effort to comprehend. I assume the width versus height of the bubbles shows the stardard deviation in the meridional and vertical directions, and that the colors show the standard devation of the norm, but no information is given about that. I would suggest simply replacing these with contours and/or heat maps.**

**Reply:** Agreed.

**Planned Action:** We will show variabilities in $v$ and $w$ in separate figures.

**Review: All figures: I suggest using a perceptually uniform colorscale, which makes viewing much easier for those who do not have a standard perception of color.**

**Reply:** We have tried many different colour scales but the alternatives did not seem convincing to us.

**Planned Action:** We will present the same plots in other color scales in the supplement. Beyond this, we will make the original data available, thus the readers can plot then in their preferred representation.

**Review: Line 133: I am not sure that I can agree that the vertical motion over this range creates a transport barrier.**

I would rather say it suggests one, at the most. But it could also be interpreted as a latitude (or a section of a latitude) where the circulation splits. In that case, there's not really a barrier to horizontal transport, but only relatively little forcing towards horizontal transport / a stronger forcing towards vertical transport. I'm not sure what the case truly is, but I dont think it's certain that this represents a barrier.

**Reply:** We will rephrase this sentence to avoid misunderstanding.

**Planned Action:** We will rephrase lines 133-134 to: "The direct vertical motion over 30°S suggests the existence of a region where horizontal transport is minimal compared to vertical transport; the location of this region is in good agreement with the location of the subtropical transport barrier (e.g. Stiller et al., 2017)."

**Review: Line 138: "would likely look" - I think it's highly likely, even, but no definitive statement can be made until the analysis is done. On that topic, I think that analysis would be very interesting for future work.**

**Reply:** Agreed.

**Planned Action:** We will reword this: "Representation in equivalent latitudes would be more adequate to analyze this phenomenon but since that representation would not be optimal for global analyses, it is deferred to a future study."

**Review: Line 142: You might consider showing the tropopause and stratopause levels in your figures. I think that would be very helpful, and could alleviate a lot of confusion.**

**Reply:** Monthly averaged tropopause altitudes can be very misleading.

**Planned Action:** none

**Review: Line 150: It would be more useful to replace the**

values shown in Figures 5 and 6 with the values discussed in this sentence. That would be a more direct indication for the reader of where the circulation is consistent. Otherwise, they need to compare these values themselves, which is rather tedious if a thorough comparison is desired.

**Reply:** If we understand correctly, the reviewer recommends to present the ratio between variability and absolute velocity instead of the variabilities themselves. (in the sense "inferred velocities exceed their variabilities by a factor of ..."). We tried this but due to the often small velocities, this representation is not easy to interpret either. The plot would be dominated by large but meaningless ratios related to tiny velocities while regions of interest with large absolute variabilities would no longer be obvious. That is why we didn't choose this representation.

**Planned Action:** none

**Review: Line 189:** Leaving aside the term "latitudinal barrier" (again, I am not sure how to distinguish between a barrier to latitudinal transport or a region of weak latitudinal transport), I do not see that agree that with the term "contribute to the formation".

**Reply:** We will rephrase this sentence.

**Planned Action:** Lines 189-190 will be rewritten as "This feature will evolve in the following months as a region where uplift motion clearly overtakes horizontal transport around 60°N."

**Review: Line 234:** I've mentioned this already, but I think this case shows clearly why the variability should be depicted in another way. It's too difficult to compare the standard deviations here to the circulation strength, for the most part. But your argument does seem plausible.

**Reply:** We agree that this is difficult.

**Planned Action:** We will quote the numbers in the text and and

rephrase to make it clearer that the figure illustrates our argument rather than quantify it.

**Review: Line 237: It seems like you wanted to specify a figure in Ploeger et al. 2017. I suppose you mean Figure 5? You should specify the figure.**

**Reply:** Agreed. Indeed we mean Figure 5.

**Planned Action:** The Figure will be specified.

**Review: Line 323: What is independent about the AN-CISTRUS results?**

**Reply:** ANCISTRUS results are independent from each other in the sense that the result of an ANCISTRUS run for one month is never used a first guess, a priori, or similar of an ACISTRUS run of another month. All ANCISTRUS runs can thus be performed independently, and any artificial autocorrelation of the results is thus excluded.

**Planned Action:** We will slightly reword this: " from the results of independent ANCISTRUS runs".

**Review: Line 323: "resulting fields are stable" The statement regarding field stability should be more nuanced. Some parts of the fields do seem to be stable, sure, but this statement suggests that the fields are generally/always consistent.**

**Reply:** Agreed.

**Planned Action:** This statement will be made more specific.

**Review: Line 324: "increases confidence in the robustness of the analysis method" - I do not agree. If the method is robust, then a rather stable circulation field over the approx. decade of measurements in one region would suggest that the circulation field is a typical phenomenon for that**

region, at least for that decade. However, the robustness of the inverse method cannot be assessed by seeing consistency in its results without a second point of reference (preferably, other observation-based circulation estimates, which - to my knowledge - do not exist).

**Reply:** Well, this depends what how the term 'robustness' is understood. We understand 'robustness' as the characteristic that the solution is not overly sensitive to varying input. We do not claim here to have shown that the method is accurate. That is also why our initial wording was "increases confidence in the robustness" instead of "shows the results are robust".

**Planned Action:** We will clarify what we mean: " The stability of results from independent ANCISTRUS runs increases confidence in the robustness of the analysis method in the sense that it produces similar results for similar input fields."

**Review: Line 330: It's not clear to me if a clear separation between these two pathways would be expected or not. Could you provide any context on that, in terms of earlier literature?**

**Reply:** Agreed

**Planned Action:** we insert: "..., as suggested by the schematic shown in Fig. 1 of Boenisch et al. (2011)..."; Reference: Atmos. Chem. Phys. 11(8), 3937-3948, doi10.5194/acp-11-3937-2011, 2011),

**Review: Line 333: "consistent with the assumption" - Has anyone previously suggested this idea, or are you saying that your results would only be consistent with a northward pole-to-pole circulation if it was above the domain of MIPAS? If nobody has suggested this, then this statement should be written differently to clarify the novelty of the result.**

**Reply:** Well, we think that the existence of the pole-to-pole overturning circulation is well established. We see velocities going up in the north and downward velocities in the south, but the meridional velocities which would close this circulation are above our data domain. We neither claim to have found a novel circulation path nor do we refer to a specific assumption written in the literature. Thus our careful wording.

**Planned Action:** We will rewrite this without the term 'assumption': "Our data are consistent with - but do not directly support - a southward pole to pole circulation from March to May at altitudes not covered by MIPAS data"

**Review: Line 337: To my understanding, this intrepretation of the tropical pipe would be novel. I only wish to note that here.**

**Reply:** Yes, indeed. We have intentionally chosen a very careful wording here ('suggests'...'may not be'...). We are willing to make the wording even more careful.

**Planned Action:** We will change the wording to "This seems to suggest that...not always...

**Review: Line 341: This could be consistent with some earlier results. See Butchart 2014 (The Brewer-Dobson Circulation, Rev. Geophys.) Figure 6 and discussion of that figure. If mean downwelling during winters where the polar vortex is not disturbed is the same between both hemispheres, then one would expect stronger climatological descent in the southern hemisphere because the vortex is disturbed less often there.**

**Reply:** We understand our (old) lines 340-341 as an introductory sentence for the five following more specific points. Thus the quite specific comment seems to refer to line 342 (#1 in the list) rather than to line 341. We will add a sentence there.

**Planned Action:** We shall add to #1: "This is consistent with stronger southern than northern polar winter subsidence which is

associated with less perturbed polar vortices there (Butchart 2014, Section 5.1)." Reference: Rev. Geophys. 52(2), doi10.1002/2013RG000448, 157-184 (2014)

**Review: Line 343: To my knowledge, this result is not expected.**

**Reply:** Ok, we will mention this.

**Planned Action:** We shall add: "To the best of our knowledge this has not been reported before either."

**Review: Line 347: I've mentioned it already, but I do not think the term "barriers" is justified in this context.**

**Reply:** Agreed

**Planned Action:** We will replace the term 'barriers' by 'areas with near zero ... velocities.'

**Review: Line 348: Same to point 2.**

**Reply:** ok, we will mention this.

**Planned Action:** We shall add: "To our best knowledge this also this has not been reported before."

**Review: Line 364: In the broad stroke, I agree with this statement. However, the absence of a southward mesospheric overturning circulation means that this statement cannot be written in the absolute. Furthermore, the results do not characterize these patterns in an expected fashion (tropical pipe, for example). This statement should be rewritten to reflect those points.**

**Reply:** The overturning circulation is not absent but just not covered by the MIPAS measurements. But we agree to reword our statement.

**Planned Action:** We will write: "The ANCISTRUS method applied to MIPAS data broadly reproduces well the known atmospheric meridional circulation patterns, although with some unexpected features. Additional information ..."

**Review: Technical Comments**

**Review: Abstract, line 1: HCFC-22.**

**Reply:** Thanks for spotting

**Planned Action:** This will be corrected.

**Review: Line 14: The citation of Brewer and Dobson 1949 is incorrect. That's a single-author publication, just from Brewer.**

**Reply:** Thanks for spotting

**Planned Action:** This will be corrected.

**Review: Line 14: I think it's better to write abbreviations in a separate set of parentheses, just so it's clear that the abbreviation isn't some part of a citation.**

**Reply:** This comment has become obsolete after rewriting in reply to a specific comment (see above).

**Planned Action:** No additional action.

**Review: Line 16: The last part of this sentence seems to suggest that only aerosols affect major chemistry-climate processes. It would be more clear with "as aerosols, all of which affect major chemistry".**

**Reply:** Agreed.

**Planned Action:** This will be corrected

**Review: Line 19: This sentence is a bit of a run-on. It would be better to write one sentence for Engel's balloon studies and another for the satellite studies.**

**Reply:** We agree that the original sentence is too long. We prefer, however, to split the sentence immediately after the Butchart reference.

**Planned Action:** Will be changed to "...Butchart et al. (2006). This triggered..."; Reference: Clim. Dyn. 27(7-8), 727-741, doi10.1007/s00382-006-0162-4 (2006).

**Review: Line 22: "Offline model simulations ... analysis data have also confirmed ...", or add a comma after "Also".**

**Reply:** Thanks!

**Planned Action:** A comma will be inserted.

**Review: Line 28: It looks like you meant to write "has been investigated" or something similar. At the moment, the sentence doesn't make sense.**

**Reply:** Agreed.

**Planned Action:** This will be corrected.

**Review: Line 30: If Funke et al. (from all those years) showed this, then I would write "has been" not "could be".**

**Reply:** Agreed.

**Planned Action:** This will be corrected.

**Review: Line 35: It would be more precise to simply say that the picture of the middle atmospheric circulation is better resolved in space and time, cutting out the "more detailed" part.**

**Reply:** Agreed.

**Planned Action:** This will be corrected.

**Review: Line 38: The ending, "over the years", isn't necessary here.**

**Reply:** We had added 'over the years' to make clear that we do not talk about the inter-annual, not the intra-annual (month to month) variability. We will try to reword this in a clearer clumsy way.

**Planned Action:** 'over the years' will be deleted, and 'inter-annual' will be inserted before 'variability'.

**Review: Line 46: I would say not just monthly but "monthly-mean". Furthermore, the sentence suggests that this is the only way to infer the circulation, so you should specify "is inferred in this work".**

**Reply:** Agreed for "in this work". The original text already reads "monthly zonal mean" and we think that it is clear that averaging was made in both domains.

**Planned Action:** "in this work" will be added.

**Review: Line 49: "The resulting circulation fields..."**

**Reply:** Thanks!

**Planned Action:** This will be corrected.

**Review: Line 62: I don't understand what "related software" refers to. That's the inversion method, right?**

**Reply:** Yes, it is.

**Planned Action:** This will be reworded as suggested.

**Review: Line 79: "source reactions were also considered"**

**Reply:** Thanks for spotting!

**Planned Action:** This will be corrected.

**Review: Line 83: "the neglect of sinks above that altitude"**

**Reply:** Thanks!

**Planned Action:** This will be corrected.

**Review: Line 91: See comment on line 14 about abbreviations.**

**Reply:** As far as we know, our way to set the parentheses here is the one which complies with the COPERNICUS rules. I think here the copy editors have the last word.

**Planned Action:** We'll wait what the copy editors say.

**Review: Line 103: "From MIPAS, measurements"**

**Reply:** MIPAS here is a sort of attribute or specifications. With the comma inserted, the meaning would change towards "measurements are calculated from MIPAS", which is not what we mean.

**Planned Action:** None

**Review: Line 104: You explained the data gaps in the last section.**

**Reply:** Yes, indeed.

**Planned Action:** we will include the 2006 data gap in line 95 and reword here: "... with data gaps as reported above ."

**Review: Line 116: "up to 30 km"**

**Reply:** Agreed.

**Planned Action:** This will be corrected.

**Review: Line 123: move "also" from "also the standard..."
to "are also shown"(...). Furthermore, it should be clear to
all readers that standard deviations are a measure of vari-
ability, so the "which are a measure of their variability" is
not necessary.**

**Reply:** Thanks!

**Planned Action:** This will be corrected.

**Review: (...). Furthermore, it should be clear to all readers
that standard deviations are a measure of variability, so the
"which are a measure of their variability" is not necessary.**

**Reply:** A standard deviation is a measure of the width of a dis-
tribution but it is not clear if it represents variability, uncertainty,
probability, or other. In particular in our community, uncertainties
and estimated errors are often reported in terms of standard devia-
tion. Thus, we find it necessary to be specific here.

**Planned Action:** None.

**Review: Line 127: You can just say northern hemisphere
or winter hemisphere.**

**Reply:** Agreed.

**Planned Action:** "local winter" will be deleted.

**Review: Line 146: "has its maximum ... and at 30°S"**

**Reply:** Agreed.

**Planned Action:** This will be reworded as suggested.

**Review: Line 155: You are clearly comparing this month-**

**pair with the previous, but this sentence should make that explicit.**

**Reply:** Agreed.

**Planned Action:** "seen in January-February" will be inserted.

**Review: Line 167: "will give rise"**

**Reply:** Thanks!

**Planned Action:** This will be corrected.

**Review: Line 170: The abbreviation SH was already used before this point. The notification of the abbrevation should be shifted to the first usage of "southern hemisphere". Ditto for NH.**

**Reply:** Thanks for spotting.

**Planned Action:** This will be corrected.

**Review: Line 379: "their figure" - This part of the sentence addresses a particular figure, but the earlier part speaks generally of schematics. I suggest sticking to one approach or the other.**

**Reply:** Agreed.

**Planned Action:** We will split the sentence to make clear what refers to such schematics in general and what refers to this particular figure.

**Review #2:**

**Review:** The authors present an estimate of meridional circulation patterns in the middle atmosphere based on measurements from 2002 though 2012 by the MIPAS instrument of a range of trace gas species. The estimate is based on an inverse method that infers an effective flow field in the meridional plane from the continuity equation along with an estimate of chemical sources and sinks. The methodology is updated from previous work by the first two authors through inclusion of further chemical sources and sinks, and by inferring only an 'effective' meridional flow that includes the effects of mixing/eddy transport.
The main results shown are the month by month estimates of the decadal-averaged meridional flow, as well as an estimate of the interannual variability of the flow. In as much as this estimate is a relatively direct observational estimate of a difficult to measure quantity, this result is of potential value to the broader community.

**Reply:** We thank the reviewer for this encouraging evaluation.

**Planned Action:** None to this point.

**Review:** My main concern is that if this is to be the case, enough quantitative details should be given in order to facilitate comparisons with these results; this is largely the case but there are a few ambiguities that should be addressed (see below).

**Reply:** Agreed.

**Planned Action:** See specific actions below.

**Review:** Beyond this I have a few questions and comments about the presentation of these results (in particular I find the presentation of the interannual variability difficult to understand), but otherwise feel this is appropriate for publication with some minor revisions.

**Reply:** We agree that the presentation of the inter-annual variability should be better explained.

**Planned Action:** See specific actions below.

**Review: Specific comments**
**1) As mentioned above there are a few points that would be helpful for making quantitative comparisons with these results. Firstly, does the inferred circulation conserve mass?**

**Reply:** Yes, it largely does, except for transport into and out of the model domain at the upper and lower boundary, which may not necessarily be balanced. But besides the continuity equation of species, also the continuity equation of air density is one of the determining equations of our system.

**Planned Action:** None.

**Review: If so the authors may want to consider showing a mass stream function instead of the vector plots.**

**Reply:** Any representation which involves weighting by air density suffers from the large dynamical range of air density with altitude. We have tried various representations of this type but always all structure and information in the middle atmosphere was lost. Only values at the lowermost layer were discernable.

**Planned Action:** none

**Review: If not, the choice of units for Figs. 1 through 4 are a bit confusing; surely the velocities should be homogeneous in units (e.g. m/s)?**

**Reply:** With homogeneous units the vertical velocities would be invisible, although important. The norm we use for the colour scale and for the direction of the arrows roughly corresponds to the aspect ratio of the plots where the vertical axis does not represent the true geometric conditions either but is heavily exaggerated.

**Planned Action:** The original data will be made available in digital form. With this the user can represent the data in the preferred way, most suitable for the respective application.

**Review: The note regarding the colour scales in the 2) Figures 5 and 6 show standard deviations of the inferred effective velocities, but the visualization is not explained.**

**Reply:** The explanation is indeed missing in the original submission.

**Planned Action:** An explanation of the visualization will be included.

**Review: One assumes that the axes of the ellipses are scaled relative to the variance of the y and z components of the velocities but it seems no account is being taken of their covariance if that is the case. How are the colors chosen?**

**Reply:** The axes of the ellipses represent the standard deviations in the the y and z components. The covariances are not represented. The colours are chosen by adjusting the colour scale to the maximum value of the individual plot.

**Planned Action:** We have decided to represent the variabilities in separate plots for $v$ and $w$. We will use a common colour scale for the entire series of plots. Further we will mention that the same norm is applied to the standard deviations as for the effective velocities.

**Review: More importantly, are these estimates of the standard deviation of the mean (implied by the figure caption) or sample standard deviations?**

**Reply:** The title was indeed misleading. We present the sample standard deviations, not the standard error of the mean. This is because we are interested in the variability, not in the uncertainty. The standard error of the mean would become lower for a larger

sample and is thus not the adequate measure.

**Planned Action:** Titles in the plots will be corrected.

**Review: In sum the interannual variability is difficult to assess in comparison to the mean circulation from these figures and is not very satisfyingly discussed in the text.**

**Reply:** Agreed

**Planned Action:** New plots will be provided and adequately described.

**Review 3) In all figures, different years are included in each panel with no discussion; why?**

**Reply:** First we have some data gaps in the MIPAS data, and second, a (small) number of the inversions did not converge.

**Planned Action:** This information will be provided.

**Review: 4) The methodology used in the present work includes the role of chemical sources and sinks; this has been updated from von Clarmann and Grabowski (2016). The value of these updates should be demonstrated.**

**Reply:** The impact is indeed substantial, and we are happy to show respective plots. However, we think that this fits in much better with the paper on validation and sensitivity studies discussed in reply to reviewer #1.

**Planned Action:** This issue will be deferred to the validation paper of which the draft will be made available to the reviewers and reference to that paper together with a brief explanation will be included to the current manuscript.

**Review: It would also be useful to make some assessment of the role of mixing, again in order to facilitate quantitative comparisons with the mean meridional flow from models,**

**for instance.**

**Reply:** We agree that this is interesting, and we have actually submitted a research proposal which will tackle exactly this question. We think that this is a research topic in its own right and defer this to a future paper.

**Planned Action:** none for this paper.

**Review #3 (This reply refers to the uploaded comment, not to the comment sent to the editor. There are differences between these):**

**Review: In this paper, the authors use measurements of a variety of trace gases from MIPAS to infer the stratospheric and mesospheric circulation. They calculate a climatology and determine that the deep branch of the Brewer Dobson circulation is connected to the mesospheric pole-to-pole circulation. They verify a number of known characteristics of the circulation, such as sudden stratospheric warmings increasing variability.**

**Reply:** We are happy that the reviewer confirms that the presented climatology verifies a number of characteristics of the BDC.

**Planned Action:** none to this point.

**Review: Using chemical tracers to infer the circulation is an excellent idea. Tracers are what we can measure from space, so to validate any model, we need to quantitatively relate the tracers to the dynamical output from climate models.**

**Reply:** We agree that tracer measurements are essential for an empirical assessment of circulation. However, validating modelled tracer distributions is not enough, particularly if there are discrepancies between the model prediction and the observed atmospheric state. We are primarily studying the atmosphere but not (climate) models that try to reproduce the atmospheric processes correctly. Adjusting the models to the observed processes is a second, nevertheless necessary step that is, however, not our primary concern in this paper.

**Planned Action:** none to this point.

**Review: The inverse methods used here are promising. Unfortunately, the approach from the authors is lacking in a number of ways. 1) The validity of the method has not**

**been established.**

**Reply:** Our method is clearly based on the established validity of the continuity equation. And as the Reviewer acknowledges here above the results we obtain successfully reproduce a number of BDC characteristics. In addition a validation paper is ready for submission.

**Planned Action:** none to this point. However, a validation of the algorithm will be published in a separate paper; see reply to reviewer #1.

**Review: 2) Uncertainties are not calculated and [...],**

**Reply:** Reviewer #1 has raised a similar concern before. We consider the estimated uncertainties as largely redundant. We present the variabilities of the results. These include both the precision of the results and the natural variability. The variance describing the precision of the results cannot be larger than the variance describing the variability of the results. Thus, our presented variabilities are an upper estimate of the precision of our method.

**Planned Action:** See our related reply to Reviewer #1.

**Review: perhaps most importantly, 3) The utility of the resulting product from the method is unclear.**

**Reply:** The utility is that we provide an empirical diagnostic of the stratospheric circulation which does not suffer from the main drawbacks of either the direct comparison of modeled versus observed trace gas fields or the age-of-air based methods. While the former is very unspecific with respect to causes of discrepancies, the latter's drawbacks are the dependence of assumed age of air spectra and artificial overaging to unaccounted mesospheric sinks of tracers. Our results contain considerably more information than the trace gas fields and their variation with time, and it provides a better time-resolved understanding of the circulation than the age-of-air method (which integrates over the time an air parcel spent in the stratosphere).

**Planned Action:** We will include a couple of sentences specifically stating the utility of our product and the capability of our method.

**Review: The authors have not done any test that would demonstrate that this inversion does actually recover velocity fields in a model.**

**Reply:** Any comparison with model results would suffer from the fact that in the case of discrepancies it is not clear if they are caused by a failure of the new method or the inadequacy of the model. Furthermore, the interpretation of 2D fields inferred from 3D model output depends on certain approximations (see Appendix of von Clarmann and Grabowski, Atmos. Chem. Phys. 16(22), 14563-14584, doi10.5194/acp-16-14563-2016", 2016). Nevertheless, we have tested how far velocity fields can be recovered by our inversion.

**Planned Action:** These tests are included in a separate validation paper whose draft will be made available to the reviewers.

**Review: The closest is a very idealized case in their 2016 paper and even in that case, idealized tracers are used instead of real tracers.**

**Reply:** The idealization was made on purpose. Only in these simple cases it is possible to predict (without the help of another model) what the result should be, and to check if, e.g., the transport scheme does what it is supposed to do. In a simple test, with a constant velocity field of x degrees per month it is straight forward to check if a structure has actually moved by x degrees in a month; how the shape of the structure is conserved; what kinds of wiggles are created. With any close-to-realistic fields it is virtually impossible to judge if any wiggles are caused by diffusion or dispersion or are real phenomena. Also the over-exaggerated structures in the idealized test are, due to the large gradients in the fields, a particular challenge for the transport scheme, and can be considered almost as a worst case study. All methods used (the McCormack transport scheme, matrix inversion, etc) are well established methods.

**Planned Action:** The validation paper with further validation test cases will be made available to the reviewers.

**Review: In order to use the method on data, essentially an entire separate modeling study needs to be performed: a) using a CCM, with full knowledge of the tracer fields used here, the inversion needs to be performed and compared to the model velocity and stream function.**

**Reply:** We agree that such a modelling study is interesting, and it is actually under way. We object, however, that such a modelling study is the only possible approach to corroborate the validity of our scheme.

**Planned Action:** Model recovery tests are included in the validation paper mentioned above.

**Review: If this is successful, then the next step is: b) with the same CCM, the sampling characteristics of MIPAS (coverage and averaging kernels) need to be applied to the tracer fields so that now the limitations due to sampling and retrieval characteristics are applied. This seems especially important for vertical and horizontal resolution.**

**Reply:** MIPAS provides dense sampling. Of course the sampling varies slightly from year to year. Still the year-to-year variability of the inferred circulation is quite small. If sampling was an issue, how can it be then explained, that in different years so similar circulation patterns are obtained?
By the way: Different standards seem to be applied to different methods. MIPAS sampling is dense, and we have representative zonal means. Other observational studies use single snapshots of the atmosphere obtained by balloon observations (e.g. Engel et al., Engel et al., 2009, 10.1038/ngeo388; cited approvingly by the reviewer) to infer the strength of the Brewer-Dobson circulation. What is the purpose of applying such a different standard to different methods? The issue of vertical resolution and related implications have already been discussed and solved in von Clarmann and Grabowski, Atmos.

Chem. Phys., 16, 14563-14584, Section 3.5.

Again, why is our work judged by a different standard than other work? We do not know any age-of-air related work where vertical resolution has been considered. In the work of the reviewer, quoted in her review, vertical resolution is not even mentioned. We expect our work to be judged by the same standards as previously published work on this field. This preaching-water-and-drinking-wine stance does not fit into a neutral review!

**Planned Action:** none to this point.

**Review: Then the inversion needs to be done and compared once again to the model velocity and stream function output. This test will illuminate what the method actually means.**

**Reply:** What the method actually means is quite clear: it provides the most possible direct observational access to the temporally and spatially resolved effective 2D circulation. The appendix of the paper explains how the same quantities shown in the paper can be derived from 3D model fields. This allows a direct comparison when a model validation is the topic of further work.

**Planned Action:** see the discussion of the model recovery tests and the reply to reviewer #1.

**Review: The errors caused by the method with full tracer fields and then with the limited sampling can then be characterized for the model as well, hopefully beginning to address point 2) above.**

**Reply:** The sampling in one month over the years does vary. Still we get small standard deviations. This furnishes evidence that the method is not sensitive to MIPAS sampling issues.

**Planned Action:** We will include this argument in the paper.

**Review: This would also work towards addressing point 3) above. This "transport circulation" that the authors**

**obtain is not obviously relevant. Without being able to meaningfully compare values to model output or reanalysis products, this quantity does not seem to be of interest,[...]**

**Reply:** Our study does provide a new dataset based on observations to characterise atmospheric circulation processes, which will also additionally serve for model and reanalysis comparisons. The appendix of the paper explains how the same quantities as presented from observational data can be produced from model data or reanalyses. This allows a far more detailed comparison/model validation than age of air or the quantity "strength of the BDC". Since the BDC was posited to explain the effective transport of trace gases from low to high latitudes, our effective velocities capture the essence of the BDC, and every move towards quantities represented in 3D models would move away from this nature of the BDC. Actually the models fall short to predicting temporally and spatially resolved measures of the circulation which can be directly validated by observations. The stance that only observations which are related to model output are relevant is not tenable. Since Ian Hacking (1983, Representing and Intervening, Cambridge University Press) it is established that the task of observations goes beyond the mere verification of model predictions and that empirical science is a science in its own right.

**Planned Action:** none

**Review: [...] and so the authors claims of being able to assess quantitatively the variability of the circulation fall flat.**

**Reply:** In this paper we have focussed on the structures and the seasonal variations of these structures of the BDC. We would like to point out that between purely qualitative work and fully quantitative work, there is the wide field of work on structures (often ignored. Too many people misconceive qualitative vs. quantitative as a dichotomy!). Further, the conclusions of our paper do not depend on the absolute accuracies of the inferred effective velocities.

**Planned Action:** none

**Review: In fact, this is the reason that age of air is such**

**a useful tracer – it has been quantitatively related to the circulation of the stratosphere in a way that allows direct comparison of data (including the MIPAS data) to models (Linz et al. 2017).**

**Reply:** The age of air cannot be directly observed, it must be inferred from tracer measurements. This inference of the age of air from tracer measurements is based on assumptions, some of which we challenge. The method we present in this paper does not make these assumptions. We do not state that age-of-air based methods are not useful or should not be used by models, but we have used age-of-air based methods long enough to know their weaknesses and to find legitimate to search for alternative methods which avoid some known weaknesses. And as said above, the quantities derived from observations in this paper can all be calculated from models and reanalyses as well, as described in the Appendix.

**Planned Action:** none

**Review: I would strongly encourage the authors to perform such a study and then to rethink their results for this work in the context of the information provided by their validation study. That would be an excellent paper that I would be truly excited to see.**

**Reply:** A validation study (model recovery test) will be presented but we do not consider a model-based validation study as the optimal approach.

**Planned Action:** see reply to reviewer #1.

**Review: Beyond this overall assessment, I have included more detailed comments below: 24: What about the lifting of the circulation? (e.g. Oberlander-Hayn 2016)**

**Reply:** In our paper, we deal with the climatology of transport vectors and their year-to-year variability. Long-term trends as tackled in the Oberländer-Hayn et al. paper are beyond the scope of our paper.

**Planned Action:** Nevertheless, we will mention the paper in the introduction.

**Review: Overall introduction: What is the gap that this research is filling? The introduction reviews the literature but does not identify any motivation for the present study.**

**Reply:** It does. The motivation is clearly stated in the sentence criticised before: "In this study we aim to provide a picture of the meridional middle atmospheric circulation (better resolved in space and time than that provided by age-of-air based methods.)"

**Planned Action:** Include "(better resolved in space and time than that provided by age-of-air based methods.)"

**Review: 2.1 This discussion of age of air is surprising. What is this so called "traditional observation-based characterization of the circulation"? The authors do not provide a citation.**

**Reply:** The review paper by Waugh and Hall (Rev. Geophys. 40(4), doi10.1029/2000RG000101, 2002) gives an excellent introduction.

**Planned Action:** We will include this reference.

**Review: Some recent work that uses age of air observations to characterize the circulation is Linz et al. 2017. Recent work by Ray et al 2016 combines the TLP with chemistry to examine transport, and the improvement this offers over that method should be addressed. Ray et al. 2010 is also a relevant comparison here.**

**Reply:** Linz et al (10.1038/NGEO30132017, 2017) have characterized the BDC by a single profile. Ray et al. (10.1029/2010JD014206, 2010 and 10.1002/2015JD024447, 2016) introduce the leaky pipe model to explain age of air, ozone, CFCs, and their trends. However, since we do not deal with age of air in this paper, we do not

see how these papers are related to our work. The reviewer seems to assume that we are not familiar with the current literature and the approaches used so far. We wish to state that the contrary is true, and our previous work with age of air has made us aware of the weaknesses of this approach and the need to find an improved observation-based access to the BDC.

**Planned Action:** none

**Review: Furthermore, it is not clear how the method can reveal causes of "discrepancies" between these age and chemical tracer based methods since there are no error estimates.**

**Reply:** The standard deviations representing the precision are by definition smaller than the standard deviations that we show. Thus the standard deviations can serve as upper estimates of the random errors of each monthly field. The uncertainty of the average over the years is accordingly smaller. The rationale behind our claim that our quantities are better suited to determine causes of discrepancies than age of air is fairly trivial. If, say, in the polar stratosphere there is a discrepancy, we still do not know when (since the air entered the stratosphere) or where (along the trajectory of the air parcel) the discrepancy was caused. Our method provides quantities that are resolved latitudinally, vertically and temporally. Thus a much clearer idea can be developed where the model world and the observational world begin to diverge. This is exactly where our method provides an advantage over the age-of-air based methods.

**Planned Action:** see reply to reviewer #1

**Review: 2.1.2 There is no discussion of degrees of freedom. How much independent information is gained by including additional tracers?**

**Reply:** Tests have been shown that inclusion of further species predominantly reduce the error estimates. This effect is seen even in cases where the resulting circulation does not change. For more details, see response to Reviewer #1.

**Planned Action:** see validation paper.

**Review: Specifically, how are sinks included?**

**Reply:** This is described in Section 2.1.2.

**Planned Action:** Some clarifying amendments as requested by reviewer #1 will be included in the description of the sinks.

**Review: 3.1: Plots are very hard to understand. Streamfunctions would be much better.**

**Reply:** We concede that the changing colour scales between the panels of a figure were not optimal. To better serve the needs of the data users, we will make the data available. Then every user can represent the data in their favourite way. Our vector representation offers the advantage that it can directly be compared to the often reproduced schematic by Boenisch et al.

**Planned Action:** Colour scales will be homogenized.

**Review: 3.1.1 How are horizontal transport barriers identified? Why, physically, are they associated with this vertical motion? Is this purely a result of continuity and the fact that this is a 2-D calculation? If so, this should not be referred to as a barrier.**

**Reply:** We have identified horizontal transport barriers as consecutive latitude/altitude bins where the meridional transport velocity is zero, while meridional transport vectors point in opposite directions in the two meridionally adjacent bins. We agree with Reviewer #1 that this might indeed just identify a splitting or bifurcation and that our wording needs editing.

**Planned Action:** We will rephrase lines 133-134 as specified in reply to reviewer #1.

**Review: 3.1.6 How precisely do you identify that this circu-**

lation is associated with the monsoon? Are there particular tracers (e.g. water vapor) that mark this as a monsoon signal? Or is it just about the timing and the fact that it's in the Northern Hemisphere, in which case the link is suggested at best. "Our results show overall agreement with the one shown by Ploeger et al. (2017)[...].";

**Reply:** Indeed it is the timing, the altitude range and the fact that it appears in the NH only, that we link this to the monsoon.

**Planned Action:** We will edit the main text to make this clearer.

**Review: What "one"?**

**Reply:** Their Fig. 5

**Planned Action:** see reply to reviewer #1.

**Review: What is meant by "overall agreement"?**

**Reply:** We mean agreement related to the structures.

**Planned Action:** We will write "structural agreement."

**Review: 3.2.1 337: What is meant by this? How does this reconcile with the well-established water vapor tape recorder results?**

**Reply:** We concede that our wording may lead to misunderstanding.

**Planned Action:** We will reword this statement.

**Review: 368-374: This seems to be saying that this study is a good validation of the method. That may be, but it's not the stated goal, and more stringent validation is needed especially so as to be able to actually interpret the resulting "effective velocities".**

**Reply:** We think that these indicators of robustness are an important piece of information. They are by no means obsolete, even with a model recovery test in place. The validation will be published in a separate paper.

**Planned Action:** Model recovery tests will be performed and published in a separate paper which will be made available to the reviewers.

**Review: 384: What applications would use these effective velocities?**

**Reply:** The Brewer-Dobson circulation explains large ozone amounts over the poles while ozone is predominantly generated at low latitudes. In other words, it uses the effective 2D transport as an explanation of the trace gas distributions in the stratosphere. Thus, the effective 2D transport velocities are the natural measure of the BDC because they directly capture the essence of the BDC.

Further, these effective velocities can be understood as inverse age increments per segment of a mean trajectory. They can thus be related to the age of air but are time-resolved. The effective velocities are an empirical diagnostic in their own right. And in the appendix we relate them to the usual model quantities. Review #2 provides evidence that a significant part of the community regards this quantity as useful.

Regardless if models are able to produce such quantities or not, we think that we can learn a lot from them: Trends, variabilities etc. In this first scientific application paper we have restricted ourselves to climatologies, because these depend less on quantitative validation. The observed transport patterns and their variation contain a wealth of information in a structural sense even if one is sceptical about the associated numbers.

**Planned Action:** Some sentences on possible future work will be included.

---

## Author Response (AR1)

The authors thank the reviewers for their thorough, detailed, and insightful reviews. We have used all these recommendations to improve the clarity of this paper. In the following we insert our replies directly into the review.

**Review #1:**

**Review: General Comments**
von Clarmann et al. 2019 present results from an inverse method which uses observed (MIPAS) zonal-mean tracer fields to calculate residual circulation fields which are resolved in altitude, latitude, and time. This work expands on the work of von Clarmann and Grabowski 2016 (hereafter CG16) by providing time-resolved circulation fields, and continues the line of investigation of a number of other studies which have sought to constrain the strength of the residual circulation. However, the present work seeks to provide a substantial expansion in this direction by quantifying the circulation strength in terms of two-dimensional, time-resolved velocities. Only one previous work, to my knowledge, has quantified velocities at all - that being Fu, Hu, and Yang 2007 GRL - but this was only for a single profile of upwelling, while other work has provided some sense of two-dimensional motion but without velocities, such as the work of Stiller et al. 2012.
The results show several inconsistencies with current theory. For example: The mesopheric overturning circulation is considerably higher (at least 10 km, which seems very unexpected) when southward-bound as opposed to northward-bound; the tropical pipe shows quite a bit of meridional movement rather than isolated upwelling. If reliable, such results would be of substantial and immediate interest for a large section of the middle atmospheric research community.

**Reply:** The authors thank the reviewer for the appreciation of our results.

**Action:** None to this point.

**Review:** However, there are considerable issues with the

validity of the results, and I do not think the work should be published until they are addressed. I outline them in the following three points:

1. The inverse model robustness (specifically in terms of sensitivity to input fields) has not been explored. My impression from reading CG16 is that the inverse method requires multiple tracers but that the limit on the minimum number of tracers needed is rather soft (i.e. it is not strictly necessary to have X or more tracers). I suppose that it is possible to use a subset of the nine tracers applied, and thereby estimate the robustness of the method with respect to input data. In my opinion, having even a simple estimate is necessary. Having read this paper, I am left without an idea of how the results depend on the input tracer fields. Even if the method is mathematically sound, there may be biases in the results which extend from errors in tracer measurement or the calculation of chemical sources and sinks, and until this possibility is addressed the results remain in a somewhat skeptical position. In particular, I would propose something like a jackknife test, calculating the fields with only eight tracers, excluding one tracer for each calculation, each tracer in turn, and seeing how strongly the velocity fields vary. It may not need to be a recalculation of the entire approx. decade-long climatology, but I think a test like this (or something more advanced) in multiple seasons is necessary to establish the validity of the method.

**Reply:** We have meanwhile submitted a validation paper of the ANCISTRUS (Analysis of the circulation of the stratosphere using spectroscopic measurements) method available (enclosed). It includes an assessment of the relevance of sinks versus transport of patterns, the jackknife tests, model recovery tests, and an assessment of the adequacy of the regularization strength chosen. In a nutshell, the results are: below 30 km ANCISTRUS is quite accurate in a quantitative sense. Above, due to the regularization of the inversion along with less measurement information, peak velocities are underestimated. All structures and patterns, however, are nicely reproduced, and no artifical patterns are generated. Since none of

the conclusions of the paper under review are fully quantitative but are related to structures and patterns, we are confident that the validity of our method for the purpose of our paper has now been sufficiently established, and it can safely be excluded that the patterns detected are mere artefacts.

**Action:** The draft of the validation paper is enclosed to the resubmission. In the climatology paper we make reference to the results of the validation paper.

**Review: 2. The inverse model accuracy has not been quantified well. In CG16, an accuracy test was performed where dynamical quantities were prescribed in simple distributions, which showed that the method had some inaccuracy in the center of the domain and much more inaccuracy on the borders.**

**Reply:** The inaccuracies at the borders were caused by the fact that the test fields, which were chosen in an ad hoc manner, did not satisfy the continuity equation. A method that allows only solutions which do satisfy the continuity equation will never be able to reproduce such fields. That is to say, the problem was the test fields, not the method. More adequate tests are included in the validation paper mentioned above.

**Action:** See above.

**Review: I think a test with dynamical and chemical fields that are more similar to observations (i.e. less homogenous and, simply-said, messy) is necessary to ascertain the uncertainty of the results. I would suggest using CCM model data to produced inversted circulation fields and compare them with the actual, directly-calculated model fields. To be clear, the results of such a test could not be used to pin a quantity directly onto the method's results (since, for example, the inverse method includes some effects of mixing, although knowing how the "effective" velocities could differ from the actual velocities would be valuable). However, this would provide some level of confidence in the**

results where, at the moment, the only indication extends from an example which simply does not resemble the observed atmosphere. As a final note on this, it would be best if the test could also examine the difference between CCM and inverted velocities, and not the residual between tracer distributions (as was done in CG16). The velocities are what matter, after all. As an example of possible data, the ESCIMO (Joeckel et al. 2016 GMD) simulations have a variety of chemical species included (all the chemicals used in this study are in the model output) and calculations of the residual circulation strength have also been performed. I am not involved in that work, but I would guess that the members of that project would be interested in sharing the data.

**Reply:** A validation with realistic test fields has been performed and included in the validation paper mentioned above. Comparisons to models are interesting in their own right, and this is on our agenda for the future. We think, however, that validation does not necessarily have to rely on model data.

**Action:** See above.

**Review: 3. The inverse model result uncertainty has not been quantified. This is a rather small point, and not as important as the previous two. In CG16, quantifications of uncertainties of wind and diffusion coefficient fields were shown (in Figure 5) for fields computed using MIPAS data. Those uncertainties seem rather small, but it seems imprudent to exclude an estimation of uncertainties when a method for estimating them is possible. I suggest including a description or depiction of those uncertainties, if the method still applies for the present work.**

**Reply:** The method is still applicable and we have the data available, but we consider this as largely redundant in this context. We present the standard deviations of the monthly averages, and this quantity includes both the uncertainty of the inversion and the year-to-year atmospheric variability. Thus, the standard deviations characterize how well the climatology of a month can represent any particular month of the sample. The CG16 estimates of the random uncertainty represent the mapping of the measurement errors on the inferred velocity fields only, but not the representativeness problems due to natural variability. But since both types of uncertainties are independent from each other and add up quadratically, small standard deviations indicate that also the measurement-error-based uncertainties must be small. The standard deviations are even a more reliable estimate of the upper bound of the precision because they do not depend on any assumption on the measurement uncertainty.

**Action:** We have added to the manuscript: "(To diagnose this effect, the standard deviations of the circulation vectors, which are a measure of their variability, are also shown in Figs 5–8.) This variability is caused by the natural variability of the circulation over the years and its random uncertainty. The latter is the random uncertainty of the MIPAS measurements propagated onto the circulation vectors.

**Review: In my view, the first two points are necessary before these results should be published. Without a clear indication of the model validity, the novel results shown in this work seem as likely to be artifacts of the inverse method (or the calculation of chemical balances) as they are to correspond to reality.**

**Reply:** The validation is reported in a separate paper, see above. There it is shown that the method does not create artificial patterns.

**Action:** As stated above, the draft of the validation paper is enclosed to the resubmission, and in the revised version of the paper under review we make reference to it..

**Review: Furthermore, I find that the figures in the manuscript are difficult to interpret, and should be changed before publication. I have made more specific comments on this topic below. In general, the figures provide a qualitative idea of the circulation, in that I can examine the figures and know where the circulation is the strongest and which way**

it is headed at any one time and know how that strongest location changes with time. That does provide a general characterization of the circulation, but knowing how the circulation strength changes with time in each location is of equal value to knowing where the circulation is strongest. The changing color scales in the first 12 panels makes that assessment difficult.

**Reply:** While we think that a qualitative empirical representation at this detail level is still unprecedented, we do agree that the figures should be improved.

**Action:** Circulation patterns are now represented on a common color scale. Beyond this, the original data will be made publically available via KITOpen, with a document identification number, thus fully quotable, allowing each user to plot them in their preferred representation.

**Review:** The figures showing standard devations need to be explained clearly, because they are so unusual, but do not seem to be properly explained anywhere. I think the figures should be remade showing contours or heatmaps, or some combination of both methods, so that readers can assess the strength of the vertical and meridional velocities and the variability of them. Maybe including stream lines would maintain the ease in understanding the flow direction. I understand that this likely means a doubling of the number of figures (assuming there is not a clever way to combine the meridional and vertical information), but I think that it is warranted for the clarity of interpretation.

**Reply:** We have decided to follow the suggestion to represent the variabilities in a different way.

**Action:** Separate plots for variabilities in $v$ and $w$ are now provided.

**Review:** Furthermore, I have the following two points which I think warrant explanation, but are not addressed in the text: Why do boundaries seem to be included in results,

**when they were problematic following CG16? Are they
somehow excluded?**

**Reply:** No, boundaries are not excluded. These problems in CG16
were not a problem of our transport and inversion schemes but they
were due to the simplified velocity fields used there which were not
consistent with the continuity equation. E.g., if I have a poleward
flow but no backward flow it does not come as a surprise that funny
things happen near the pole. In CG16 such simple velocity fields
were used because they allowed much simpler tests whether the
transport scheme does what it is supposed to do, e.g. if a feature
is transported with the right velocity etc. Applied to real data this
type of problem does not exist because the true concentration fields
are well described by the continuity equation, and the Earth's air is
not forced to accumulate at the pole because of unrealistic velocity
fields, or to do other funny things.

**Action:** None specific to this point. We have, however, summa-
rized the results of the validation paper and discuss which kind of
conclusions are supported by the data.

**Review: How were the sources and sinks of each species
calculated?**

**Reply:** The general description of sources and sinks was already
described in Section 2.1.2 of the original paper. Some missing in-
formation has been included, particularly the assumptions on the
abundancies of involved species.

**Action:** For details, see below under comment on line 78.

**Review: To conclude, I would like to stress that results
from this method should be published in the future if the
reliability of the method can be addressed. They would be
of very substantial interest, and therefore I wholeheartedly
recommend resubmission.**

**Reply:** A validation paper is ready has been submitted. It confirms
that the conclusions in the paper under review are robust.

In this context we would like to mention that during the recent meeting of the EGU's publication committee (Utrecht, NL, 1 Oct 2019), which is responsible for the EGU journals including ACP, the following guideline was agreed: In the case of novel methods or results based on novel methods the authors should be given the benefit of doubt, and the risk should be taken that it may be necessary to revise these results later. After the preparation of our validation paper we think that we actually do not need to invoke this board decision, but we think that it applies *a fortiori* when a validation is available.

**Action:** A validation paper is now available, see above.

**Review: Specific Comments**
Line 14: Neither Brewer nor Dobson suggest upper and lower branches to the BDC, nor the mesospheric overtunring circulation. I think it would be best to cite somebody here who talks about that. Maybe Butchart's 2014 review paper would be good, to provide a general reference.

**Reply:** Agreed.

**Action:** The related paragraph has been rewritten and the Butchart (Rev. Geophys. 52(2), doi10.1002/2013RG000448, 157-184, 2014) reference has been included.

**Review: Line 20:** The new sentence in this line seems to dismiss the value of having a single estimate of the upwelling mass flux / intensity. Perhaps it's just the use of the words "merely" and "far" (more/too). I'm not sure if anyone has ever suggested that a single quantity could sufficiently characterize the circulation, but certainly the upwelling mass flux has provided a lot of value as a broad estimate of the circulation. I suggest simplifying the sentence - "These studies suggest that the ... is too complicated and detailed to be fully characterized by a scalar intensity." - or something similar.

**Reply:** agreed to remove the words which make the sentence sound dismissive.

**Action:** These words have been deleted.

**Review: Line 56: I don't understand what is meant by point b. Stiller et al. also uses MIPAS data. What does the present method do differently? Since the chemistry of $SF_6$ is not considered, any chemistry that is actually happening would create biases in those regions where it occurs (but I am no expert on this, and cannot say if those regions are included here). As a secondary point, there is no over-aging involved in this method because ages are not calculated. What would you expect in the case of your inversion method, if you had this bias? I would assume the result would be a faster lower-strat to upper-middle atmosphere circulation.**

**Reply:** The main $SF_6$ depletion happens in the mesosphere. Thus in the following we refer only to air parcels travelling through the mesosphere and back to the stratosphere. The method by Stiller et al., as well as earlier studies using this approach, are sensitive to the destruction of $SF_6$ along the entire trajectory of an air parcel from the stratospheric entry point, through the stratosphere and mesosphere and back into the stratosphere, because the age is calculated by comparison of actual $SF_6$ stratospheric mixing ratios and past $SF_6$ mixing ratios at the entry-point. Our method is different in that we use $SF_6$ mixing ratios measured at the upper boundary as a reference for gradient calculations in the uppermost model layer. Thus, any calculation of differences used for the calculation of the gradients (needed to solve the transport equation) is based on $SF_6$ mixing ratios which are, if coming from the mesosphere, already depleted in $SF_6$. Thus, mesospheric $SF_6$ destruction cannot lead to artefacts in the gradients. The only $SF_6$ loss that we possibly miss is $SF_6$ destruction within the **stratosphere** from one month to the other, which is a minor inaccuracy compared to the problem of mesospheric $SF_6$ depletion in age-of-air applications. In short: The reference $SF_6$ used by Stiller is the (past) tropospheric $SF_6$ while

our reference $SF_6$ is the depleted $SF_6$ in airmasses intruding from the mesosphere.

Beyond this, one result of our validation paper is that $SF_6$ contributes relatively little information. Therefore, inaccuracies due to the neglect of $SF_6$ sinks inside our analysis domain are not very relevant in quantitative terms.

**Action:** A clarifying sentence has been added.

**Review: Line 78: Where did the data for OH, $O^1(D)$, and Cl come from? As I understand it, those JPL publications only contain reaction rate and cross-section information. Did you obtain that data from somewhere else (does MIPAS have those species?) or did you model those in some other way?**

**Reply:** We estimate OH using the parametrization by Minschwaner et al. (Atmos. Chem. Phys. 11(3), 955-962, doi10.5194/acp-11-955-2011, 2011). $O^1(D)$ is estimated using the equilibrium equation 5.38 in Brasseur & Solomon (2005, Springer), applied to MIPAS ozone; Cl is estimated by interpolating a climatological noon profile (Brasseur & Solomon, 2005, Fig 5.50) to the actual atmospheric state. Inaccuracies in the latter are considered to be less important because the atomic Cl sink is of much less relevance than the other sinks.

**Action:** This missing information has been included in Section 2.1.2.

**Review: Line 86: How necessary is the stabilization of the inversion, that mixing coefficients were assumed to be zero? Can you compute mixing coefficients for even a single pair of months, or perhaps for a pair of boreal summer and a pair of boreal winter months? Otherwise, it's difficult to say how the effective velocities compare to advective velocities.**

**Reply:** We think that the effective velocities represent the essence of the Brewer-Dobson circulation in the sense that they conflate all the effects (advection, correlation effects, mixing) that bring, in a

2D world, a trace gas from here to there. From comparison with zonal mean advective velocities we can learn about the relative contribution of the non-advective terms.

Technically speaking, all monthly results are inferred independently. That is to say, the instability does not come from accumulation of errors over the months but is inherent in the analysis of each single month. The cause of the instability is the following: The system of equations solved tends towards linear dependencies as soon as velocities and mixing coefficients are to be retrieved simultaneously. The matrix to be inverted has an extremely high condition number. This can, in principle, be remedied by regularization. We found out, however, that in this case (contrary to the velocity-only analysis) the result depends strongly on the chosen regularization and is, thus, not robust. As a consequence, we have decided to constrain the mixing coefficients to zero and to re-interpret the resulting velocities as those 2D-velocities which best describe the combined effect of advection, eddy transport and eddy mixing. It cannot be expected that these effective velocities equal the zonal mean advective velocities.

**Action:** A clarifying sentence has been added.

**Review: Line 88: What does the word "efficient" in "efficient 2D circulation" mean? Do you mean effective? If not, the meaning should be clarified.**

**Reply:** This is a wording error and should read 'effective' instead. Thanks for spotting!

**Action:** This wording error has been corrected.

**Review: Line 89: I have never seen the term "Fickian mixing" before. Having done some searching, I think what you are referring to is more commonly called diffusion. If that's the case, I would use that term. Otherwise, the meaning should be explained.**

**Reply:** We intentionally avoid the term 'diffusion' in this context for the following reason: 'Diffusion' we understand is a physical process happening on a molecular scale. The processes we describe still abide to Fick's law of diffusion but are macroscopic processes. Thus we consider the term 'diffusion' in this context as misleading.

**Action:** We have added a footnote explaining the meaning of 'Fickian mixing'.

**Review: Section 2.3: Why do you average every pair of months? I would guess that is due to interannual variability of the phase (and I use that word very loosely here, perhaps it would be better to say timing) of the circulation. Whatever the case is, it should be stated.**

**Reply:** There seems to be a fundamental misunderstanding. We do not average over two months. The velocity field labelled, say, March-April, is the velocity field that best reproduces the change of the monthly mean March mixing ratio field to the monthly mean April mixing ratio field.

**Action:** We have added some clarification to avoid this misunderstanding. We think, however, that this clarification fits much better in the Section 'The General Approach' than here; thus we have included it there.

**Review: Figures 1,2,3, and 4: I find these figures very difficult to interpret. First, the changing color scale does not seem necessary, as the maximum values do not vary strongly between the figures - most of them are around 11/12 - although it would create an issue for the December-January figure. At the moment, however, it is difficult to assess how the magnitude of the circulation changes at each point, month-to-month, except in the most starkly contrasting cases.**

**Reply:** We do agree that the changing colour scales of the original submission were disadvantageous.

**Action:** New plots with fixed colour scales have been included.

**Review: Second, it is difficult (if not for most cases prati­cally impossible) to obtain even a rough magnitude of the vector components because the color scale shows a norm of the vectors. I suggest using contours or heatmaps of the separate vector components instead.**

**Reply:** The length and direction of the arrows represent the veloc­ity components. The colour scale was meant just as an additional guide of the eye. The length units of the arrows are ad hoc, that is to say, they are not consistent with the $\phi$ and $z$ axes intervals of the plots.

**Action:** The original data will be made available in digital form on tha data repository 'KITOpen', as described above.

**Review: Third - and this has nothing to do with inter­pretation - you show the boundary velocities of these re­sults although it is clear from CG16 that the vectors at the boundaries are difficult for the inverse model.**

**Reply:** As already stated above, within the reply to the general comments, the problem at the boundaries is not a problem of the inversion scheme but a problem of inconsistent test data which rep­resent non-realistic circulations where air accumulates at the bound­ary of the domain. By the way, the arrows in the boundary tiles refer to transport in the 80-90 degrees latitude band (and not to the 90 deg. latitude; i,e., we have no northward transport at the North pole.)

**Action:** None.

**Review: Figures 5 and 6: These figures are unusual and require some considerable effort to comprehend. I assume the width versus height of the bubbles shows the stardard deviation in the meridional and vertical directions, and that the colors show the standard devation of the norm, but no information is given about that. I would suggest simply replacing these with contours and/or heat maps.**

**Reply:** Agreed.

**Action:** We now show variabilities in $v$ and $w$ in separate figures.

**Review: All figures: I suggest using a perceptually uniform colorscale, which makes viewing much easier for those who do not have a standard perception of color.**

**Reply:** We have tried many different colour scales but the alternatives did not seem convincing to us.

**Action:** We now present the same plots in a different color scale in the supplement. Beyond this, we will make the original data available, thus the readers can plot then in their preferred representation.

**Review: Line 133: I am not sure that I can agree that the vertical motion over this range creates a transport barrier. I would rather say it suggests one, at the most. But it could also be interpreted as a latitude (or a section of a latitude) where the circulation splits. In that case, there's not really a barrier to horizontal transport, but only relatively little forcing towards horizontal transport / a stronger forcing towards vertical transport. I'm not sure what the case truly is, but I dont think it's certain that this represents a barrier.**

**Reply:** We will rephrase this sentence to avoid misunderstanding.

**Action:** We have rephrased lines 133-134 to: "The direct vertical motion over 30°S suggests the existence of a region where horizontal transport is minimal compared to vertical transport; the location of this region is in good agreement with the location of the subtropical transport barrier (e.g. Stiller et al., 2017)."

**Review: Line 138: "would likely look" - I think it's highly likely, even, but no definitive statement can be made until the analysis is done. On that topic, I think that analysis would be very interesting for future work.**

**Reply:** Agreed.

**Action:** We have reworded this: "Representation in equivalent latitudes would be more adequate to analyze this phenomenon but since that representation would not be optimal for global analyses, it is deferred to a future study."

**Review: Line 142: You might consider showing the tropopause and stratopause levels in your figures. I think that would be very helpful, and could alleviate a lot of confusion.**

**Reply:** Monthly averaged tropopause altitudes can be very misleading.

**Action:** none

**Review: Line 150: It would be more useful to replace the values shown in Figures 5 and 6 with the values discussed in this sentence. That would be a more direct indication for the reader of where the circulation is consistent. Otherwise, they need to compare these values themselves, which is rather tedious if a thorough comparison is desired.**

**Reply:** If we understand correctly, the reviewer recommends to present the ratio between variability and absolute velocity instead of the variabilities themselves. (in the sense "inferred velocities exceed their variabilities by a factor of ..."). We tried this but due to the often small velocities, this representation is not easy to interpret either. The plot would be dominated by large but meaningless ratios related to tiny velocities while regions of interest with large absolute variabilities would no longer be obvious. That is why we didn't choose this representation.

**Action:** none

**Review: Line 189: Leaving aside the term "latitudinal barrier" (again, I am not sure how to distinguish between a barrier to latitudinal transport or a region of weak latitudinal transport), I do not see that agree that with the term**

**"contribute to the formation".**

**Reply:** We have rephrased this sentence.

**Action:** Lines 189-190 have been rewritten as "This feature will evolve in the following months as a region where uplift motion clearly overtakes horizontal transport around 60°N."

**Review: Line 234: I've mentioned this already, but I think this case shows clearly why the variability should be depicted in another way. It's too difficult to compare the standard deviations here to the circulation strength, for the most part. But your argument does seem plausible.**

**Reply:** We agree that this is difficult with the original plots.

**Action:** We now quote the numbers in the text and and rephrase to make it clearer that the figure illustrates our argument rather than quantify it.

**Review: Line 237: It seems like you wanted to specify a figure in Ploeger et al. 2017. I suppose you mean Figure 5? You should specify the figure.**

**Reply:** Agreed. Indeed we mean Figure 5.

**Action:** The Figure has been specified.

**Review: Line 323: What is independent about the ANCISTRUS results?**

**Reply:** ANCISTRUS results are independent from each other in the sense that the result of an ANCISTRUS run for one month is never used a first guess, a priori, or similar of an ACISTRUS run of another month. All ANCISTRUS runs can thus be performed independently, and any artificial autocorrelation of the results is thus excluded.

**Action:** We have slightly reworded this for clarity: "from the results of independent ANCISTRUS runs".

**Review: Line 323: "resulting fields are stable" – The statement regarding field stability should be more nuanced. Some parts of the fields do seem to be stable, sure, but this statement suggests that the fields are generally/always consistent.**

**Reply:** Agreed.

**Action:** This statement has been made more specific: "resulting fields are stable over the years of the MIPAS mission (2002–2012) in the sense that the annual variation of the resulting circulation patterns is large only in regions where one would expect large natural interannual variability.

**Review: Line 324: "increases confidence in the robustness of the analysis method" - I do not agree. If the method is robust, then a rather stable circulation field over the approx. decade of measurements in one region would suggest that the circulation field is a typical phenomenon for that region, at least for that decade. However, the robustness of the inverese method cannot be assessed by seeing consistency in its results without a second point of reference (preferably, other observation-based circulation estimates, which - to my knowledge - do not exist).**

**Reply:** Well, this depends what how the term 'robustness' is understood. We understand 'robustness' as the characteristic that the solution is not overly sensitive to varying input. We do not claim here to have shown that the method is accurate.
That is also why our initial wording was "increases confidence in the robustness" instead of "shows the results are robust".

**Action:** We have clarified what we mean: " The stability of results from independent ANCISTRUS runs increases confidence in the robustness of the analysis method in the sense that it produces similar results for similar input fields."

**Review: Line 330: It's not clear to me if a clear separation between these two pathways would be expected or not. Could you provide any context on that, in terms of earlier literature?**

**Reply:** Agreed

**Action:** we have inserted: "..., as suggested by the schematic shown in Fig. 1 of Boenisch et al. (2011)..."; Reference: Atmos. Chem. Phys. 11(8), 3937-3948, doi10.5194/acp-11-3937-2011, 2011),

**Review: Line 333: "consistent with the assumption" - Has anyone previously suggested this idea, or are you saying that your results would only be consistent with a northward pole-to-pole circulation if it was above the domain of MIPAS? If nobody has suggested this, then this statement should be written differently to clarify the novelty of the result.**

**Reply:** Well, we think that the existence of the pole-to-pole overturning circulation is well established. We see velocities going up in the north and downward velocities in the south, but the meridional velocities which would close this circulation are above our data domain. We neither claim to have found a novel circulation path nor do we refer to a specific assumption written in the literature. Thus our careful wording.

**Action:** We have rewritten this without the term 'assumption': "Our data are consistent with - but do not directly support - a southward pole to pole circulation from March to May at altitudes not covered by MIPAS data"

**Review: Line 337: To my understanding, this intrepretation of the tropical pipe would be novel. I only wish to note that here.**

**Reply:** Yes, indeed. We have intentionally chosen a very careful wording here ('suggests'...'may not be'...). We have now made the wording even more careful.

**Action:** We have changed the wording to "This seems to suggest that...not always...

**Review: Line 341: This could be consistent with some earlier results. See Butchart 2014 (The Brewer-Dobson Circulation, Rev. Geophys.) Figure 6 and discussion of that figure. If mean downwelling during winters where the polar vortex is not disturbed is the same between both hemispheres, then one would expect stronger climatological descent in the southern hemisphere because the vortex is disturbed less often there.**

**Reply:** We understand our (old) lines 340-341 as an introductory sentence for the five following more specific points. Thus the quite specific comment seems to refer to line 342 (#1 in the list) rather than to line 341. We have added a sentence there.

**Action:** We have added to #1: "This is consistent with stronger southern than northern polar winter subsidence which is associated with less perturbed polar vortices there (Butchart 2014, Section 5.1)." Reference: Rev. Geophys. 52(2), doi10.1002/2013RG000448, 157-184 (2014)

**Review: Line 343: To my knowledge, this result is not expected.**

**Reply:** Ok, we have now mentioned this.

**Action:** We have added: "To the best of our knowledge these altitude differences have not been reported before either."

**Review: Line 347: I've mentioned it already, but I do not think the term "barriers" is justified in this context.**

**Reply:** Agreed

**Action:** We will replace the term 'barriers' by 'areas with near zero ... velocities.'

**Review: Line 348: Same to point 2.**

**Reply:** agreed.

**Action:** We have added: "To the best of our knowledge this also has not been reported before."

**Review: Line 364: In the broad stroke, I agree with this statement. However, the absence of a southward mesospheric overturning circulation means that this statement cannot be written in the absolute. Furthermore, the results do not characterize these patterns in an expected fashion (tropical pipe, for example). This statement should be rewritten to reflect those points.**

**Reply:** The overturning circulation is not absent but just not covered by the MIPAS measurements. But we agree to reword our statement.

**Action:** We have rewritten: "The ANCISTRUS method applied to MIPAS data broadly reproduces well the known atmospheric meridional circulation patterns, although with some unexpected features. Additional information ..."

**Review: Technical Comments**

**Review: Abstract, line 1: HCFC-22.**

**Reply:** Thanks for spotting!

**Action:** Corrected.

**Review: Line 14: The citation of Brewer and Dobson 1949 is incorrect. That's a single-author publication, just from Brewer.**

**Reply:** Thanks for spotting!

**Action:** Corrected.

**Review: Line 14: I think it's better to write abbreviations in a separate set of parentheses, just so it's clear that the abbreviation isn't some part of a citation.**

**Reply:** This comment has become obsolete after rewriting in reply to a specific comment (see above).

**Action:** No additional action.

**Review: Line 16: The last part of this sentence seems to suggest that only aerosols affect major chemistry-climate processes. It would be more clear with "as aerosols, all of which affect major chemistry".**

**Reply:** Agreed.

**Action:** Rephrased as suggested.

**Review: Line 19: This sentence is a bit of a run-on. It would be better to write one sentence for Engel's balloon studies and another for the satellite studies.**

**Reply:** We agree that the original sentence was too long. We prefer, however, to split the sentence immediately after the Butchart reference.

**Action:** Will have changed this to "...Butchart et al. (2006). This triggered..."; Reference: Clim. Dyn. 27(7-8), 727-741, doi10.1007/s00382-006-0162-4 (2006).

**Review: Line 22: "Offline model simulations ... analysis data have also confirmed ...", or add a comma after "Also".**

**Reply:** Thanks!

**Action:** A comma has been inserted.

**Review: Line 28:** It looks like you meant to write "has been investigated" or something similar. At the moment, the sentence doesn't make sense.

**Reply:** Agreed.

**Action:** Corrected.

**Review: Line 30:** If Funke et al. (from all those years) showed this, then I would write "has been" not "could be".

**Reply:** Agreed.

**Action:** Corrected.

**Review: Line 35:** It would be more precise to simply say that the picture of the middle atmospheric circulation is better resolved in space and time, cutting out the "more detailed" part.

**Reply:** Agreed.

**Action:** Corrected.

**Review: Line 38:** The ending, "over the years", isn't necessary here.

**Reply:** We had added 'over the years' to make clear that we do not talk about the inter-annual, not the intra-annual (month to month) variability. We have now reworded this in a clearer, less clumsy way.

**Action:** 'over the years' has been deleted, and 'inter-annual' has been inserted before 'variability'.

**Review: Line 46:** I would say not just monthly but "monthly-mean". Furthermore, the sentence suggests that this is the only way to infer the circulation, so you should specify "is inferred in this work".

**Reply:** Agreed for "in this work". The original text already reads "monthly zonal mean" and we think that it is clear that averaging was made in both domains.

**Action:** "in this work" has been added.

**Review: Line 49: "The resulting circulation fields..."**

**Reply:** Thanks!

**Action:** Corrected.

**Review: Line 62: I don't understand what "related software" refers to. That's the inversion method, right?**

**Reply:** Yes, it is.

**Action:** This has been reworded as suggested.

**Review: Line 79: "source reactions were also considered"**

**Reply:** Thanks for spotting!

**Action:** Corrected.

**Review: Line 83: "the neglect of sinks above that altitude"**

**Reply:** Thanks!

**Action:** Corrected.

**Review: Line 91: See comment on line 14 about abbreviations.**

**Reply:** As far as we know, our way to set the parentheses here is the one which complies with the COPERNICUS rules. I think here the copy editors will have the last word.

**Action:** None so far. We will wait what the copy editors say.

**Review: Line 103: "From MIPAS, measurements"**

**Reply:** MIPAS here is a sort of attribute or specifications. With the comma inserted, the meaning would change towards "measurements are calculated from MIPAS", which is not what we mean. We mean "...were calculated from MIPAS measurements..."

**Action:** None

**Review: Line 104: You explained the data gaps in the last section.**

**Reply:** Yes, indeed.

**Action:** we have included the 2006 data gap in line 95 and have reword here: "... with data gaps as reported above ."

**Review: Line 116: "up to 30 km"**

**Reply:** Agreed.

**Action:** Corrected.

**Review: Line 123: move "also" from "also the standard..." to "are also shown"(...).**

**Reply:** Thanks!

**Action:** Corrected.

**Review: (...). Furthermore, it should be clear to all readers that standard deviations are a measure of variability, so the "which are a measure of their variability" is not necessary.**

**Reply:** A standard deviation is a measure of the width of a distribution but it is not clear if it represents variability, uncertainty, probability, or other. In particular in our community, uncertainties and estimated errors are often reported in terms of standard deviation. Thus, we find it necessary to be specific here.

**Action:** The text has been reworded.

**Review: Line 127: You can just say northern hemisphere or winter hemisphere.**

**Reply:** Agreed.

**Action:** "local winter" has been deleted.

**Review: Line 146: "has its maximum ... and at 30°S"**

**Reply:** Agreed.

**Action:** This has been reworded as suggested.

**Review: Line 155: You are clearly comparing this month-pair with the previous, but this sentence should make that explicit.**

**Reply:** Agreed.

**Action:** "seen in January-February" has been inserted.

**Review: Line 167: "will give rise"**

**Reply:** Thanks!

**Action:** Corrected.

**Review: Line 170: The abbreviation SH was already used before this point. The notification of the abbrevation should be shifted to the first usage of "southern hemisphere". Ditto for NH.**

**Reply:** Thanks for spotting.

**Action:** Corrected.

**Review: Line 379: "their figure" - This part of the sentence addresses a particular figure, but the earlier part speaks generally of schematics. I suggest sticking to one approach or the other.**

**Reply:** Agreed.

**Action:** We have split the sentence to make clear what refers to such schematics in general and what refers to this particular figure.

**Review #2:**

**Review:** The authors present an estimate of meridional circulation patterns in the middle atmosphere based on measurements from 2002 though 2012 by the MIPAS instrument of a range of trace gas species. The estimate is based on an inverse method that infers an effective flow field in the meridional plane from the continuity equation along with an estimate of chemical sources and sinks. The methodology is updated from previous work by the first two authors through inclusion of further chemical sources and sinks, and by inferring only an 'effective' meridional flow that includes the effects of mixing/eddy transport.
The main results shown are the month by month estimates of the decadal-averaged meridional flow, as well as an estimate of the interannual variability of the flow. In as much as this estimate is a relatively direct observational estimate of a difficult to measure quantity, this result is of potential value to the broader community.

**Reply:** We thank the reviewer for this encouraging evaluation.

**Action:** None.

**Review:** My main concern is that if this is to be the case, enough quantitative details should be given in order to facilitate comparisons with these results; this is largely the case but there are a few ambiguities that should be addressed (see below).

**Reply:** Agreed.

**Action:** See specific actions below.

**Review:** Beyond this I have a few questions and comments about the presentation of these results (in particular I find the presentation of the interannual variability difficult to understand), but otherwise feel this is appropriate for publication with some minor revisions.

**Reply:** We agree that the presentation of the inter-annual variability was insufficiently explained. In reply to reviewer #1 we have decided to present the variabilities separately for vertical and horizontal effective velocities.

**Action:** See specific actions below.

**Review: Specific comments**
**1) As mentioned above there are a few points that would be helpful for making quantitative comparisons with these results. Firstly, does the inferred circulation conserve mass?**

**Reply:** Yes, it largely does, except for transport into and out of the model domain at the upper and lower boundary, which may not necessarily be balanced. But besides the continuity equation of species, also the continuity equation of air density is one of the determining equations of our system.

**Action:** None.

**Review: If so the authors may want to consider showing a mass stream function instead of the vector plots.**

**Reply:** Any representation which involves weighting by air density suffers from the large dynamical range of air density with altitude. We have tried various representations of this type but always all structure and information in the middle atmosphere was lost. Only values at the lowermost layer were discernable in such representations.

**Action:** None

**Review: If not, the choice of units for Figs. 1 through 4 are a bit confusing; surely the velocities should be homogeneous in units (e.g. m/s)?**

**Reply:** With homogeneous units the vertical velocities would be invisible, although important. The norm we use for the colour scale

and for the direction of the arrows roughly corresponds to the aspect ratio of the plots where the vertical axis does not represent the true geometric conditions either but is heavily exaggerated.

**Action:** The original data will be made available in digital form. With this the user can represent the data in the preferred way, most suitable for the respective application. The original data are actually provided in units of m/s.

**Review: The note regarding the colour scales in the Figures 5 and 6 show standard deviations of the inferred effective velocities, but the visualization is not explained.**

**Reply:** The explanation was indeed missing in the original submission.

**Action:** In reply to Reviewer #1, the standard deviations will be represented as heat-maps.

**Review: One assumes that the axes of the ellipses are scaled relative to the variance of the y and z components of the velocities but it seems no account is being taken of their covariance if that is the case. How are the colors chosen?**

**Reply:** The axes of the ellipses represented the standard deviations in the the y and z components. The covariances were not represented. The colours had been chosen by adjusting the colour scale to the maximum value of the individual plot.

**Action:** We have decided to represent the variabilities in separate plots for $v$ and $w$. A common colour scale is now used for the entire series of plots. Further, we mention that the same norm is applied to the standard deviations as for the effective velocities.

**Review: More importantly, are these estimates of the standard deviation of the mean (implied by the figure caption) or sample standard deviations?**

**Reply:** The title was indeed misleading. We present the sample

standard deviations, not the standard error of the mean. This is because we are interested in the variability, not in the uncertainty. The standard error of the mean would become lower for a larger sample and is thus not the adequate measure.

**Action:** Titles in the plots have been corrected.

**Review: In sum the interannual variability is difficult to assess in comparison to the mean circulation from these figures and is not very satisfyingly discussed in the text.**

**Reply:** Agreed.

**Action:** New plots are now provided and adequately described.

**Review 3) In all figures, different years are included in each panel with no discussion; why?**

**Reply:** First we have some data gaps in the MIPAS data, and second, a (small) number of the inversions did not converge.

**Action:** This information is now provided in the figure caption.

**Review: 4) The methodology used in the present work includes the role of chemical sources and sinks; this has been updated from von Clarmann and Grabowski (2016). The value of these updates should be demonstrated.**

**Reply:** The impact is indeed substantial, and we are happy to show respective plots. However, we think that this fits in much better with the paper on validation and sensitivity studies discussed in reply to reviewer #1.

**Action:** This issue has been deferred to a validation paper of which the draft is made available to the reviewers, and reference to that paper together with a brief explanation has been included to the current manuscript.

**Review: It would also be useful to make some assessment of**

**the role of mixing, again in order to facilitate quantitative comparisons with the mean meridional flow from models, for instance.**

**Reply:** We agree that this is interesting, and we have actually submitted a research proposal which will tackle exactly this question. We think that this is a research topic in its own right and defer this to a future paper.

**Action:** none for this paper.

**Review #3 (This reply refers to the uploaded comment, not to the comment sent to the editor. There are differences between these):**

**Review: In this paper, the authors use measurements of a variety of trace gases from MIPAS to infer the stratospheric and mesospheric circulation. They calculate a climatology and determine that the deep branch of the Brewer Dobson circulation is connected to the mesospheric pole-to-pole circulation. They verify a number of known characteristics of the circulation, such as sudden stratospheric warmings increasing variability.**

**Reply:** We are happy that the reviewer confirms that the presented climatology verifies a number of characteristics of the BDC.

**Action:** None to this point.

**Review: Using chemical tracers to infer the circulation is an excellent idea. Tracers are what we can measure from space, so to validate any model, we need to quantitatively relate the tracers to the dynamical output from climate models.**

**Reply:** We agree that tracer measurements are essential for an empirical assessment of circulation. However, validating modelled tracer distributions is not enough, particularly if there are discrepancies between the model prediction and the observed atmospheric state. We are primarily studying the atmosphere but not (climate) models that try to reproduce the atmospheric processes correctly. Adjusting the models to the observed processes is a second, nevertheless necessary step that is, however, not our primary concern in this paper.

**Action:** None to this point.

**Review: The inverse methods used here are promising. Unfortunately, the approach from the authors is lacking in a number of ways. 1) The validity of the method has not**

**been established.**

**Reply:** Our method is clearly based on the established validity of the continuity equation. And as the Reviewer acknowledges here above the results we obtain successfully reproduce a number of BDC characteristics. In addition a validation paper has been submitted.

**Action:** None to this point. However, the validation of the algorithm has been described in a separate manuscript that has just been submitted to ACP; the manuscript number is acp-2020-72; see reply to reviewer #1.

**Review: 2) Uncertainties are not calculated and [...],**

**Reply:** Reviewer #1 has raised a similar concern before. We consider the estimated uncertainties as largely redundant. We present the variabilities of the results. These include both the precision of the results and the natural variability. The variance describing the precision of the results cannot be larger than the variance describing the variability of the results. Thus, our presented variabilities are an upper estimate of the precision of our method.

**Action:** See our related reply to Reviewer #1.

**Review: perhaps most importantly, 3) The utility of the resulting product from the method is unclear.**

**Reply:** The utility is that we provide an empirical diagnostic of the stratospheric circulation which does not suffer from the main drawbacks of either the direct comparison of modeled versus observed trace gas fields or the age-of-air based methods. While the former is very unspecific with respect to causes of discrepancies, the latter's drawbacks are the dependence of assumed age of air spectra and artificial overaging to unaccounted mesospheric sinks of tracers. Our results contain considerably more information than the trace gas fields and their variation with time, and they provide a better time-resolved understanding of the circulation than the age-of-air method (which integrates over the time an air parcel spent in the stratosphere).

**Action:** We have included a couple of sentences specifically stating the utility of our product and the capability of our method.

**Review: The authors have not done any test that would demonstrate that this inversion does actually recover velocity fields in a model.**

**Reply:** Any comparison with model results would suffer from the fact that in the case of discrepancies it is not clear if they are caused by a failure of the new method or the inadequacy of the model. Furthermore, the interpretation of 2D fields inferred from 3D model output depends on certain approximations (see Appendix of von Clarmann and Grabowski, Atmos. Chem. Phys. 16(22), 14563-14584, doi10.5194/acp-16-14563-2016", 2016). Nevertheless, we have tested in the validation paper mentioned how far velocity fields can be recovered by our inversion.

**Action:** Model recovery tests are included in a separate validation paper whose draft is available to the reviewers.

**Review: The closest is a very idealized case in their 2016 paper and even in that case, idealized tracers are used instead of real tracers.**

**Reply:** The idealization was made on purpose. Only in these simple cases it is possible to predict (without the help of another model) what the result should be, and to check if, e.g., the transport scheme does what it is supposed to do. In a simple test, with a constant velocity field of x degrees per month it is straight forward to check if a structure has actually moved by x degrees in a month; how the shape of the structure is conserved; what kinds of wiggles are created. With any close-to-realistic fields it is virtually impossible to judge if any wiggles are caused by diffusion or dispersion or are real phenomena. Also the over-exaggerated structures in the idealized test are, due to the large gradients in the fields, a particular challenge for the transport scheme, and can be considered almost as a worst case study. All methods used (the McCormack transport scheme, matrix inversion, etc) are well established methods.

For the transport medelling we consider tha tests presented in von Clarmann and Grabowski, 2016, as severe. The validation of the inversion scheme is included in the validation paper mentioned before.

**Action:** The validation paper with further validation test cases is made available to the reviewers.

**Review: In order to use the method on data, essentially an entire separate modeling study needs to be performed: a) using a CCM, with full knowledge of the tracer fields used here, the inversion needs to be performed and compared to the model velocity and stream function.**

**Reply:** We agree that such a modelling study is interesting, and it is actually under way. We object, however, that such a modelling study is the only possible approach to corroborate the validity of our scheme.

**Action:** Model recovery tests are included in the validation paper mentioned above.

**Review: If this is successful, then the next step is: b) with the same CCM, the sampling characteristics of MIPAS (coverage and averaging kernels) need to be applied to the tracer fields so that now the limitations due to sampling and retrieval characteristics are applied. This seems especially important for vertical and horizontal resolution.**

**Reply:** MIPAS provides dense sampling. Of course the sampling varies slightly from year to year. Still the year-to-year variability of the inferred circulation is quite small. If sampling was an issue, how can it be then explained, that in different years so similar circulation patterns are obtained?
By the way: Different standards seem to be applied to different methods. MIPAS sampling is dense, and we have representative zonal means. Other observational studies use single snapshots of the atmosphere obtained by balloon observations (e.g. Engel et al., Engel et al., 2009, 10.1038/ngeo388; cited approvingly by the reviewer) to infer the strength of the Brewer-Dobson circulation. What is the

purpose of applying such a different standard to different methods? The issue of vertical resolution and related implications have already been discussed and solved in von Clarmann and Grabowski, Atmos. Chem. Phys., 16, 14563-14584, Section 3.5.

Again, why is our work judged by a different standard than other work? We do not know any age-of-air related work where vertical resolution has been considered. In the work of the reviewer, quoted in her review, vertical resolution is not even mentioned. We expect our work to be judged by the same standards as previously published work on this field. This preaching-water-and-drinking-wine stance does not fit into a neutral review!

**Planned Action:** None to this point.

**Review: Then the inversion needs to be done and compared once again to the model velocity and stream function output. This test will illuminate what the method actually means.**

**Reply:** What the method actually means is quite clear: it provides the most possible direct observational access to the temporally and spatially resolved effective 2D circulation. The appendix of the paper explains how the same quantities shown in the paper can be derived from 3D model fields. This allows a direct comparison when a model validation is the topic of further work.

**Action:** see the discussion of the model recovery tests and the reply to reviewer #1.

**Review: The errors caused by the method with full tracer fields and then with the limited sampling can then be characterized for the model as well, hopefully beginning to address point 2) above.**

**Reply:** The sampling in one month over the years does vary. Still we get small standard deviations. This furnishes evidence that the method is not sensitive to MIPAS sampling issues.

**Action:** We have included this argument in the paper.

**Review: This would also work towards addressing point 3) above. This "transport circulation" that the authors obtain is not obviously relevant. Without being able to meaningfully compare values to model output or reanalysis products, this quantity does not seem to be of interest,[...]**

**Reply:** This is a presumptuous statement from a biased modeler's perspective. Our study does provide a new dataset based on observations to characterise atmospheric circulation processes, which will also additionally serve for model and reanalysis comparisons. The appendix of the paper explains how the same quantities as presented from observational data can be produced from model data or reanalyses. This allows a far more detailed comparison/model validation than age of air or the quantity "strength of the BDC". Since the BDC was posited to explain the effective transport of trace gases from low to high latitudes, our effective velocities capture the essence of the BDC, and every move towards quantities represented in 3D models would move away from this nature of the BDC. Actually the models fall short to predicting temporally and spatially resolved measures of the circulation which can be directly validated by observations. The stance that only observations which are related to model output are relevant is not tenable. Since Ian Hacking (1983, Representing and Intervening, Cambridge University Press) it is established that the task of observations goes beyond the mere verification of model predictions and that empirical science is a science in its own right.

**Planned Action:** None.

**Review: [...] and so the authors claims of being able to assess quantitatively the variability of the circulation fall flat.**

**Reply:** In the original paper this claim has not been made.In this paper we have focussed on the structures and the seasonal variations of these structures of the BDC. We would like to point out that between purely qualitative work and fully quantitative work, there is the wide field of work on structures (often ignored. Too many people misconceive qualitative vs. quantitative as a dichotomy!). Further,

the conclusions of our paper do not depend on the absolute accuracies of the inferred effective velocities.

**Planned Action:** none

**Review: In fact, this is the reason that age of air is such a useful tracer – it has been quantitatively related to the circulation of the stratosphere in a way that allows direct comparison of data (including the MIPAS data) to models (Linz et al. 2017).**

**Reply:** The age of air cannot be directly observed, it must be inferred from tracer measurements. This inference of the age of air from tracer measurements is based on assumptions, some of which we challenge. The method we present in this paper does not make these assumptions. We do not state that age-of-air based methods are not useful or should not be used by models, but we have used age-of-air based methods long enough to know their weaknesses and to find legitimate to search for alternative methods which avoid some known weaknesses. And as said above, the quantities derived from observations in this paper can all be calculated from models and reanalyses as well, as described in the Appendix.

**Planned Action:** None.

**Review: I would strongly encourage the authors to perform such a study and then to rethink their results for this work in the context of the information provided by their validation study. That would be an excellent paper that I would be truly excited to see.**

**Reply:** A validation study (model recovery test) has been performed and the related paper has been submitted. This paper furnishes evidence that our method is robust enough for all the conclusions in the paper under discussion. For reasons discussed above, we do not consider a model-based validation study as the optimal approach.

**Action:** See reply to reviewer #1.

**Review: Beyond this overall assessment, I have included more detailed comments below: 24: What about the lifting of the circulation? (e.g. Oberlander-Hayn 2016)**

**Reply:** In our paper, we deal with the climatology of transport vectors and their year-to-year variability. Long-term trends as tackled in the Oberländer-Hayn et al. paper are beyond the scope of our paper.

**Action:** Nevertheless, we now mention their paper in the introduction.

**Review: Overall introduction: What is the gap that this research is filling? The introduction reviews the literature but does not identify any motivation for the present study.**

**Reply:** It does. The motivation is clearly stated in the sentence criticised before: "In this study we aim to provide a picture of the meridional middle atmospheric circulation (better resolved in space and time than that provided by age-of-air based methods.)"

**Action:** Included "(better resolved in space and time than that provided by age-of-air based methods.)"

**Review: 2.1 This discussion of age of air is surprising. What is this so called "traditional observation-based characterization of the circulation"? The authors do not provide a citation.**

**Reply:** The review paper by Waugh and Hall (Rev. Geophys. 40(4), doi10.1029/2000RG000101, 2002) gives an excellent introduction.

**Action:** We have included this reference.

**Review: Some recent work that uses age of air observations to characterize the circulation is Linz et al. 2017. Recent work by Ray et al 2016 combines the TLP with chemistry**

**to examine transport, and the improvement this offers over that method should be addressed. Ray et al. 2010 is also a relevant comparison here.**

**Reply:** Linz et al (10.1038/NGEO30132017, 2017) have characterized the BDC by a single profile. Ray et al. (10.1029/2010JD014206, 2010 and 10.1002/2015JD024447, 2016) introduce the leaky pipe model to explain age of air, ozone, CFCs, and their trends. However, since we do not deal with age of air in this paper, we do not see how these papers are related to our work. The reviewer seems to assume that we are not familiar with the current literature and the approaches used so far. We wish to state that the contrary is true, and our previous work with age of air has made us aware of the weaknesses of this approach and the need to find an improved observation-based access to the BDC.

**Action:** None.

**Review: Furthermore, it is not clear how the method can reveal causes of "discrepancies" between these age and chemical tracer based methods since there are no error estimates.**

**Reply:** The standard deviations representing the precision are by definition smaller than the standard deviations that we show. Thus the standard deviations can serve as upper estimates of the random errors of each monthly field. The uncertainty of the average over the years is accordingly smaller. The rationale behind our claim that our quantities are better suited to determine causes of discrepancies than age of air is fairly trivial. If, say, in the polar stratosphere there is a discrepancy, we still do not know when (since the air entered the stratosphere) or where (along the trajectory of the air parcel) the discrepancy was caused. Our method provides quantities that are resolved latitudinally, vertically and temporally. Thus a much clearer idea can be developed where the model world and the observational world begin to diverge. This is exactly where our method provides an advantage over the age-of-air based methods.

**Action:** see reply to reviewer #1

**Review: 2.1.2 There is no discussion of degrees of freedom. How much independent information is gained by including additional tracers?**

**Reply:** Tests have been shown that inclusion of further species predominantly reduce the error estimates. This effect is seen even in cases where the resulting circulation does not change. For more details, see response to Reviewer #1.

**Action:** see validation paper.

**Review: Specifically, how are sinks included?**

**Reply:** This is described in Section 2.1.2.

**Action:** Some clarifying amendments as requested by reviewer #1 have been included in the description of the sinks.

**Review: 3.1: Plots are very hard to understand. Streamfunctions would be much better.**

**Reply:** We concede that the changing colour scales between the panels of a figure were not optimal. To better serve the needs of the data users, we make the data available in digital form. Then every user can represent the data in their favourite way. Our vector representation offers the advantage that it can directly be compared to the often reproduced schematic by Boenisch et al.

**Action:** Colour scales have been homogenized.

**Review: 3.1.1 How are horizontal transport barriers identified? Why, physically, are they associated with this vertical motion? Is this purely a result of continuity and the fact that this is a 2-D calculation? If so, this should not be referred to as a barrier.**

**Reply:** We have identified horizontal transport barriers as consecutive latitude/altitude bins where the meridional transport velocity is

zero, while meridional transport vectors point in opposite directions in the two meridionally adjacent bins. We agree with Reviewer #1 that this might indeed just identify a splitting or bifurcation and that our wording needed editing.

**Action:** We have rephrased lines 133-134 as specified in reply to reviewer #1.

**Review: 3.1.6 How precisely do you identify that this circulation is associated with the monsoon? Are there particular tracers (e.g. water vapor) that mark this as a monsoon signal? Or is it just about the timing and the fact that it's in the Northern Hemisphere, in which case the link is suggested at best. "Our results show overall agreement with the one shown by Ploeger et al. (2017)[...].";**

**Reply:** Indeed it is the timing, the altitude range and the fact that it appears in the NH only, that we link this to the monsoon.

**Action:** We have edited the main text to make this clearer.

**Review: What "one"?**

**Reply:** Their Fig. 5

**Action:** see reply to reviewer #1.

**Review: What is meant by "overall agreement"?**

**Reply:** We mean agreement related to the structures.

**Action:** We have changed this to "agreement [...] with respect to the circulation structures."

**Review: 3.2.1 337: What is meant by this? How does this reconcile with the well-established water vapor tape recorder results?**

**Reply:** We concede that our wording may lead to misunderstanding.

**Action:** We have reworded this statement.

**Review: 368-374: This seems to be saying that this study is a good validation of the method. That may be, but it's not the stated goal, and more stringent validation is needed especially so as to be able to actually interpret the resulting "effective velocities".**

**Reply:** We think that these indicators of robustness are an important piece of information. They are by no means obsolete, even with a model recovery test in place. The validation will be published in a separate paper.

**Action:** Model recovery tests have been performed and the related paper has been submitted to ACP. The manuscript is be made available to the reviewers.

**Review: 384: What applications would use these effective velocities?**

**Reply:** The Brewer-Dobson circulation explains large ozone amounts over the poles while ozone is predominantly generated at low latitudes. In other words, it uses the effective 2D transport as an explanation of the trace gas distributions in the stratosphere. Thus, the effective 2D transport velocities are the natural measure of the BDC because they directly capture the essence of the BDC.
Further, these effective velocities can be understood as inverse age increments per segment of a mean trajectory. They can thus be related to the age of air but are time-resolved. The effective velocities are an empirical diagnostic in their own right. And in the appendix we relate them to the usual model quantities. Review #2 provides evidence that a significant part of the community regards this quantity as useful.
Regardless if models are able to produce such quantities or not, we think that we can learn a lot from them: Trends, variabilities etc. In this first scientific application paper we have restricted ourselves to climatologies, because these depend less on quantitative validation. The observed transport patterns and their variation contain a wealth of information in a structural sense even if one is sceptical about the associated numbers.

**Action:** Some sentences on possible future work have been included.

[revised manuscript text omitted]

---

## Referee Report (RR1)

OVERVIEW:

The quality of the previous manuscript was hampered by the absence of
any significant validation of the method. This has now been at least
partially rectified by the validation paper, which has shown that the
qualitative features of ANCISTRUS results are at least somewhat
reliable. With that in mind, I believe that this paper can be published
without additional validation work. Before publication, I recommend
that minor revisions be undertaken for clarification of the paper. In
two places ("major" comments L027-029 and L201), some changes in
content are warranted.

I want to provide one important comment to the author with
regards to the validation of their work. The ANCISTRUS recovery tests
performed in the validation paper are, in my view, a partial validation
of the method. Comparing reference and result effective velocities only
shows the effectiveness of the method in reproducing its own results.
This does give credibility to the claim that the ANCISTRUS tool does
not produce spurious structures, which is the most critical requirement
of an inverse method. However, the quantitative claims in the paper are
difficult to verify because no differences between the reference and
result fields were shown. See my comment under "Non-Revision Comments"
for L133-134 for more thoughts on that. I am inclined to believe the
text in your validation paper on those quantitative claims, but as I
cannot verify these results myself, I will not be able to remove my
doubt.

Furthermore, the absence of a comparison between effective velocities
and "standard" quantities remains as a source of doubt on the
interpretation of your results. By standard quantities, I mean the
various residual circulation quantities. A comparison of effective
velocites derived from model tracer fields with the model residual
circulation velocities which produced those tracer fields is, in my
view, a necessary task. Without this, the relationship between
effective velocities and residual velocites will remain unclear, and
any comparison of the two from different sources will be clouded by
the absence of this validation. For example, your results show a
mesospheric overturning circulation and BDC deep branch that are not
clearly separated, but what if mixing is playing a role in the
effective velocities at this location?

That all being said, I think it would be excessive to request a
comparison of effective and residual velocities from model results
be performed now, before publication. Nor should a more explicity
quantitative investigation be requested. However, I think these
absences weaken what would otherwise be an exceptionally strong
contribution to middle atmospheric science.
* * *
MAJOR:

L027-L029: "the true picture of middle-atmospheric circulation is more
detailed and too complicated to be characterized by a scalar intensity
of the circulation." This statement suggests that scalar measures
cannot characterize the stratospheric circulation at all. In my view,
scalar measures are the very first or most basic characterization of
the circulation. I do, however, agree with the notion that scalar
measures cannot capture the details of the circulation (such as the
activity of different pathways), and that progress towards
understanding the stratospheric circulation will require understanding
these details. I suggest changing this statement to focus on the
insufficiency of scalar measures to characterize the details (nuances,
pathways, etc., I think a lot of terminology would fit here) of the
circulation, or on the insufficiency of scalar measures to provide a
complete description of the circulation.

L039-L040: A comma or hyphen is necessary between "overturing
circuation" and "which brings" and between "ozone chemistry" and
"has been". This is necessary to provide a clear boundary on the aside

about NOx rich air and stratospheric ozone chemistry.

L047-L050: The meaning of this sentence ("While the...") is not clear to me. Are you suggesting that the comparison of modelled trace gas fields with observed fields is used to estimate the stratospheric circulation? I can't think of any true, published attempts at that. Is that what you mean? It seems like this sentence needs rewriting. Also, in relation to age of air methods you could cite the work of Fritsch et al. (Fritsch, F., Garny, H., Engel, A., Bönisch, H., and Eichinger, R.: Sensitivity of age of air trends to the derivation method for non-linear increasing inert SF6, Atmos. Chem. Phys., 20, 8709–8725, https://doi.org/10.5194/acp-20-8709-2020, 2020), who demonstrate some difficulties of using age of air methods (in particular the AirCore-derived results of Engel et al.).

L050-L052: This sentence ("Our results...") seems to state the same information as the following sentence. Furthermore, the mention of "our results" comes before any mention of the work of this study, which is an atypical form. I do not see any point to this sentence, nor to the previous sentence. I suppose the goal of this paragraph was to clearly establish the need for your work (and there is certainly need) but I think this has not been effectively communicated.

L201: You write here about air mass transport. I agree with the basic principle (weaker velocities at lower levels may easily transport more air mass than stronger velocities at higher levels). I have some concern, however, that the effective velocities may not correspond directly to air mass transport. In my view, noting the possiblity of a discrepancy in this regard is important.

L216-217: You write here about the stratopause location. Would it be possible to include estimates (even approximate) of the stratopause and tropopause locations on your figures? I'm not sure what data source would be most appropriate for that information. Even a long-term monthly-resolved climatology would be very helpful for the reader to interpret the results. In my view, this would be very helpful, but not necessary.

Figure 1: I think you should consider using streamfunctions to visualize the velocity field. The figures as they are do somewhat show the qualitative information, particularly for stronger velocities, but the quantitative information is somewhat difficult to interpret. In my view, steamfunctions would be a more effective and familiar quantity for the visualization of this qualities of the effective circulation.

Figure 1 caption: You mention "non-converged inversions" here, but I do not see that mentioned anywhere else in the text. In my view, that should be addressed in the main text somewhere.

Figure 5: I like these figures, but I think you only quantitatively reference variability in comparison with local velocities. Due to that, you should consider showing variability relative to the local monthly-mean velocity. In my view, this isn't necessary, but it has the potential to assist the reader in interpretation greatly.

L496-499: The region you describe here sounds a lot like the startospheric surf zones, but that's not mentioned in the text anywhere. You might consider making a mention of that here, if you also agree that the variability in velocities that you've found in the region could be related to surf zone activity.

L525: Mentioning funding is not really appropriate for a peer-reviewed publication. This should be removed and some other introduction to the sentence should take its place.
* * *
MINOR:

L055: Probably you should say "effective circulation vectors" right

away, as opposed to leaving the information about the "effective"
nature of your results for later.

L083: "than with the age-based method" suggests that there is only
one age-based method, which I think is not the case, so this should
be "than with age-based methods".

L086-087: Here you write the name of the method, but that should
probably be written the first time the method is mentioned, which is
earlier in this section.

L089: "future tracer gas" would be more precise. Or "subsequent".

L117: Can you estimate these inaccuracies?

L118: You should either provide some quantification of this "minor
relevance" or at least provide some citation for that information.

L124: Do you mean SF6 sinks? It's not clear to me. That should be
clarified.

L179: About the words "allows to better resolve", because resolution is
something you discuss as an advantage of the method, "resolve" seems to
suggest that there is some difference in the method here. I suggest
replacing this with the phrase "allow easier interpretation of".

L181: Saying "inter-annual averages" or "climatological averages" might
be helpful for the reader.

L190-191: Was this uncertainty quantified?

L208: About "signal of subsidence": do you mean in Figure 6? If so,
that should be specified ("signal of subsidence in the inter-annual
variability"). If the velocity figures are indicated instead, I think
this should be expanded upon, as there are certainly some cases where
subsidence seems to occur, in particular november-december of Figure 4.

L217: New paragraph at "Most parts".

Figure 1 caption: You mention missing species are indicated in the
headers, but I don't see any species indicated.

L298: "there" is not very precise. I think "present" would be more
precise.

L343: The word "it" could refer to multiple entities. It would be
better to state this explicitly.

L416-417: "Figures 7-8 (middle right panels)" or "the righthand middle
panels of Figures 7 and 8"

L444: "large interannual variability is expected based on current
theory" would be more precise

L444-L446: About "The stability...", I don't see how this sentence
helps. As far as I can tell, the flow of the paragraph would be better
without this sentence and the meaning of the paragraph would not
change. The next sentence is much more to-the-point anyway.

L461: "From our results" is better, as you just mentioned MIPAS "data".

L494: I don't understand what "this" is referring to, or what this part
of the sentence (everything after "km," means).

L510: "common" is confusing here because it seems to leave the
possibility that you use an uncommon a priori distribution to nudge
your method.

L511: After "patterns." might be a good place to add a sentence briefly

describing the iterative nature of the method. "An initial velocity distribution was used to begin the interative inversion calculation, but the choice of this initial field does not have significant effects on the resulting fields" or something like that.

L515: I think what you mean by "features" is "novel results". Of course there are very many more features, but these are certainly the most interesting ones of your results. They are very interesting, by the way.

L520: "The particular figure quoted" is somewhat strange. I recommend removing the citation in the previous sentence and replacing this text with "For example, the schematic of Bönisch et al 2011 (their Figure 1)" or something similar.
* * *
TYPOS:

L006: "THE stratospheric circulation is found to be"

L018: "and is called THE 'Brewer-Dobson circulation' "

L027: "the true picture of THE middle-atmospheric"

L090: Comma after "coefficients"

L098: 'field' not 'fields'

L102: 'is started', not 'ist'

L102: 'final' not 'finally resulting'

L104: "Since inferred velocities, due to the correlation of velocities and atmospheric composition, are not the zonally-averaged velocities but include eddy transport effects, we call the inferred velocities 'effective velocities'."

L109: Comma after $H_2O$.

L110: Comma after "band". These commas, and the ones I mentioned earlier, are called "Oxford commas" if you want to look that up. It's a practice used to avoid confusion in lists.

L112: "photolysis"

L112: "and" instead of ", as well as"

L115: Probably you want to say something like "equilibrium assumption", but I know this as the "steady-state assumption".

L119: Comma after CO.

Figure 1 caption: ", the months" should be ", and the months"

L459: "pole-to-pole"

L467: "THE NH atmospheric circulation"

L490: "transport pattern"

L501: "broadly reproduces well" doesn't make sense, just say "broadly reproduces"

L502: "but" instead of "however"

L506: remove "of air sampling instruments" as it's not necessary

L507: "the sense" not "a sense"

L518: Comma after e.g.

L522: "future steps" should read "future steps for this work"

L523: "analysis"

L523: remove "the" in front of "interannual"
* * *
NON-REVISION COMMENTS (I.E. NO CHANGE SUGGESTED):

L133-134: I found it difficult to verify this claim in the validation
paper. The validation paper does show the two (reference and result, if
you will) velocity fields, but does not display the differences between
them as far as I can tell. Because the claims of the present paper do
not depend on quantitative information, there is no need to establish
this point further. However, I think you should be aware of this in
future work. What would help is a simple depiction of the reference
velocities, the result velocites, and the differences between them,
alll next to each other. That would make interpretation rather easy for
readers. Again, I do not think this is necessary for this paper, but
please consider this for future work.

L525: In my view, the distinction of transport and mixing is absolutely
the most important future step for this work. It is still not clear to
me what aspects of your results are due to the inclusion of mixing, and
this brings me to view the results with some uncertainty. It would also
be very, very cool to have estimates of mixing in the resolution that
your results have.

---

## Author Response (AR3)

The authors thank the reviewers for their thorough, detailed, and insightful reviews. They have helped to improve the paper.

**Review #1:**

**OVERVIEW: The quality of the previous manuscript was hampered by the absence of any significant validation of the method. This has now been at least partially rectified by the validation paper, which has shown that the qualitative features of ANCISTRUS results are at least somewhat reliable. With that in mind, I believe that this paper can be published without additional validation work. Before publication, I recommend that minor revisions be undertaken for clarification of the paper. In two places ("major" comments L027-029 and L201), some changes in content are warranted.**

**Reply:** We thank the reviewer for this encouraging evaluation.

**Action:** See in the detailed list of comments.

**I want to provide one important comment to the author with regards to the validation of their work. The ANCISTRUS recovery tests performed in the validation paper are, in my view, a partial validation of the method. Comparing reference and result effective velocities only shows the effectiveness of the method in reproducing its own results. This does give credibility to the claim that the ANCISTRUS tool does not produce spurious structures, which is the most critical requirement of an inverse method. However, the quantitative claims in the paper are difficult to verify because no differences between the reference and result fields were shown. See my comment under "Non-Revision Comments" for L133-134 for more thoughts on that. I am inclined to believe the text in your validation paper on those quantitative claims, but as I cannot verify these results myself, I will not be able to remove my doubt.**

**Reply:** Figs 8-10 in the validation paper ARE difference plots. Beyond this, for reasons of traceability, all quantitative results of the tests presented in the validation paper have been published in the KITopen repository.

**Action:** We have now inserted a reference to the data file.

**Furthermore, the absence of a comparison between effective velocities and "standard" quantities remains as a source of doubt on the interpretation of your results. By standard quantities, I mean the various residual circulation quantities. A comparison of effective velocites derived from model tracer fields with the model residual circulation velocities which produced those tracer fields is, in my view, a necessary task. Without this, the relationship between effective velocities and residual velocites will remain unclear, and any comparison of the two from different sources will be clouded by the absence of this validation. For example, your results show a mesospheric overturning circulation and BDC deep branch that are not clearly separated, but what if mixing is playing a role in the effective velocities at this location? That all being said, I think it would be excessive to request a comparison of effective and residual velocities from model results be performed now, before publication. Nor should a more explicity quantitative investigation be requested. However, I think these absences weaken what would otherwise be an exceptionally strong contribution to middle atmospheric science.**

**Reply:** We have studies to disentangle transport and mixing processes on our agenda, but this is a full project in its own right and exceeds by far what we can be done within the framework of this study.

**Action:** None to this point.

**MAJOR:**
**L027-L029: "the true picture of middle-atmospheric circulation is more detailed and too complicated to be characterized by a scalar intensity of the circulation." This state-**

ment suggests that scalar measures cannot characterize the stratospheric circulation at all. In my view, scalar measures are the very first or most basic characterization of the circulation. I do, however, agree with the notion that scalar measures cannot capture the details of the circulation (such as the activity of different pathways), and that progress towards understanding the stratospheric circulation will require understanding these details. I suggest changing this statement to focus on the insufficiency of scalar measures to characterize the details (nuances, pathways, etc., I think a lot of terminology would fit here) of the circulation, or on the insufficiency of scalar measures to provide a complete description of the circulation.

**Reply:** Agreed.

**Action:** Changed to "the true picture of middle-atmospheric circulation is more detailed and too complicated to be FULLY characterized by a scalar intensity of the circulation."

**L039-L040:** A comma or hyphen is necessary between "overturing circuation" and "which brings" and between "ozone chemistry" and "has been". This is necessary to provide a clear boundary on the aside about NOx rich air and stratospheric ozone chemistry.

**Reply:** Agreed.

**Action:** Corrected as suggested.

**L047-L050:** The meaning of this sentence ("While the...") is not clear to me. Are you suggesting that the comparison of modelled trace gas fields with observed fields is used to estimate the stratospheric circulation? I can't think of any true, published attempts at that. Is that what you mean? It seems like this sentence needs rewriting. Also, in relation to age of air methods you could cite the work of Fritsch et al. (Fritsch, F., Garny, H., Engel, A., Bnisch, H., and Eichinger, R.: Sensitivity of age of air trends to the derivation method for non-linear increasing inert SF6, Atmos. Chem. Phys., 20, 87098725, https:// doi.org/10.5194/acp-20-8709-2020, 2020), who demonstrate some difficulties of using age of air methods (in particular the AirCore-derived results of Engel et al.).

**Reply:** The "While the..." sentence does not presume that stratospheric circulation is inferred from such kind of comparison. Instead, this sentence is meant as a justification why some kind of inverse method is needed.

**Action:** The "While the..." sentence has been rewritten: "The direct comparison of modelled trace gas fields with measured ones is very unspecific with respect to causes of discrepancies, because it reveals only the consequences of any deficiency in the model but provides no direct clue how the discrepancies came about." The suggested reference has been included. Also, references covering the mesospheric sink issue have been included.

**L050-L052: This sentence ("Our results...") seems to state the same information as the following sentence. Furthermore, the mention of "our results" comes before any mention of the work of this study, which is an a typical form. I do not see any point to this sentence, nor to the previous sentence. I suppose the goal of this paragraph was to clearly establish the need for your work (and there is certainly need) but I think this has not been effectively communicated.**

**Reply:** Agreed. We have reorganized these paragraphs in a way that the former "While the..." paragraph describes only the issues with the existing methods, without mentioning our method and results. These are now mentioned only in the following paragraph. With this, we have removed the redundancy.

**Action:** We have deleted this sentence and added instead "Beyond this, age- of-air based methods integrate over the time an air parcel spent in the stratosphere and thus provide information on the middle atmospheric circulation only at quite limited temporal and

spatial resolution."

**L201: You write here about air mass transport. I agree with the basic principle (weaker velocities at lower levels may easily transport more air mass than stronger velocities at higher levels). I have some concern, however, that the effective velocities may not correspond directly to air mass transport. In my view, noting the possiblity of a discrepancy in this regard is important.**

**Reply:** Agreed.

**Action:** We have replaced 'velocities' with 'effective velocities'. Further, we have reworded the second part of the statement such that it does no longer claim that transport, driven by real velocities, is actually the driving mechanism. "While its EFFECTIVE velocities and vertical extension are smaller, due to the larger air density at these lower altitudes smaller velocities still can transport considerable airmass to higher latitudes."

**L216-217: You write here about the stratopause location. Would it be possible to include estimates (even approximate) of the stratopause and tropopause locations on your figures? I'm not sure what data source would be most appropriate for that information. Even a long-term monthly-resolved climatology would be very helpful for the reader to interpret the results. In my view, this would be very helpful, but not necessary.**

**Reply:** We are reluctant to do this, because average tropopause (or stratopause) altitudes along with average transport paths can be extremely misleading. Such figures can easily let a tropospheric process look like a stratospheric one or vice versa.

**Action:** None.

**Figure 1: I think you should consider using streamfunctions to visualize the velocity field. The figures as they are do somewhat show the qualitative information, particularly**

for stronger velocities, but the quantitative information is somewhat difficult to interpret. In my view, steamfunctions would be a more effective and familiar quantity for the visualization of this qualities of the effective circulation.

**Reply:** Since, after the long time of the reviewing process, key scientists are no longer available and meanwhile work on other projects, we are not in a position to apply major changes to the visualization software.

**Action:** We have made the original data available via KITopen to allow the readers to visualize the data in their preferred way. A link to the original data (with doi) is provided.

**Figure 1 caption: You mention "non-converged inversions" here, but I do not see that mentioned anywhere else in the text. In my view, that should be addressed in the main text somewhere.**

**Reply:** Agreed.

**Action:** At the end of the first paragraph of the "Results" section we have added: "The years which went into the mean fields are indicated. Missing years are chiefly attributed to MIPAS data gaps. Only in a few cases (November- December 2003, March-April 2007, December-January 2009, March-April 2009, and June-July 2011) the inversion did not converge or another technical problem was encountered."

**Figure 5: I like these figures, but I think you only quantitatively reference variability in comparison with local velocities. Due to that, you should consider showing variability relative to the local monthly-mean velocity. In my view, this isn't necessary, but it has the potential to assist the reader in interpretation greatly.**

**Reply:** We have tried this, but there are many places where the effective velocities are more or less zero. In a relative or percentage representation these areas dominate the picture with close to infinite

relative variabilities, and the variabilities in the physically interesting regions are no longer resolved by the colour scale. Thus we have decided against the relative representation. We see the intention to make small but significant velocities in the lower stratosphere better visible, but actually this type of plot is dominated by areas where nothing is going on.

**Action:** None.

**L496-499: The region you describe here sounds a lot like the startospheric surf zones, but that's not mentioned in the text anywhere. You might consider making a mention of that here, if you also agree that the variability in velocities that you've found in the region could be related to surf zone activity.**

**Reply:** This is a good point.

**Action:** We have added: "This region coincides with that where planetary-wave breaking is observed, which was first observed by McIntyre and Palmer (1984) and to which they named the stratospheric 'surf zone'."

**L525: Mentioning funding is not really appropriate for a peer-reviewed publication. This should be removed and some other introduction to the sentence should take its place.**

**Reply:** Although we have enjoyed writing this, we do agree.

**Action:** Changed to: "More ambituous researchers may even plan an ANCISTRUS model in other than ..."

**MINOR: L055: Probably you should say "effective circulation vectors" right away, as opposed to leaving the information about the "effective" nature of your results for later.**

**L083: "than with the age-based method" suggests that there is only one age- based method, which I think is not**

the case, so this should be "than with age- based methods".

**Reply:** Agreed.

**Action:** Changed as suggested.

**L086-087: Here you write the name of the method, but that should probably be written the first time the method is mentioned, which is earlier in this section.**

**Reply:** Agreed.

**Action:** We now introduce the name of the method in the first sentence of this section.

**L089: "future tracer gas" would be more precise. Or "subsequent".**

**Reply:** agreed.

**Action:** 'later' replaced by 'subsequent'.

**L117: Can you estimate these inaccuracies?**

**Reply:** This is hardly possible because for the underlying model calculations no uncertainty estimates are available either.

**Action:** None

**L118: You should either provide some quantification of this "minor relevance" or at least provide some citation for that information.**

**Reply:** We concede that this statement was too strong; we replace it with a weaker one.

**Action:** The statement has been replaced by "Inaccuracies in the latter estimates are deemed tolerable since the related loss reaction is only one of three relevant stratospheric loss reactions (Brasseur

and Solomon, 2005)."

**L124: Do you mean SF6 sinks? It's not clear to me. That should beclarified.**

**Reply:** The Stiller et al. reference where this problem is discussed refers to $SF_6$ but the statement why these sinks are not important in our case is valid for all gases.

**Action:** Inserted "Since FOR ALL SPECIES UNDER CONSIDERATION values at the upper boundary are prescribed using..."

**L179: About the words "allows to better resolve", because resolution is something you discuss as an advantage of the method, "resolve" seems to suggest that there is some difference in the method here. I suggest replacing this with the phrase "allow easier interpretation of".**

**Reply:** We do agree that the term "resolve", that is used as a technical term elsewhere in the paper should not be used in a common language sense here. However, we doubt that "interpretation" is optimal, because this term goes beyond the mere perception.

**Action:** We have replaced "resolve" with "discern"

**L181: Saying "inter-annual averages" or "climatological averages" might be helpful for the reader.**

**Reply:** Agreed but in this context we prefer "multi-annual" over "inter-annual".

**Action:** added "" ... are built from MULTI-ANNUAL averages, ...

**L190-191: Was this uncertainty quantified?**

**Reply:** This uncertainty is routinely quantified but not reported in this context, because for the interpretation of multi-annual averages requires the total uncertainty due to measurement errors and natural variability to be considered.

**Action:** None

**L208: About "signal of subsidence": do you mean in Figure 6? If so, that should be specified ("signal of subsidence in the multi-annual variability"). If the velocity figures are indicated instead, I think this should be expanded upon, as there are certainly some cases where subsidence seems to occur, in particular november-december of Figure 4.**

**Reply:** Here we refer again to Figure 1. We do not mean that these subsidence effects always average out perfectly but that they can cancel out.

**Action:** We have inserted another reference to Fig. 1 after "...no clear signal of subsidence." Further, we have modified the next sentence: "which OFTEN causes subsidence effects to be averaged out when latitudinal averages are considered"

**L217: New paragraph at "Most parts". Figure 1 caption: You mention missing species are indicated in theheaders, but I don't see any species indicated.**

**Reply:** Yes, indeed. This text seems to refer to older versions of the plots.

**Action:** The obsolete sentence has been deleted.

**L298: "there" is not very precise. I think "present" would be more precise.**

**Reply:** Agreed.

**Action:** Changed as suggested.

**L343: The word "it" could refer to multiple entities. It would be better to state this explicitly.**

**Reply:** Agreed.

**Action:** "it" replace by "this branch".

**L416-417: "Figures 7-8 (middle right panels)" or "the right hand middle"**

**Reply:** Agreed

**Action:** "middle right panels" now in parentheses.

**L444: "large inter-annual variability is expected based on current theory" would be more precise**

**Reply:** Agreed.

**Action:** Changed as suggested.

**L444-L446: About "The stability...", I don't see how this sentence helps. As far as I can tell, the flow of the paragraph would be better without this sentence and the meaning of the paragraph would not change. The next sentence is much more to-the-point anyway.**

**Reply:** Agreed.

**Action:** Sentence deleted.

**L461: "From our results" is better, as you just mentioned MIPAS "data".**

**Reply:** Agreed.

**Action:** Changed as suggested.

**L494: I don't understand what "this" is referring to, or what this part of the sentence (everything after "km," means).**

**Reply:** The respective subclose is indeed confusing rather than

clarifying.

**Action:** The subclause "as this is also ... can reach" has been deleted.

**L510: "common" is confusing here because it seems to leave the possibility that you use an uncommon a priori distribution to nudge your method.**

**Reply:** Agreed. The prior we use is zero throughout. Thus, all signal we see is from the data. We have reworded this to remove the detected ambiguity.

**Action:** The sentence has been rewritten: "This behaviour cannot be attributed to the use of any a priori velocities that would push the results towards the expected circulation patterns. On the contrary, our a priori effective velocities are zero throughout, which guarantees that all structures seen in the results are produced by the measured trace gas contributions."

**L511: After "patterns." might be a good place to add a sentence briefly describing the iterative nature of the method. "An initial velocity distribution was used to begin the interative inversion calculation, but the choice of this initial field does not have significant effects on the resulting fields" or something like that.**

**Reply:** Agreed.

**Action:** Added: "The zero a priori field is also used as initial guess of the iterative inversion, but its only effect on the results is a certain smoothing of the retrieved structures (von Clarmann and Grabowski, 2021). Another sign of the stability of our method is that the transitions between the circulation... patterns of subsequent months"

**L515: I think what you mean by "features" is "novel results". Of course there are very many more features, but these are certainly the most interesting ones of your re-**

sults. They are very interesting, by the way.

**Reply:** Agreed. We are happy about the appreciation of these results.

**Action:** Changed as suggested.

**L520:** "The particular figure quoted" is somewhat strange. I recommend removing the citation in the previous sentence and replacing this text with "For example, the schematic of Bnisch et al 2011 (their Figure 1)" or something similar.

**Reply:** Agreed.

**Action:** Changed as suggested.

**Typos etc. L006:** "THE stratospheric circulation is found to be"
**L018:** "and is called THE 'Brewer-Dobson circulation' "
**L027:** "the true picture of THE middle-atmospheric"
**L090:** Comma after "coefficients"
**L098:** 'field' not 'fields'
**L102:** 'is started', not 'ist'
**L102:** 'final' not 'finally resulting'
**L104:** "Since inferred velocities, due to the correlation of velocities and atmospheric composition, are not the zonally-averaged velocities but include eddy transport effects, we call the inferred velocities'effective velocities'."
**L109:** Comma after H2O.
**L110:** Comma after "band". These commas, and the ones I mentioned earlier, are called "Oxford commas" if you want to look that up. It's a practice used to avoid confusion in lists.
**L112:** 'photolysis"
**L112:** "and" instead of ", as well as"
**L115:** Probably you want to say something like "equilibrium assumption",but I know this as the "steady-state assumption".
**L119:** Comma after CO.

**Figure 1 caption:** ", the months" should be ", and the months"
**L459:** "pole-to-pole"
**L467:** "THE NH atmospheric circulation"
**L490:** "transport pattern"
**L501:** "broadly reproduces well" doesn't make sense, just say "broadly reproduces"
**L502:** "but" instead of "however"

**Reply:** We thank the reviewer for their careful reading of our manuscript.

**Action:** All corrections suggested so far have been implemented.

**L506: remove "of air sampling instruments" as it's not necessary**

**Reply:** Due to a typo ("of" instead of "or") the meaning of our statement was distorted. With this typo-correction in place, we think it does make sense to mention the air sampling instruments.

**Action:** Typo corrected.

**L507: "the sense" not "a sense"**

**Reply:** Agreed.

**Action:** Corrected.

**L518: Comma after e.g.**

**Reply:** This correction has become obsolete after rewriting as described above.

**Action:** See above.

**L522: "future steps" should read "future steps for this work"**
**L523: "analysis"**

**L523: remove "the" in front of "interannual"**

**Reply:** Agreed.

**Action:** These three corrections have been implemented as suggested.

**NON-REVISION COMMENTS (I.E. NO CHANGE SUGGESTED):**
**L133-134: I found it difficult to verify this claim in the validation paper. The validation paper does show the two (reference and result, if you will) velocity fields, but does not display the differences between them as far as I can tell. Because the claims of the present paper do not depend on quantitative information, there is no need to establish this point further. However, I think you should be aware of this in future work. What would help is a simple depiction of the reference velocities, the result velocites, and the differences between them, all next to each other. That would make interpretation rather easy for readers. Again, I do not think this is necessary for this paper, but please consider this for future work.**

**Reply:** The differences **are** shown in the lower panels of the relevant figures of the validation paper.

**Action:** None

**L525: In my view, the distinction of transport and mixing is absolutely the most important future step for this work. It is still not clear to me what aspects of your results are due to the inclusion of mixing, and this brings me to view the results with some uncertainty. It would also be very, very cool to have estimates of mixing in the resolution that your results have.**

**Reply:** We agree that this is interesting, and we have it on our agenda. However, our primary intention has been a refinement of the age-of-air approach. Indeed our results can be interpreted as

inverse incremental age differences. And the age-of-air concept does not disentangle transport and mixing either.

**Action:** None for this paper.

**Review #2:**

This paper uses MIPAS satellite observations of 9 long-lived trace gases, with various lifetimes, in an inversion model to calculate the effective mean vertical and meridional transport velocities over the 2002-2012 period. The inversion model was validated in a separate publication. This method provides interesting results that show two dimensional mapping (latitude-height) of the seasonality of the middle atmospheric circulation and mixing. The standard deviations of vertical and meridional transport velocities, also mapped, identify where transport is most variable. These results are important because they are observationally-based and they provide a unique spatially and temporally resolved quantitative analysis of transport in the stratosphere and mesosphere the entire middle atmosphere. As important and interesting as these results are, the paper needs a number of revisions before it should be published. Below you will find comments organized by the topic areas in need of revision. Kudos to the authors for the Supplemental figures with an alternative color scheme! That's a very considerate touch and please do not hide this information in a footnote; state this in the main text.

**Reply:** We thank the reviewer for the encouraging evaluation of the paper.

**Action:** The hint at the alternative versions of the figures has been moved to the main text as suggested.

**Topic areas requiring Revision**

**Climatology.** The authors note in a footnote that 10 years is not a climatologically relevant period. I strongly agree,

so please do not call it one; there is an alternative. MIPAS has measured the atmosphere for 1 decade, so I recommend framing this paper as a BDC analysis of the period 2002-2012. Thats factually what this is, but by calling it that you can set up the idea that your analysis provides a basis for comparison in future studies. (Such a future study with Aura MLS data is even called out in the conclusions.) This will be helpful for examining the question of whether the middle atmospheric transport is changing. You are probably aware that this is of enormous interest to many, including chemistry climate modelers who are predicting an acceleration of the stratospheric circulation this century (e.g., Polvani et al. 2019, JGR), while observations show a different and asymmetric response (Strahan et al., 2020 GRL). This paper can be an important part of the answers we need. My recommendation is that you remove climatology everywhere it occurs and rewrite as a 10-yr mean or analysis. Because climatology appears in the title, this too needs change, perhaps: The Middle Atmospheric Meridional Circulation for 2002-2012 derived from MIPAS observations

**Reply:** Agreed.

**Action:**

- Title changed as suggested, and short title changed in the same spirit.

- "Monthly climatologies of" changed to "Multi-annual monthly mean"

- "From these we calculate a climatology of the circulation in terms of multi-annual monthly means" changed to "From these we calculate multi-annual monthly mean circulations ...". The now obsolete footnote has been deleted.

- "and our scheme to calculate climatologies from the monthly circulation patterns" replaced by "our scheme to calculate multiannual monthly mean circulation fields from the individual monthly circulation fields"

- "Our derived climatologies of middle atmospheric circulation" changed to "Our derived multi-annual monthly mean circulation fields are discussed".

- "The resulting circulation fields are analyzed in terms of first and second moment statistics to provide a climatology of the middle atmospheric circulation." replaced by "The resulting multi-annual monthly mean circulation fields are analyzed in terms of first and second moment statistics.".

- "The Climatology of middle atmospheric meridional circulation" replaced with "The multi-annual monthly mean middle atmospheric meridional circulation".

- "to form the 12-monthly climatology" changed to "to form the 12-monthly data set"

- "appears weaker in these climatologies" changed to "appears weaker in these multi-annual monthly means"

- "new climatology of middle atmospheric circulation fields" changed to "data-set of multi-annual monthly mean middle atmospheric circulation fields"

- "The climatologies..." changed to "These circulation fields..."

- "...in the climatology." changed to "in the multi-annual monthly mean."

- "from this new climatology" changed to "from this new data set"

- "seen in these climatology fields" changed to "seen in these multi-annual monthly averages"

**Attributing all variability to SSWs - wheres the discussion of the QBO? There is no mention of the Quasi Biennial Oscillation (QBO) anywhere in this manuscript, yet the QBO is the largest driver of stratospheric variability after the annual cycle. For example, around line 205, NH winter polar variability is attributed (without proof) to SSWs. Sure,**

**SSWs certainly cause NH high latitude variability in winter, but any discussion of the cause also requires a discussion of the QBO. The QBO modulates SSW occurrences. The QBO exists in the stratosphere and mesosphere, and affects both tropical and extratropical latitudes. Baldwin et al. (2001, Rev. Geophys.) provides a great overview, with discussions of the QBOs impact on the meridional circulation and chemical constituents. The solution is either to talk about the QBOs influence on variability, or to keep Section 3.1 completely descriptive no attribution of features without an analysis or suitable citation. This applies to all of Section 3.**

**Reply:** We partly agree. We agree that the QBO should be mentioned as a driver of variability. However, the analysis of multi-annual monthly means, where the annual cycle is resolved, while periodic or quasi-periodic variations of longer time-scales are not resolved, is a less than optimal framework for the analysis of QBO effects. We thus defer this to a study which will be based in the full time-series rather than multi-annual monthly means. Beyond this, many of our statements related to the large variability of effective velocities in polar regions refer to altitudes above 50 km, and the knowledge of the causal chain between stratospheric tropical winds and mesospheric dynamics still seems to have some gaps. We are not even sure if the direction of the causal arrow is known with certainty. While in the revised version of the manuscript we consider the QBO as a driver of variability, we take care not to over-exaggerate this issue. Relevant parts of the manuscript are:

- Abstract: We agree that both SSWs and QBO should be mentioned.

- Intro, par. 4: We think in this context it is correct to limit the discussion to SSWs.

- Section 3.1.1 (January-February): The old text referred to the maximum of variability above 50 km. We do agree that the secondary maximum at about 30 km deserves to be discussed, too.

- Section 3.1.12 (December-January): Agreed to mention the QBO here.

- Section 3.2.3 (Variable Phenomena): We agree to mention the QBO as another important driver of variability.

- Section 4. Discussion: Agreed to mention the QBO here.

**Action:**

- Abstract: "Sudden stratospheric warmings cause increased year-to-year variability of the vertical component of the circulation." replaced by "Sudden stratospheric warmings and the quasi-biennial oscillation cause a pronounced year-to-year variability of the meridional circulation."

- Intro, par. 4: No action.

- Section 3.1.1 (January-February): The related part has been rewritten: "Above 50 km at Northern polar latitudes there is some subsidence. Associated year-to-year variability in vertical effective velocities is large, reflecting the irregular appearance of sudden stratospheric warmings (Fig. 6, upper left panel). Their irregular occurance and the related impact on subsidence is discussed, e.g., in Funke et al. (2014). Large variability over the North pole at stratospheric altitudes does not come unexpected, since Haenel et al. (2015, see their Fig. 9) found in their age-of-air time series analysis largest amplitudes of the signal of the quasi-biennial oscillation (QBO) at polar latitudes. Strahan 2020 et al. (and references therein) highlight the importance of the QBO for stratospheric circulation. Baldwin et al. (2001, 2021) also discuss the possible interaction between the QBO and sudden stratospheric warmings and mention mesopheric QBO effects."

- Section 3.1.12 (December-January): Inserted: "As discussed in Section 3,1,1, the QBO is another important driver of the interannual variability of circulation.

- Section 3.2.3 (Variable Phenomena): Inserted: "[between 25 and 30 km,] which is further enhanced by an interaction between the QBO and vortex dynamics (e.g., Strahan et al., 2015)."

- Section 4. Discussion: Inserted at the end of the 1st paragraph: "The QBO is another driver of stratospheric variability, and

Haenel et al. (2015) found that the the contribution of the QBO to the explanation of age-of-air time series is largest in the polar stratosphere."

**Similarly, at lines 357 and 381 a feature is attributed to the Asian monsoon circulation. Please give proof or cite an appropriate paper.**

**Reply:** We have seen the Monsoon effect in time-resolved longitudinally resolved MIPAS data. We now make reference to the relevant work.

**Action:** Inserted: "A clear monsoon signal is visible in MIPAS data resolved in time and longitude (See, e.g., Vogel et al., 2019), and is obviously strong enough to survive zonal averaging."

**At line 384 (and somewhere earlier in the paper), the equinoctial mesospheric pole to pole transport toward the winter pole is mentioned. This is a is well known feature and should be referenced somewhere.**

**Reply:** We do not understand this. Which pole is, at equinox, the winter pole? This statement looks somewhat like an oxymoron. We take the reviewer to be speaking about the solstice pole-to-pole circulation, that is indeed well known, but this is not what we are discussing here.

**Action:** None.

**In the Conclusions, lines 512-514 incorrectly attribute variability to SSWs. They are not the only phenomenon driving large interannual variability (IAV) in the winter hemisphere - the QBO does too. In fact, the SH shows large IAV yet has no SSWs. See Strahan et al (2015, GRL) for an example of observed large IAV in the SH in the 25-30 km range and an explanation for the QBOs effect on trace gases in Ploeger & Birner (2016, GRL). Recently large amplitude extratropical variability with a 5-7 yr period was identified in observations that is likely driven by the QBO (Strahan et al 2020, GRL).**

**Reply:** We find larger variability in the northern polar winter atmosphere, and we explicitly attribute this NH variability to the SSWs. In addition we now mention the QBO as a driver of variability. Besides the Strahan et al (2015) reference we add also the reference to Haenel (2015) et al.

**Action:** We have added: "The QBO is another driver of stratospheric variability (see, e.g., Strahan et al., 2015,2020), and Haenel et al. (2015) found that the contribution of the QBO to the explanation of age-of-air time series is largest in the polar stratosphere."

**Improvements to Figures. Consider that the data for each month for altitudes below 30 km are actually included in 4 different figure panels! Currently each figure type requires 2 pages of 6-panel figures. Reduce redundancy by using 6 bi-monthly averages (e.g., Jan-Feb, Mar-Apr, etc.) and then the annual cycle for a given variable, such as effective meridional velocity, can easily be viewed and an understood from a single figure. This cuts the number of figures by 2.**

**Reply:** The increase of the number of figures was in reply to the first review of reviewer #1. We do not see a lot of redundancy. Bi-monthly averaging would remove important information. Similarly, a lot of information would be lost if only annual cycles of certain selected variables were presented. We would no longer see the interesting processes like the subsidence of the deep branch of the BDC over the winter, or the connection between the deep branch and the overturning circulation. It is the spatial and temporal resolution which makes our results special, and we are reluctant to hide this information by averaging or picking out single locations for time series. If we pick out one latitude/altitude bin and show the seasonal variation, we totally miss the relation to neighbouring bins, i.e. the phase shift that occurs by the subsidence of the deep BDC branch.

**Action:** None.

**For figures 5 and 6, the standard deviations would provide more information if each were divided by its mean value.**

**Currently, because means and std devs are so much smaller at lower altitudes, these figures tell us very little about the lower and middle stratosphere. Normalizing the std devs by the mean (i.e., to show fractional or percentage deviations) fixes this and allows the reader to easily identify enhancements in variability at all altitudes.**

**Reply:** We have tried this, but these figures were not useful. Huge percentage errors where velocities are almost zero masked all useful information.

**Action:** None.

**Id like to suggest a new figure that sums up how the transport processes vary spatially and seasonally. Identify the month where each point of the velocity field (or its std dev) maximizes. Then make a contour plot that shows how the timing of these maxima varies pole to pole, 6-68 km. This might make a really interesting 4-panel figure (vel meridional, vel vertical, and each of their std devs). At a glance one could grasp the timing and location of where and when transports processes are the greatest. (Conversely you could try this for the minimum of the variable.) In addition, this would also show the relationship between vertical and horizontal transport processes (easier to see because all the panels would be in the same figure).**

**Reply:** Maybe this would work as an additional figure but cannot replace the existing figures. The velocities are so different over altitude that the lower part of the stratosphere would be totally ignored (the largest velocities are always in the upper stratosphere and mesosphere). And they are not necessarily from the same transport branch. Look, for example, at the two very first panels: Fig. 1, J/F and F/M: Here the maximum meridional velocity would jump from 20N/45km to 50S/60km. All the other structures in the circulation would be lost. Such a presentation might be appropriate for a dedicated study on a specific process, but not for this paper where we present an overview over the multi-annual monthly mean results.

**Action:** None.

**The atmosphere has 4 seasons but Section 3.1 has 12 sections. This section feels very long. Conceptually this discussion ought to be about seasonal behavior, i.e., what happens near solstice or equinox, and how the transitions between them occur. Currently this section contains a blow-by-blow description of every figure panel. Please organize these descriptions around seasons and transitions, as this is how the conclusions are described.**

**Reply:** The first draft of this paper was indeed organized as suggested by the reviewer but we have given up this way to present the data. The processes we see cannot unambiguously be assigned to seasons. Each altitude regime seems to have its own seasons, and it is the smooth transition of the global picture from one month to the next that makes our results interesting. Also the processes seen cannot be categorized. Sometimes there is a smooth transition between the symmetric stratospheric circulation cells and the pole-to-pole circulation. That is to say, many of the processes seen cannot be unambiguously categorized. This had led to excessive cross-references in the first draft. Thus we have decided against reorganizing the current manuscript.

**Action:** None.

---

## Author Response (AR4)

**Comment: This paper has gone through several rounds of reviews and I think that it is appropriate that no further substantial revision is demanded before it appears in ACP. I strongly recommend the following minor changes in order to make the content of the paper as clear as possible to the reader. Please provide a revised version of the paper in which these changes or something equivalent have been made and I will then be pleased to accept the paper for publication.**

**Reply:** We are glad that on the large and whole the manuscript is now acceptable. In particular we appreciate the wording suggestions that make the revision much easier for us. The wording suggestions maintain the spirit of the original manuscript and none of them distorts the message of the paper.

**Action:** In almost all cases we have included the suggestions. For details, see below.

**Comment: l4-8: You should be explicit about what information you are presenting in the paper – to make sure that a reader does not look at your figures and imagine that they are showing something that they are not. Please modify to 'The method used for this purpose was the direct inversion of the two-dimensional continuity equation for the trace gas concentrations. This inversion predicts an 'effective velocity" that gives the best fit for the evolution of the concentrations on the assumption that Fickian diffusion can be neglected and it is this 'effective velocity' field that is used to characterise the mean meridional circulation. Multiannual monthly mean effective velocity fields are presented along with their variabilities. According to this measure the stratospheric circulation is ...'. [Note that the above is exploiting the term 'effective velocity' which I think is useful and which you have already chosen to introduce in the main body of the text.]**

**Reply:** On the large and whole, we agree, however, with some minor modifications: (1) We use not only the continuity equation of mixing ratios, but also that of air density. We now mention this in the abstract. (2) We use multiple trace gases. Thus, we have replaced 'trace gase concentration' by 'concentrations of trace gases'. (3) We assume that the contribution of Fickian diffusion is somehow aliased into the effective velocities. In order to avoid that the reader misunderstands that the effective velocities are free of any diffusion effects, we restrict the qualification to the EXPLICIT TREATMENT of Fickian diffusion. (4) With all this the new sentence is much longer. Thus we split it.

**Action:** The related part of the abstract now reads: "direct inversion of the two-dimensional continuity equation for the concentrations of trace gases and air density. This inversion predicts an 'effective velocity' that gives the best fit for the evolution of the concentrations on the assumption that an explicit

treatment of Fickian diffusion can be neglected. These 'effective velocity' fields are used to characterise the mean meridional circulation. Multiannual monthly mean effective velocity fields are presented along with their variabilities. According to this measure the stratospheric circulation ..."

**Comment: l55:** 'Our results contain considerably more information ... ( ... spent in the stratosphere).' – this doesn't make any sense at this point because you have said nothing at all about what 'your results' are. I recommend moving to the end of the following paragraph.

**Reply:** Fully agreed.

**Action:** Moved as suggested.

**Comment: l96:** 'and (optionally) mixing coefficients' – it is vital that the reader understands that mixing coefficients are **NOT** being determined by the method that you actually use – please remove 'and (optionally) mixing coefficients' – if you wish you could follow with a parenthetical separate sentence – '(In principle it is possible also to determine mixing coefficients from this inversion, but that is not done in the calculations used for this paper.)'.

**Reply:** Fully agreed

**Action:** Changed as suggested.

**Comment:** [The following comments are less important – it is up to you whether you take account of them or not.] **l9:** 'The deep branch ... are not separate but intertwined phenomena' – personally I think that it would be better to keep on emphasising that this an similar conclusions are 'according to the effective velocity characterisation of the circulation' – further work is needed to establish which (if either) of the traditional 'TEM advection + eddy mixing' description or your 'effective velocity' characterisation is the most physically meaningful.

**Reply:** Fully agreed.

**Action:** Inserted "According to the effective velocity characterisation of the circulation..."

**Comment: l55 (again)** 'contains considerably more information .. and they provide a better time-resolved understanding ... than the age-of-air method' – that is still open to argument – if one wants a simple measure of the effect of the circulation over several years then the age-of-air measure may be more useful than information on the month-to-month time variation of the circulation.

**Reply:** We did not say that our results contain 'considerably more information than the age of air method. We said 'considerably more information than the trace gas fields and their variation with time', and a 'better time-resolved understanding of the circulation than the age-of-air method. We realize that our wording is easily misunderstood, thus we have slightly changed it.

**Action:** These lines now read: "Our results contain considerably more specific information on the circulation than the trace gas fields and their variation with time alone. They also provide an understanding of the circulation better resolved in space and time than the age-of-air method (which integrates over the time an air parcel spent in the stratosphere)."

**Comment: A final question that occurs to me is how well the effective velocity would function in a forward model where trace gases were assumed to be transported by advection by the effective velocity with minimal added diffusion. Do you have any information on that?**

**Reply:** We agree that this is an interesting question. We have this issue on our agenda but all related results available so far are based on an old ANCISTRUS version where the sinks were not yet included; thus we do not want to over-interprete these results.

**Action:** None for this manuscript.